# GENERALIZATION GUARANTEES OF GRADIENT DESCENT FOR SHALLOW NEURAL NETWORKS

## ABSTRACT

Recently, significant progress has been made in understanding the generalization of neural networks (NNs) trained by gradient descent (GD) using the algorithmic stability approach. However, most of the existing research has focused on one-hidden-layer NNs and has not addressed the impact of different network scaling. Here, network scaling corresponds to the normalization of the layers. In this paper, we greatly extend the previous work (Lei et al., 2022; Richards & Kuzborskij, 2021) by conducting a comprehensive stability and generalization analysis of GD for two-layer and three-layer NNs. For two-layer NNs, our results are established under general network scaling, relaxing previous conditions. In the case of three-layer NNs, our technical contribution lies in demonstrating its nearly co-coercive property by utilizing a novel induction strategy that thoroughly explores the effects of over-parameterization. As a direct application of our general findings, we derive the excess risk rate of $\mathcal{O}(1/\sqrt{n})$ for GD algorithms in both two-layer and three-layer NNs. This sheds light on sufficient or necessary conditions for under-parameterized and over-parameterized NNs trained by GD to attain the desired risk rate of $\mathcal{O}(1/\sqrt{n})$. Moreover, we demonstrate that as the scaling factor increases or the network complexity decreases, less over-parameterization is required for GD to achieve the desired error rates. Additionally, under a low-noise condition, we obtain a fast risk rate of $\mathcal{O}(1/n)$ for GD in both two-layer and three-layer NNs.

## 1 INTRODUCTION

Deep neural networks (DNNs) trained by (stochastic) gradient descent (GD) have achieved great success in a wide spectrum of applications such as image recognition (Krizhevsky et al., 2017), speech recognition (Hinton et al., 2012), machine translation (Bahdanau et al., 2014), and reinforcement learning (Silver et al., 2016). In practical applications, most of the deployed DNNs are over-parameterized, i.e., the number of parameters is far larger than the size of the training data. In Zhang et al. (2016), it was empirically demonstrated that over-parameterized NNs trained with SGD can generalize well to the test data while achieving a small training error. This has triggered a surge of theoretical studies on unveiling this generalization mastery of DNNs.

In particular, norm-based generalization bounds are established using the uniform convergence approach (Bartlett et al., 2017; 2021; Golowich et al., 2018; Long & Sedghi, 2019; Neyshabur et al., 2015; 2018). However, this approach does not take the optimization algorithm and the data distribution into account. Another line of work is to consider the structure of the data distribution and provide algorithm-dependent generalization bounds. In Brutzkus et al. (2017); Li & Liang (2018), it is shown that SGD for over-parameterized two-layer NNs can achieve small generalization error under certain assumptions on the structure of the data. Allen-Zhu et al. (2019a) studies the generalization of SGD for two-layer and three-layer NNs if there exists a true (unknown) NN with low error on the data distribution. The other important line of work is the neural tangent kernel (NTK)-type approach (Arora et al., 2019a; Cao & Gu, 2019; Chizat & Bach, 2018; Nitanda et al., 2019) which shows that the model trained by GD is well approximated by the tangent space near the initialization and the generalization analysis can be reduced to those of the convex case or kernel methods. However, most of them either require a very high over-parameterization or focus on special function classes.

Recently, the appealing work Richards & Kuzborskij (2021) provides an alternative approach in a kernel-free regime, using the concept of algorithmic stability (Bousquet & Elisseeff, 2002; Hardt et al., 2016; Kuzborskij & Lampert, 2018). Specifically, it uses the model-average stability Lei &

Ying (2020a) to derive generalization bounds of GD for two-layer over-parameterized NNs. Lei et al. (2022) improves this result by deriving generalization bounds for both GD and SGD, and relaxing the over-parameterization requirement. Taheri & Thrampoulidis (2023) derives fast generalization bounds of GD with smooth activation for self-bounded losses under a separable distribution. However, the above studies only focus on two-layer NNs and a very specific network scaling.

**Contributions.** We study the stability and generalization of GD for both two-layer and three-layer NNs with generic network scaling factors. Our contributions are summarized as follows.

• We establish excess risk bounds for GD on both two-layer and three-layer NNs with general network scaling under relaxed over-parameterization conditions. As a direct application of our generalization results, we show that GD can achieve the excess risk rate $\mathcal{O}(1/\sqrt{n})$ when the network width $m$ satisfies certain qualitative conditions related to the scaling parameter $c$, the size of training data $n$, and the network complexity measured by the norm of the minimizer of the population risk (refer to Section 3 for further discussions). Further, under a low-noise condition, our excess risk rate can be improved to $\mathcal{O}(1/n)$ for both two-layer and three-layer NNs.

• A crucial technical element in the stability analysis for NNs trained by GD is establishing the almost co-coercivity of the gradient operator. This property naturally holds true for two-layer NNs due to the empirical risks' monotonically decreasing nature which no longer remains valid for three-layer NNs. Our technical contribution lies in demonstrating that the nearly co-coercive property still holds valid throughout the trajectory of GD. To achieve this, we employ a novel induction strategy that fully explores the effects of over-parameterization (refer to Section 4 for further discussions). Furthermore, we are able to eliminate a critical assumption made in Lei et al. (2022) regarding an inequality associated with the population risk of GD's iterates (see Remark 2 for additional details).

• Our results characterize a quantitative condition in terms of network complexity and scaling factor under which GD for two-layer and three-layer NNs can achieve the excess risk rate $\mathcal{O}(1/\sqrt{n})$ in under-parameterization and over-parameterization regimes. Our results shed light on sufficient or necessary conditions for under-parameterized and over-parameterized NNs trained by GD to achieve the desired risk rate $\mathcal{O}(1/\sqrt{n})$. In addition, our results show that as the scaling factor increases or the network complexity decreases, less over-parameterization is needed for GD to achieve the desired error rates for tow-layer and three-layer NNs.

## 2 PROBLEM FORMULATION

Let $P$ be a probability measure defined on a sample space $\mathcal{Z} = \mathcal{X} \times \mathcal{Y}$, where $\mathcal{X} \subseteq \mathbb{R}^d$ and $\mathcal{Y} \subseteq \mathbb{R}$. Let $S = \{z_i = (\mathbf{x}_i, y_i)\}_{i=1}^n$ be a training dataset drawn from $P$. One aims to build a prediction model $f_{\mathbf{W}} : \mathcal{X} \mapsto \mathbb{R}$ parameterized by $\mathbf{W}$ in some parameter space $\mathcal{W}$ based on $S$. The performance of $f_{\mathbf{W}}$ can be measured by the population risk defined as $L(f_{\mathbf{W}}) = \frac{1}{2} \iint_{\mathcal{X} \times \mathcal{Y}} \left( f_{\mathbf{W}}(\mathbf{x}) - y \right)^2 dP(\mathbf{x}, y)$. The corresponding empirical risk is defined as

$$L_S(f_{\mathbf{W}}) = \frac{1}{2n} \sum_{i=1}^n \left( f_{\mathbf{W}}(\mathbf{x}_i) - y_i \right)^2. \tag{1}$$

We denote by $\ell(\mathbf{W}; z) = \frac{1}{2}(f_{\mathbf{W}}(\mathbf{x}) - y)^2$ the loss function of $\mathbf{W}$ on a data point $z = (\mathbf{x}, y)$. The best possible model is the regression function $f^*$ defined as $f^*(\mathbf{x}) = \mathbb{E}[y|\mathbf{x}]$, where $\mathbb{E}[\cdot|\mathbf{x}]$ is the conditional expectation given $\mathbf{x}$. In this paper, we consider the prediction model $f_{\mathbf{W}}$ with a neural network structure. In particular, we are interested in two-layer and three-layer fully-connected NNs.

**Two-layer NNs.** A two-layer NN of width $m > 0$ and scaling parameter $c \in [1/2, 1]$ takes the form

$$f_{\mathbf{W}}(\mathbf{x}) = \frac{1}{m^c} \sum_{k=1}^m a_k \sigma(\mathbf{w}_k \mathbf{x}),$$

where $\sigma : \mathbb{R} \mapsto \mathbb{R}$ is an activation function, $\mathbf{W} = [\mathbf{w}_1^\top, \ldots, \mathbf{w}_m^\top]^\top \in \mathcal{W}$ with $\mathcal{W} = \mathbb{R}^{m \times d}$ is the weight matrix of the first layer, and a fixed $\mathbf{a} = [a_1, \ldots, a_m]$ with $a_k \in \{-1, +1\}$ is the weight of the output layer. In the above formulation, $\mathbf{w}_k \in \mathbb{R}^{1 \times d}$ denotes the weight of the edge connecting the input to the $k$-th hidden node, and $a_k$ denotes the weight of the edge connecting the $k$-th hidden node to the output node. The output of the network is scaled by a factor $m^{-c}$ that is decreasing with the network width $m$. Two popular choices of scaling are Neural Tangent Kernel (NTK) (Allen-Zhu et al., 2019b; Arora et al., 2019a;b; Jacot et al., 2018; Du et al., 2018; 2019) with $c = 1/2$ and mean field (Chizat & Bach, 2018; Chizat et al., 2019; Mei et al., 2019; 2018) with $c = 1$. Richards & Rabbat (2021) studied two-layer NNs with $c \in [1/2, 1]$ and discussed the influence of the scaling

by trading off it with the network complexity. We also focus on the scaling $c \in [1/2, 1]$ to ensure that meaningful generalization bounds can be obtained. We fix the output layer weight $\mathbf{a}$ and only optimize the first layer weight $\mathbf{W}$ in this setting.

**Three-layer NNs.** For a matrix $\mathbf{W}$, let $\mathbf{W}_{s:}$ and $\mathbf{W}_{is}$ denote the $s$-th row and the $(i, s)$-th entry of $\mathbf{W}$. For an input $\mathbf{x} \in \mathbb{R}^d$, a three-layer fully-connected NN with width $m > 0$ and scaling $c \in (1/2, 1]$ is

$$f_{\mathbf{W}}(\mathbf{x}) = \frac{1}{m^c} \sum_{i=1}^{m} a_i \sigma\big(\frac{1}{m^c} \sum_{s=1}^{m} \mathbf{W}_{is}^{(2)} \sigma(\mathbf{W}_{s:}^{(1)} \mathbf{x})\big),$$

where $\sigma : \mathbb{R} \mapsto \mathbb{R}$ is an activation function, $\mathbf{W} = [\mathbf{W}^{(1)}, \mathbf{W}^{(2)}] \in \mathcal{W} = \mathbb{R}^{m \times (d+m)}$ is the weight matrix of the neural network, and $\mathbf{a} = [a_1, \ldots, a_m]$ with $a_i \in \{-1, +1\}$ is the fixed output layer weight. Here, $\mathbf{W}^{(1)} \in \mathbb{R}^{m \times d}$ and $\mathbf{W}^{(2)} \in \mathbb{R}^{m \times m}$ are the weights at the first and the second layer, respectively. For simplicity, we assume that the widths of each hidden layer are the same. Our findings can readily extend to scenarios where the widths and scaling factors vary across layers. We consider the setting of optimizing the weights in both the first and the second layers.

We consider the gradient descent to solve the minimization problem: $\min_{\mathbf{W}} L_S(f_{\mathbf{W}})$. For simplicity, let $L(\mathbf{W}) = L(f_{\mathbf{W}})$ and $L_S(\mathbf{W}) = L_S(f_{\mathbf{W}})$.

**Definition 1** (Gradient Descent). Let $\mathbf{W}_0 \in \mathcal{W}$ be an initialization point, and $\{\eta_t : t \in \mathbb{N}\}$ be a sequence of step sizes. Let $\nabla$ denote the gradient operator. At iteration $t$, the update rule of GD is

$$\mathbf{W}_{t+1} = \mathbf{W}_t - \eta_t \nabla L_S(\mathbf{W}_t). \tag{2}$$

We say the gradient operator $\nabla L_S(\mathbf{W})$ is $\epsilon$-almost co-coercivity with $\epsilon > 0$ and $\alpha \geq 0$ if $\langle \mathbf{W} - \mathbf{W}', \nabla L_S(\mathbf{W}) - \nabla L_S(\mathbf{W}') \rangle \geq \alpha \|\nabla L_S(\mathbf{W}) - \nabla L(\mathbf{W}')\|_2^2 - \epsilon \|\mathbf{W} - \mathbf{W}' - \eta_t(\nabla L_S(\mathbf{W}) - \nabla L_S(\mathbf{W}'))\|_2^2$.

**Target of Analysis.** Let $\mathbf{W}^* = \arg\min_{\mathbf{W} \in \mathcal{W}} L(\mathbf{W})$ where the minimizer is chosen to be the one enjoying the smallest norm. For a randomized algorithm $\mathcal{A}$ to solve (1), let $\mathcal{A}(S)$ be the output of $\mathcal{A}$ applied to the dataset $S$. The generalization performance of $\mathcal{A}(S)$ is measured by its *excess population risk*, i.e., $L(\mathcal{A}(S)) - L(\mathbf{W}^*)$. In this paper, we are interested in studying the excess population risk of models trained by GD for both two-layer and three-layer NNs.

For any $\lambda > 0$, let $\mathbf{W}_\lambda^* = \arg\min_{\mathbf{W}} \{L(\mathbf{W}) + \frac{\lambda}{2} \|\mathbf{W} - \mathbf{W}_0\|_2^2\}$. We use the following error decomposition

$$\mathbb{E}[L(\mathcal{A}(S))] - L(\mathbf{W}^*) = \mathbb{E}\big[L(\mathcal{A}(S)) - L_S(\mathcal{A}(S))\big] + \mathbb{E}\big[L_S(\mathcal{A}(S)) - L_S(\mathbf{W}_\lambda^*) - \frac{\lambda}{2} \|\mathbf{W}_\lambda^* - \mathbf{W}_0\|_2^2\big]$$

$$+ \big[L(\mathbf{W}_\lambda^*) + \frac{\lambda}{2} \|\mathbf{W}_\lambda^* - \mathbf{W}_0\|_2^2 - L(\mathbf{W}^*)\big], \tag{3}$$

where we use $\mathbb{E}[L_S(\mathbf{W}_\lambda^*)] = L(\mathbf{W}_\lambda^*)$. Here, we denote $\|\cdot\|_2$ the standard Euclidean norm of a vector or a matrix, and $\|\cdot\|_{op}$ the spectral norm of a matrix. The first term in (3) is called the *generalization error*, which can be controlled by stability analysis. The second term is the *optimization error*, which can be estimated by tools from optimization theory. The third term, denoted by $\Lambda_\lambda = L(\mathbf{W}_\lambda^*) + \frac{\lambda}{2} \|\mathbf{W}_\lambda^* - \mathbf{W}_0\|_2^2 - L(\mathbf{W}^*)$, is the *approximation error* which will be estimated by introducing an assumption on the complexity of the NN (Assumption 3 in Section 3.1).

We use the following on-average argument stability (Lei & Ying, 2020a) to study generalization error.

**Definition 2** (On-average argument stability). Let $S = \{z_i\}_{i=1}^n$ and $S' = \{z_i'\}_{i=1}^n$ be drawn i.i.d. from an unknown distribution $P$. For any $i \in [n] =: \{1, \ldots, n\}$, define $S^{(i)} = \{z_1, \ldots, z_{i-1}, z_i', z_{i+1}, \ldots, z_n\}$ as the set formed from $S$ by replacing the $i$-th element with $z_i'$. We say $\mathcal{A}$ is on-average argument $\epsilon$-stable if $\mathbb{E}_{S,S',\mathcal{A}}\big[\frac{1}{n} \sum_{i=1}^n \|\mathcal{A}(S) - \mathcal{A}(S^{(i)})\|_2^2\big] \leq \epsilon^2$.

Our analysis requires the following standard assumptions (Lei et al., 2022; Richards & Kuzborskij, 2021). Both sigmoid and hyperbolic tangent activation functions satisfy Assumption 1.

**Assumption 1** (Activation function). *The activation function $a \mapsto \sigma(a)$ is continuous and twice differentiable with $|\sigma(a)| \leq B_\sigma$, $|\sigma'(a)| \leq B_{\sigma'}$ and $|\sigma''(a)| \leq B_{\sigma''}$, where $B_\sigma, B_{\sigma'}, B_{\sigma''} > 0$.*

**Assumption 2** (Inputs, labels and loss). *There exist constants $c_{\mathbf{x}}, c_y, c_0 > 0$ such that $\|\mathbf{x}\|_2 \leq c_{\mathbf{x}}$, $|y| \leq c_y$ and $\max\{\ell(\mathbf{0}; z), \ell(\mathbf{W}_0; z)\} \leq c_0$ for any $\mathbf{x} \in \mathcal{X}, y \in \mathcal{Y}$ and $z \in \mathcal{Z}$.*

Assumption 2 requires that the loss is uniformly bounded when evaluated under the initialization. One can verify that this requirement holds true for some widely used activations including sigmoid activation and hyperbolic tangent activation for any initialization.

## 3 MAIN RESULTS

In this section, we present our main results on the excess population risk of NNs trained by GD.

### 3.1 TWO-LAYER NEURAL NETWORKS WITH SCALING PARAMETERS

We say a function $\ell : \mathcal{W} \times \mathcal{Z} \to \mathbb{R}$ is $\rho$-smooth if for any $\mathbf{W}, \mathbf{W}' \in \mathcal{W}$ and $z \in \mathcal{Z}$, there holds $\ell(\mathbf{W}; z) - \ell(\mathbf{W}'; z) \le \langle \nabla \ell(\mathbf{W}'; z), \mathbf{W} - \mathbf{W}' \rangle + \frac{\rho}{2} \|\mathbf{W} - \mathbf{W}'\|_2^2$. We denote $B \asymp B'$ if there exist some universal constants $c_1, c_2 > 0$ such that $c_1 B \ge B' \ge c_2 B$, and denote $B \gtrsim B'$ ($B \lesssim B'$) if there exists a constant $c > 0$ such that $B \ge cB'$ ($cB \le B'$). Let $\rho = c_{\mathbf{x}}^2 \left( \frac{B_{\sigma'}^2 + B_\sigma B_{\sigma''}}{m^{2c-1}} + \frac{B_{\sigma''} c_y}{m^c} \right)$. The following theorem to be proved in Appendix A.1 presents generalization error bounds of GD for two-layer NNs.

**Theorem 1** (Generalization error). *Suppose Assumptions 1 and 2 hold. Let $\{\mathbf{W}_t\}$ be produced by* (2) *with $\eta_t \equiv \eta \le 1/(2\rho)$ based on S. Assume*

$$m \gtrsim \left( (\eta T)^2 (1 + \eta\rho) \sqrt{\rho(\rho\eta T + 2)}/n \right)^{\frac{2}{4c-1}} + (\eta T)^{\frac{3}{4c-1}} + (\eta T)^{\frac{1}{c}}. \tag{4}$$

*Then, for any $t \in [T]$, there holds $\mathbb{E}[L(\mathbf{W}_t) - L_S(\mathbf{W}_t)] \le \left( \frac{4e^2 \eta^2 \rho^2 t}{n^2} + \frac{4e\eta\rho}{n} \right) \sum_{j=0}^{t-1} \mathbb{E}[L_S(\mathbf{W}_j)]$.*

**Remark 1.** Theorem 1 establishes the first generalization results for general $c \in [1/2, 1]$, which shows that the requirement on $m$ becomes weaker as $c$ becomes larger. Indeed, with a typical choice $\eta T \asymp n^{\frac{c}{6\mu+2c-3}}$ (as shown in Corollary 4 below), the assumption (4) becomes $m \gtrsim n^{\frac{1}{6\mu+2c-3}(1+\frac{51-96\mu}{8(8c-3)})}$. This requirement becomes milder as $c$ increases for any $\mu \in [0, 1]$, which implies that large scaling reduces the requirement on the network width. In particular, our assumption only requires $m \gtrsim \eta T$ when $c = 1$. Lei et al. (2022) provided a similar generalization bound with an assumption $m \gtrsim (\eta T)^5/n^2 + (\eta T)^2$, and our result with $c = 1/2$ is consistent with their result.

The following theorem to be proved in Appendix A.2 develops the optimization error bounds of GD.

**Theorem 2** (Optimization error). *Suppose Assumptions 1 and 2 hold. Let $\{\mathbf{W}_t\}$ be produced by* (2) *with $\eta_t \equiv \eta \le 1/(2\rho)$ based on S. Let $\tilde{b} = c_{\mathbf{x}}^2 B_{\sigma''} \left( \frac{2B_{\sigma'} c_{\mathbf{x}}}{m^{c-1/2}} + \sqrt{2c_0} \right)$ and assume* (4) *holds. Assume*

$$m \gtrsim \left( T(\sqrt{\eta T} + \|\mathbf{W}_{\frac{1}{\eta T}}^* - \mathbf{W}_0\|_2) \left( \frac{\eta^3 \rho^2 T^2}{n^2} + \frac{\eta^2 T \rho}{n} + 1 \right) \right)^{\frac{1}{c}}. \tag{5}$$

*Then we have*

$$\mathbb{E}[L_S(\mathbf{W}_T)] - L_S(\mathbf{W}_{\frac{1}{\eta T}}^*) - \frac{1}{2\eta T} \|\mathbf{W}_{\frac{1}{\eta T}}^* - \mathbf{W}_0\|_2^2 \le \frac{2\tilde{b}}{m^c} (\sqrt{2\eta T c_0} + \|\mathbf{W}_{\frac{1}{\eta T}}^* - \mathbf{W}_0\|_2)$$

$$\times \left( \left( \frac{8e^2 \eta^3 \rho^2 T^2}{n^2} + \frac{8e\eta^2 T \rho}{n} \right) \sum_{s=0}^{T-1} \mathbb{E}[L_S(\mathbf{W}_s)] + \|\mathbf{W}_{\frac{1}{\eta T}}^* - \mathbf{W}_0\|_2^2 + \eta T [L(\mathbf{W}_{\frac{1}{\eta T}}^*) - L(\mathbf{W}^*)] \right).$$

We can combine generalization and optimization error bounds together to derive our main result of GD for two-layer NNs. It is worth mentioning that our excess population bound is dimension-independent, which is mainly due to that stability analysis focuses on the optimization process trajectory, as opposed to the uniform convergence approach which involves the complexity of function space and thus is often dimension-dependent. In addition, the following theorem holds for both random initialization and fixed initialization settings. Without loss of generality, we assume $\eta T \ge 1$.

**Theorem 3** (Excess population risk). *Suppose Assumptions 1 and 2 hold. Let $\{\mathbf{W}_t\}$ be produced by* (2) *with $\eta_t \equiv \eta \le 1/(2\rho)$. Assume* (4) *and* (5) *hold. For any $c \in [1/2, 1]$, if $\eta T m^{1-2c} = \mathcal{O}(n)$ and $m \gtrsim (\eta T(\sqrt{\eta T} + \|\mathbf{W}_{\frac{1}{\eta T}}^* - \mathbf{W}_0\|_2))^{1/c}$, then $\mathbb{E}[L(\mathbf{W}_T) - L(\mathbf{W}^*)] = \mathcal{O}\left( \frac{\eta T m^{1-2c}}{n} L(\mathbf{W}^*) + \Lambda_{\frac{1}{\eta T}} \right)$.*

**Remark 2.** Theorem 3 provides the first excess risk bounds of GD for two-layer NNs with general scaling parameter $c \in [1/2, 1]$ which recovers the previous work Richards & Kuzborskij (2021); Lei et al. (2022) with $c = 1/2$. Specifically, Richards & Kuzborskij (2021) derived the excess risk bound $\mathcal{O}\left( \frac{\eta T}{n} L(\mathbf{W}^*) + \Lambda_{\frac{1}{\eta T}} \right)$ with $m \gtrsim (\eta T)^5$, we relax this condition to $m \gtrsim (\eta T)^3$ by providing a better estimation of the smallest eigenvalue of a Hessian matrix of the empirical risk (see Lemma A.6). This bound was obtained in Lei et al. (2022) under a crucial condition $\mathbb{E}[L(\mathbf{W}_s)] \ge L(\mathbf{W}_{\frac{1}{\eta T}}^*)$ for any $s \in [T]$, which is difficult to verify in practice. Here, we are able to remove this condition by using $L(\mathbf{W}_{\frac{1}{\eta T}}^*) - L(\mathbf{W}_s) \le L(\mathbf{W}_{\frac{1}{\eta T}}^*) - L(\mathbf{W}^*) \le \Lambda_{\frac{1}{\eta T}}$ since $L(\mathbf{W}_s) \ge L(\mathbf{W}^*)$ when controlling the bound of GD iterates (see Lemma A.7 for details). Furthermore, if we ignore the effect of $\mathbf{W}_{\frac{1}{\eta T}}^*$ and $L(\mathbf{W}^*)$, Theorem 3 implies that the larger the scaling parameter $c$ is, the better the excess risk bound

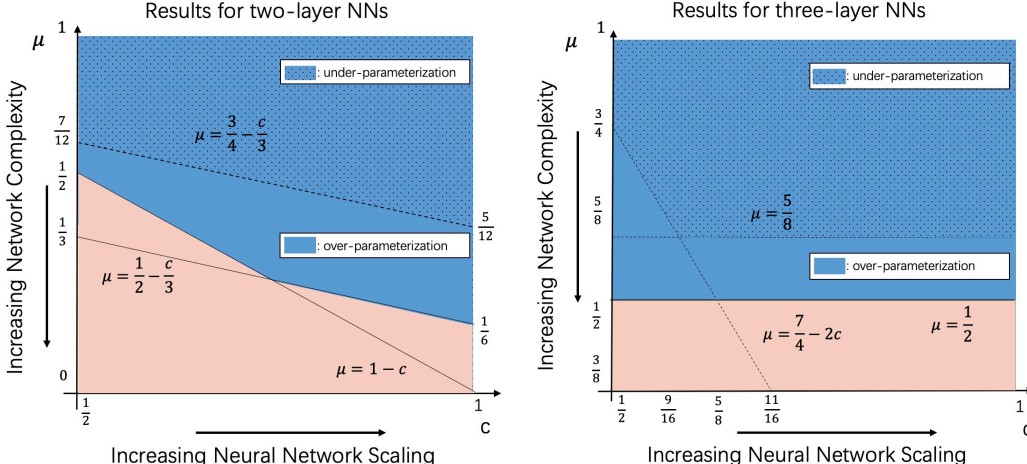

Figure 1: Scaling parameter $c$ versus Network complexity parameter $\mu$ for Part (a) in Corollary 4 (left) and Part (a) in Corollary 8 (right). *Blue Region without dots*: values of $c$ and $\mu$ where over-parameterization is necessary to achieve error bound $\mathcal{O}(1/\sqrt{n})$. *Blue Region with dots*: values of $c$ and $\mu$ where under-parameterization is sufficient to achieve error bound $\mathcal{O}(1/\sqrt{n})$. *Pink Region*: the desired bound cannot be guaranteed.

is. The reason could be that the smoothness parameter related to the objective function along the GD trajectory becomes smaller for a larger $c$ (see Lemma A.3 for details).

As a direct application of Theorem 3, we can derive the excess risk rates of GD for two-layer NNs by properly trade-offing the generalization, optimization and approximation errors. To this end, we introduce the following assumption to control the approximation error which can interpret the complexity of the network, i.e., how well the network approximates the least population risk. To achieve guarantees on the excess population risk, the approximation error should be sufficiently small, which can be interpreted as a complexity assumption of the learning problem.

**Assumption 3** (Richards & Rabbat (2021)). *There is $\mu \in [0,1]$ and population risk minimizer $\mathbf{W}^*$ with $\|\mathbf{W}^*\|_2 \leq m^{\frac{1}{2}-\mu}$.*

**Corollary 4.** *Let assumptions in Theorem 3 hold and Assumption 3 hold. Suppose $\frac{c}{3} + \mu > \frac{1}{2}$.*

(a) *If $c \in [1/2, 3/4)$ and $c + \mu \geq 1$, we can choose $m \asymp (\eta T)^{\frac{3}{2c}}$ and $\eta T$ such that $n^{\frac{c}{6\mu+2c-3}} \lesssim \eta T \lesssim n^{\frac{c}{3-4c}}$. If $c \in [3/4, 1]$, we can choose $m \asymp (\eta T)^{\frac{3}{2c}}$ and $\eta T$ such that $\eta T \gtrsim n^{\frac{c}{6\mu+2c-3}}$. Then there holds $\mathbb{E}[L(\mathbf{W}_T) - L(\mathbf{W}^*)] = \mathcal{O}(1/\sqrt{n})$.*

(b) *Assume $L(\mathbf{W}^*) = 0$. We can choose $m \asymp (\eta T)^{\frac{3}{2c}}$ and $\eta T$ such that $\eta T \gtrsim n^{\frac{2c}{6\mu+2c-3}}$, and get $\mathbb{E}[L(\mathbf{W}_T) - L(\mathbf{W}^*)] = \mathcal{O}(1/n)$.*

Corollary 4 shows that for a fixed $\mu$, the lower bound of $m$ becomes smaller as $c$ increases. It means that large $c$ relaxes the requirement to the width $m$ for GD to achieve the excess risk rate $\mathcal{O}(1/\sqrt{n})$. The difference made over $c = 3/4$ in part (a) is proof artifacts. More details is given in Appendix A.3.

**Interpretation of the Results:** Corollary 4 indicates a quantitative condition via the network complexity and scaling factor where GD for two-layer NNs can achieve the excess risk rate $\mathcal{O}(1/\sqrt{n})$ in under-parameterization and over-parameterization regimes. Here, the under-parameterization regime and over-parameterization regime correspond to the region that $m \lesssim n$ and $m \gtrsim n$, respectively.

Let us first explain the meanings of the regions and lines in the left panel of Figure 1. Specifically, the *blue regions* (with or without dots) correspond to the conditions $\frac{c}{3} + \mu > \frac{1}{2}$ and $c + \mu \geq 1$ in part (a) of Corollary 4, while our results do not hold in the *pink region* which violates the conditions in part (a). Furthermore, under conditions $\frac{c}{3} + \mu > \frac{1}{2}$ and $c + \mu \geq 1$, that the desired bound in part (a) can be achieved by choosing $m \asymp (\eta T)^{\frac{3}{2c}}$ for any $\eta T$ satisfying $\eta T \gtrsim n^{\frac{c}{6\mu+2c-3}}$ if $c \geq 3/4$ and $n^{\frac{c}{3-4c}} \gtrsim \eta T \gtrsim n^{\frac{c}{6\mu+2c-3}}$ if $c \in [1/2, 3/4)$, which further implies that GD with any width $m \gtrsim n^{\frac{3}{2(6\mu+2c-3)}}$ if $c \in [3/4, 1]$ and $n^{\frac{3}{2(3-4c)}} \gtrsim m \gtrsim n^{\frac{3}{2(6\mu+2c-3)}}$ if $c \in [1/2, 3/4)$ with suitable iterations can achieve the error rate $\mathcal{O}(1/\sqrt{n})$. This observation tells us that **the smallest width** for guaranteeing our results in part (a) is $m \asymp n^{\frac{3}{2(6\mu+2c-3)}}$ for any $c \in [1/2, 1]$. The dotted line $\mu = 3/4 - c/3$ in the figure corresponds to the setting $m \asymp n$, i.e., $\frac{3}{2(6\mu+2c-3)} = 1$. Correspondingly,

we use the *dotted blue region* above the dotted line to indicate the under-parameterization region and *the blue region without dots* below the dotted line for the over-parameterization region. We call the network under-parameterized if $m \lesssim n$ and the network over-parameterized if $m \gtrsim n$ . With the above explanations, we can interpret the left panel of Figure 1 as follows.

● Firstly, from the figure, we know that, if values of $c$ and $\mu$ are located above the dotted line $\mu \geq 3/4 - c/3$, i.e., the blue region with dots, under-parameterization is *sufficient* for GD to achieve the error rate $\mathcal{O}(1/\sqrt{n})$. It implies that the sufficient condition for under-parameterized NNs trained by GD achieving the desired rate is $\mu \geq 3/4 - c/3$. The potential reason is that the population risk minimizer $\mathbf{W}^*$ is well-behaved in terms of its norm being relatively small with $\mu$ being relatively large there. In particular, when $\mu > 1/2$, $\|\mathbf{W}^*\|_2 \leq m^{1/2-\mu}$ tends to 0 as $m$ tends to infinity. Hence, it is expected that under-parameterized NNs can learn this relatively simple $\mathbf{W}^*$ well. However, it is worthy of mentioning that over-parameterization can also achieve the rate $\mathcal{O}(1/\sqrt{n})$ since the dotted line only indicates the smallest width required for achieving such an error rate.

● Secondly, from the figure, we see that, if $c$ and $\mu$ belong to the blue region without dots which is between the solid lines and the dotted line, then over-parameterization is *necessary* for achieving the error rate $\mathcal{O}(1/\sqrt{n})$. This is because, in the blue region without dots, that the conditions of choosing $m \gtrsim n^{\frac{3}{2(6\mu+2c-3)}}$ in part (a) of Corollary 4 which will always indicate the over-parameterization region, i.e., $m \gtrsim n$. Furthermore, from the above discussions, our theoretical results indicate that the over-parameterization does bring benefit for GD to achieve good generalization in the sense that GD can achieve excess risk rate $\mathcal{O}(1/\sqrt{n})$ when $c$ and $\mu$ is in the whole blue region (with or without dots) while under-parameterization can only do so for the blue region with dots where the network complexity is relatively simple, i.e., $\mu$ is relatively large.

● Thirdly, our results do not hold for GD when values of $c$ and $\mu$ are in the pink region in the figure. In particular, when $\mu < 1/6$, our bounds do not hold for any $c \in [1/2, 1]$. We suspect that this is due to the artifacts of our analysis tools and it remains an open question to us whether we can get a generalization error bound $\mathcal{O}(1/\sqrt{n})$ when $\mu < 1/6$. In addition, our results in Corollary 4 also indicate that the requirement on $m$ becomes weaker as $c$ and $\mu$ become larger. It implies that networks with larger scaling and simpler network complexity are biased to weaken the over-parameterization for GD to achieve the desired error rates for two-layer NNs.

**Remark 3.** In Lemma A.3, we show that $f_{\mathbf{W}}$ is $B_{\sigma'}c_{\mathbf{x}}m^{\frac{1}{2}-c}$-Lipschitz. Combining this result with Assumption 3 we know $|f_{\mathbf{W}^*}(\mathbf{x}) - f_{\mathbf{0}}(\mathbf{x})|^2 = \mathcal{O}(m^{2(1-c-\mu)})$. In order for $|f_{\mathbf{W}^*}(\mathbf{x}) - f_{\mathbf{0}}(\mathbf{x})|^2$ to not vanish as $m$ tends to infinity, one needs $c + \mu \leq 1$. In Corollary 4, we also need $\frac{c}{3} + \mu > \frac{1}{2}$ to ensure the excess risk bounds vanish. Combining these two conditions together implies that $c$ can not be larger than $3/4$. That is, for the range $c \in (3/4, 1]$, the conditions in Corollary 4 restrict the class of functions the networks can represent as $m$ tends to infinity. However, we want to emphasize that even for the simplest case that $|f_{\mathbf{W}^*}(\mathbf{x}) - f_{\mathbf{0}}(\mathbf{x})|^2$ tends to 0 as $m$ tends to infinity, our results still imply that over-parameterization does bring benefit for GD to achieve optimal excess risk rate $\mathcal{O}(1/\sqrt{n})$. Besides, our corollary mainly discusses the conditions for achieving the excess risk rate $\mathcal{O}(1/\sqrt{n})$ and $\mathcal{O}(1/n)$. The above-mentioned conditions will be milder if we consider the slower excess risk rates. Then the restriction on $c$ will be weaker. Furthermore, our main result (i.e., Theorem 3) does not rely on Assumption 3, and it holds for any setting.

**Comparison with the Existing Work:** Part (b) in Corollary 4 shows fast rate $\mathcal{O}(1/n)$ can be derived under a low-noise condition $L(\mathbf{W}^*) = 0$ which is equivalent to the fact that there is a true network such that $y = f_{\mathbf{W}^*}(\mathbf{x})$ almost surely. Similar to part (a), large $c$ and large $\mu$ also help weaken the requirement on $m$ in this case. For a special case $\mu = 1/2$ and $c = 1/2$, Lei et al. (2022) proved that GD for two-layer NNs achieves the excess risk rate $\mathcal{O}(1/\sqrt{n})$ with $m \asymp n^{3/2}$ and $\eta T \asymp \sqrt{n}$ in the general case, which is further improved to $\mathcal{O}(1/n)$ with $m \asymp n^3$ and $\eta T \asymp n$ in a low-noise case. Corollary 4 recovers their results with the same conditions on $m$ and $\eta T$ for this setting.

Richards & Rabbat (2021) studied GD with weakly convex losses and showed that the excess population risk is controlled by $\mathcal{O}\big(\frac{\eta T L^2}{n} + \frac{\|\mathbf{W}_\epsilon - \mathbf{W}_0\|_2^2}{\eta T} + \epsilon(\eta T + \|\mathbf{W}^* - \mathbf{W}_0\|_2)\big)$ if $2\eta\epsilon < 1/T$ when the empirical risk is $\epsilon$-weakly convex and $L$-Lipschitz continuous, where $\mathbf{W}_\epsilon = \arg\min_{\mathbf{W}} L_S(\mathbf{W}) + \epsilon\|\mathbf{W} - \mathbf{W}_0\|_2^2$. If the approximation error is small enough, then the $\mathcal{O}(1/\sqrt{n})$ bound can be achieved by choosing $\eta T = \sqrt{n}$ if $\|\mathbf{W}_\epsilon\|_2 = \mathcal{O}(1)$. Indeed, their excess risk bound will not converge for the general case. Specifically, note that $L_S(\mathbf{W}_\epsilon) + \epsilon\|\mathbf{W}_\epsilon - \mathbf{W}_0\|_2^2 \leq L_S(0) + \epsilon\|\mathbf{W}_0\|_2^2$, then there holds $\|\mathbf{W}_\epsilon - \mathbf{W}_0\|_2^2 = \mathcal{O}(1/\epsilon)$. The simultaneous appearance of $\frac{1}{\eta T \epsilon}$ and $\eta T \epsilon$ causes the non-vanishing error bound. Richards & Rabbat (2021) also investigated the weak convexity of two-layer NNs

with a smooth activation function. Under the assumption that the derivative of the loss function is uniformly bounded by a constant, they proved that the weak convexity parameter is controlled by $\mathcal{O}(d/m^c)$. We provide a dimension-independent weak convexity parameter which further yields a dimension-independent excess risk rate $\mathcal{O}(1/\sqrt{n})$. More discussion can be found in Appendix A.3.

## 3.2 THREE-LAYER NEURAL NETWORKS WITH SCALING PARAMETERS

Now, we present our results for three-layer NNs. Let $\hat{\rho} = 4B_2(1 + 2B_1)$ and $\mathcal{B}_T = \sqrt{\eta T} + \|\mathbf{W}_0\|_2$, where $B_1, B_2 > 0$ are constants depending on $c_{\mathbf{x}}, B_{\sigma'}, B_{\sigma''}$ and $c_0$, whose specific forms are given in Appendix B.1. We first present the generalization bounds for three-layer NNs.

**Theorem 5** (Generalization error). *Suppose Assumptions 1 and 2 hold. Let* $\{\mathbf{W}_t\}$ *be produced by* (2) *with* $\eta_t \equiv \eta \le 1/(8\hat{\rho})$ *based on S. Assume*

$$m \gtrsim (\eta T)^4 + (\eta T)^{\frac{1}{4c-2}} + \|\mathbf{W}_0\|_2^{\frac{4}{8c-3}} + \|\mathbf{W}_0\|_2^{\frac{1}{6c-3}} + \big((\eta T \mathcal{B}_T)^2 + \frac{(\eta T)^{\frac{7}{2}} \mathcal{B}_T}{n}\big)^{\frac{1}{5c-\frac{1}{2}}}$$

$$+ \big((\eta T)^{\frac{3}{2}} \mathcal{B}_T^2 + \frac{(\eta T)^3 \mathcal{B}_T}{n}\big)^{\frac{1}{4c-1}} + \big((\eta T)^2 \mathcal{B}_T + \frac{(\eta T)^{\frac{7}{2}}}{n}\big)^{\frac{1}{5c-1}} + \big((\eta T)^{\frac{3}{2}} \mathcal{B}_T + \frac{(\eta T)^3}{n}\big)^{\frac{1}{4c-\frac{3}{2}}}. \quad (6)$$

*Then, for any* $t \in [T]$, $\mathbb{E}[L(\mathbf{W}_t) - L_S(\mathbf{W}_t)] \le \big(\frac{4e^2\eta^2\hat{\rho}^2 t}{n^2} + \frac{4e\eta\hat{\rho}}{n}\big) \sum_{j=0}^{t-1} \mathbb{E}[L_S(\mathbf{W}_j)]$.

Similar to Theorem 1, Theorem 5 also implies that a larger scaling $c$ relaxes the requirement on $m$.

**Remark 4.** As compared to two-layer NNs, the analysis of three-layer NNs is more challenging since we can only show that $\lambda_{\max}(\nabla^2 \ell(\mathbf{W}; z)) \le \rho_{\mathbf{W}}$ where $\rho_{\mathbf{W}}$ depends on $\|\mathbf{W}\|_2$, i.e., the smoothness parameter of $\ell$ relies on the upper bound of $\mathbf{W}$, while that of two-layer NNs is uniformly bounded. In this way, three-layer NNs do not enjoy the almost co-coercivity, which is the key step to controlling the stability of GD. To handle this problem, we first establish a crude estimate $\|\mathbf{W}_t - \mathbf{W}_0\|_2 \le \eta t m^{2c-1}$ for any $c > 1/2$ by induction strategy. Once having the estimate of $\|\mathbf{W}_t\|_2$, we can upper bound $\rho_{\mathbf{W}}$ by a constant $\hat{\rho}$ if $m$ satisfies (6). It implies that the loss is $\hat{\rho}$-smooth along the trajectory of GD. Finally, by assuming $\eta \le 1/(2\hat{\rho})$ we build the upper bound $\|\mathbf{W}_t - \mathbf{W}_0\|_2 \le \sqrt{2c_0\eta t}$. However, for the case $c = 1/2$, we cannot get a similar bound due to the condition $m^{2-4c}\|\mathbf{W}_t - \mathbf{W}_0\|_2^2 = \mathcal{O}(m^{2c-1})$. Specifically, the upper bound of $\|\mathbf{W}_t - \mathbf{W}_0\|_2$ in this case contains a worse term $2^t$, which is not easy to control. Therefore, we only consider $c \in (1/2, 1]$ for three-layer NNs. The estimate of $\|\mathbf{W}_t - \mathbf{W}_0\|_2$ when $c = 1/2$ remains an open problem. The detailed proof of the theorem is given in Appendix B.1.

Let $\mathcal{C}_{T,n} = \eta T + \eta^3 T^2/n^2$, $\hat{C}_{\mathbf{W}} = 4B_1\big(m^{-3c}(\|\mathbf{W}\|_2^2 + 2c_0\eta T) + m^{\frac{1}{2}-2c}(\|\mathbf{W}\|_2 + \sqrt{2c_0\eta T})\big)$ and $\hat{B}_{\mathbf{W}} = \big(\frac{B_{\sigma'}^2 c_{\mathbf{x}}}{m^{2c-1/2}}(\sqrt{2c_0\eta T} + \|\mathbf{W}\|_2) + \frac{B_{\sigma'}B_\sigma}{m^{2c-1}}\big)(2\sqrt{2\eta T c_0} + \|\mathbf{W}\|_2) + \sqrt{2c_0}$. The following theorem gives optimization error bounds for three-layer NNs. The proof is given in Appendix B.2.

**Theorem 6** (Optimization error). *Let Assumptions 1, 2 hold. Let* $\{\mathbf{W}_t\}$ *be produced by* (2) *with* $\eta_t \equiv \eta \le 1/(8\hat{\rho})$. *Assume $m$ is large enough such that* (6) *and*

$$m \gtrsim (\mathcal{C}_{T,n}(\mathcal{B}_T + \|\mathbf{W}^*_{\frac{1}{\eta T}}\|_2)^4)^{\frac{1}{5c-\frac{1}{2}}} + (\mathcal{C}_{T,n}(\mathcal{B}_T + \|\mathbf{W}^*_{\frac{1}{\eta T}}\|_2)^3)^{\frac{1}{4c-1}} + (\mathcal{C}_{T,n}(\mathcal{B}_T + \|\mathbf{W}^*_{\frac{1}{\eta T}}\|_2)^2)^{\frac{1}{4c-\frac{3}{2}}}$$

$$+ (\mathcal{C}_{T,n}(\mathcal{B}_T + \|\mathbf{W}^*_{\frac{1}{\eta T}}\|_2))^{\frac{1}{2c-\frac{1}{2}}}. \quad (7)$$

*Then we have*

$$\mathbb{E}[L_S(\mathbf{W}_T)] - L_S(\mathbf{W}^*_{\frac{1}{\eta T}}) - \frac{1}{2\eta T}\|\mathbf{W}^*_{\frac{1}{\eta T}} - \mathbf{W}_0\|_2^2 \le \hat{C}_{\mathbf{W}^*_{\frac{1}{\eta T}}} \hat{B}_{\mathbf{W}^*_{\frac{1}{\eta T}}} \big(\big(\frac{4e^2\eta^3 T^2 \hat{\rho}^2}{n^2} + \frac{4e\eta^2 T\hat{\rho}}{n}\big)$$

$$\times \sum_{s=0}^{T-1} \mathbb{E}[L_S(\mathbf{W}_s)] + \|\mathbf{W}^*_{\frac{1}{\eta T}} - \mathbf{W}_0\|_2^2 + \eta T[L(\mathbf{W}^*_{\frac{1}{\eta T}}) - L(\mathbf{W}^*)]\big).$$

Now, we develop excess risk bounds of GD for three-layer NNs by combining Theorem 5 and Theorem 6 together. The proof is given in Appendix B.3.

**Theorem 7** (Excess population risk). *Suppose Assumptions 1 and 2 hold. Let* $\{\mathbf{W}_t\}$ *be produced by* (2) *with* $\eta \le 1/(8\hat{\rho})$. *Assume* (6) *and* (7) *hold. For any* $c \in (1/2, 1]$, *if* $n \gtrsim \eta T$, *then there holds* $\mathbb{E}[L(\mathbf{W}_T) - L(\mathbf{W}^*)] = \mathcal{O}\big(\frac{\eta T}{n}L(\mathbf{W}^*) + \Lambda_{\frac{1}{\eta T}}\big)$.

Finally, we establish excess risk bounds of GD for three-layer NNs by assuming Assumption 3 holds.

**Corollary 8.** *Let assumptions in Theorem 7 and Assumption 3 hold. The following statements hold.*

(a) *Assume $\mu \geq 1/2$. If $c \in [9/16, 1]$, we can choose $m \asymp (\eta T)^4$ and $\eta T$ such that $n^{\frac{1}{2(8\mu-3)}} \lesssim \eta T \lesssim \sqrt{n}$. If $c \in (1/2, 9/16)$, we can choose $m \asymp (\eta T)^{\frac{1}{4c-2}}$ and $\eta T$ such that $n^{\frac{2c-1}{2\mu+4c-3}} \lesssim \eta T \lesssim \sqrt{n}$. Then $\mathbb{E}[L(\mathbf{W}_T) - L(\mathbf{W}^*)] = \mathcal{O}(1/\sqrt{n})$.*

(b) *Assume $L(\mathbf{W}^*) = 0$ and $\mu \geq 1/2$. If $c \in [9/16, 1]$, we can choose $m \asymp (\eta T)^4$ and $\eta T \gtrsim n^{\frac{1}{8\mu-3}}$. If $c \in (1/2, 9/16)$, we can choose $m \asymp (\eta T)^{\frac{1}{4c-2}}$ and $\eta T$ such that $\eta T \gtrsim n^{\frac{4c-2}{4c+2\mu-3}}$. Then $\mathbb{E}[L(\mathbf{W}_T) - L(\mathbf{W}^*)] = \mathcal{O}(1/n)$.*

**Discussion of the Results:** Part (a) in Corollary 8 shows GD for three-layer NNs can achieve the excess risk rate $\mathcal{O}(1/\sqrt{n})$ with $m \asymp (\eta T)^4$ and $n^{\frac{1}{2(8\mu-3)}} \lesssim \eta T \lesssim \sqrt{n}$ for the case $c \in [9/16, 1]$ and $m \asymp (\eta T)^{\frac{1}{4c-2}}$ and $n^{\frac{1}{2\mu+4c-3}} \lesssim \eta T \lesssim \sqrt{n}$ for the case $c \in (1/2, 9/16)$, respectively. Note that there is an additional assumption $\mu \geq 1/2$ in part (a). Combining this assumption with $\|\mathbf{W}^*\|_2 \leq m^{\frac{1}{2}-\mu}$ together, we know that the population risk minimizer $\mathbf{W}^*$ cannot be too large to reach the power of the exponent of $m$. The potential reason is that we use a constant to bound the smoothness parameter in the analysis for three-layer NNs. Part (a) also indicates a quantitative condition in terms of $m$ and $c$ where GD for three-layer NNs can achieve the excess risk rate $\mathcal{O}(1/\sqrt{n})$ in under-parameterization and over-parameterization regimes, which is interpreted in the right panel of Figure 1. The results in part (a) tell us that the smallest width for guaranteeing the desired bounds are $m \asymp n^{\frac{2}{8\mu-3}}$ for $c \in [9/16, 1]$ and $m \asymp n^{\frac{1}{2(2\mu+4c-3)}}$ for $c \in (1/2, 9/16)$. Similar to the left panel of Figure 1, the dotted lines $\mu = \frac{5}{8}$ and $\mu = \frac{7}{4} - 2c$ in the right panel of Figure 1 correspond to the setting $m \asymp n$, i.e., $\frac{2}{8\mu-3} = 1$ and $\frac{1}{2(2\mu+4c-3)} = 1$. Hence, when $c$ and $\mu$ belong to the blue region with dots, under-parameterization is *sufficient* to achieve the desired rate. When $c$ and $\mu$ are located in the blue region without dots which is between the solid lines and the dotted line, over-parameterization is *necessary* for GD to achieve the rate $\mathcal{O}(1/\sqrt{n})$. Our results for three-layer NNs also imply that the over-parameterization does bring benefit for GD to achieve good generalization in the sense that GD can achieve excess risk rate $\mathcal{O}(1/\sqrt{n})$. Under a low-noise condition, part (b) implies that the excess risk rate can be improved to $\mathcal{O}(1/n)$ with suitable choices of $m$ and $\eta T$. These results imply that the larger the scaling parameter is, the less over-parameterization is needed for GD to achieve the desired error rate for both the general case and low-noise case.

**Comparison with the Existing Work:** Richards & Rabbat (2021) studied the minimal eigenvalue of the empirical risk Hessian for a three-layer NN with a linear activation in the first layer for Lipschitz and convex losses (e.g., logistic loss), while we focus on NNs with more general activation functions for least square loss. See more detailed discussion in Appendix B.4. Ju et al. (2022) studied the generalization performance of overparameterized three-layer NTK models with the absolute loss and ReLU activation. They showed that the generalization error is in the order of $\mathcal{O}(1/\sqrt{n})$ when there are infinitely many neurons. They only trained the middle-layer weights of the networks. To the best of our knowledge, our work is the first study on the stability and generalization of GD to train both the first and the second layers of the network in the kernel-free regime.

## 4 MAIN IDEA OF THE PROOF

In this section, we present the main ideas for proving the main results in Section 3.

**Two-layer Neural Networks.** (3) decomposes the excess population risk into three terms: generalization error, optimization error and approximation error. We estimate these three terms separately.

*Generalization error.* From Lemma A.2 we know the generalization error can be upper bounded by the on-average argument stability of GD. Hence, it remains to figure out the on-average argument stability of GD. A key step in our stability analysis is to show that the loss is strongly smooth and weakly convex, which can be obtained by the following results given in Lemma A.3:

$$\lambda_{\max}(\nabla^2 \ell(\mathbf{W}; z)) \leq \rho \text{ and } \lambda_{\min}(\nabla^2 \ell(\mathbf{W}; z)) \geq -\left(c_{\mathbf{x}}^3 B_{\sigma'} B_{\sigma''} m^{\frac{1}{2}-2c} \|\mathbf{W}-\mathbf{W}_0\|_2 + c_{\mathbf{x}}^2 B_\sigma \sqrt{2c_0} m^{-c}\right).$$

Here $\rho = \mathcal{O}(m^{1-2c})$, $\lambda_{\max}(A)$ and $\lambda_{\min}(A)$ denote the largest and the smallest eigenvalue of $A$, respectively. The lower bound of $\lambda_{\min}(\nabla^2 \ell(\mathbf{W}; z))$ is related to $\|\mathbf{W} - \mathbf{W}_0\|_2$. We further show it is uniformly bounded (see the proof of Theorem A.6). Then the smoothness and weak convexity of the loss scale with $m^{1-2c}$ and $m^c$, which implies that the loss becomes more smooth and more convex for wider networks with a larger scaling.

Based on the above results, we derive the following uniform stability bounds $\|\mathbf{W}_t - \mathbf{W}_t^{(i)}\|_2 \leq 2\eta e T\sqrt{2c_0\rho(\rho\eta T+2)}/n$, where $\mathbf{W}_t^{(i)}$ is the iterates derived by applying SGD to $S^{(i)}$. Here, the weak convexity of the loss plays an important role in presenting the almost co-coercivity of the gradient operator, which helps us establish the recursive relationship of $\|\mathbf{W}_t - \mathbf{W}_t^{(i)}\|_2$. Note $\rho = \mathcal{O}(m^{1-2c}) = \mathcal{O}(1)$. Then we know $\|\mathbf{W}_t - \mathbf{W}_t^{(i)}\|_2 = \mathcal{O}(\frac{m^{1-2c}}{n}(\eta T)^{3/2})$, which implies that GD is more stable for a wider neural network with a large scaling.

*Optimization error.* A key step is to use the smoothness and weak convexity to control

$$\mathbb{E}[\|\mathbf{W}^*_{\frac{1}{\eta T}}-\mathbf{W}_t\|_2^2]\leq\big(\frac{8e^2\eta^3\rho^2t^2}{n^2}+\frac{8e\eta^2t\rho}{n}\big)\sum_{s=0}^{t-1}\mathbb{E}[L_S(\mathbf{W}_s)]+2\mathbb{E}\big[\|\mathbf{W}^*_{\frac{1}{\eta T}}-\mathbf{W}_0\|_2^2\big]+2\eta T\big[L(\mathbf{W}^*_{\frac{1}{\eta T}})-L(\mathbf{W}^*)\big]$$

in Lemma A.3. We use $L(\mathbf{W}^*_{\frac{1}{\eta T}})-L(\mathbf{W}_s) \leq L(\mathbf{W}^*_{\frac{1}{\eta T}})-L(\mathbf{W}^*) \leq \Lambda_{\frac{1}{\eta T}}$ to remove the condition $\mathbb{E}[L(\mathbf{W}_s)] \geq L(\mathbf{W}^*_{\frac{1}{\eta T}})$ for any $s \in [T]$ in Lei et al. (2022). Then the optimization error can be controlled by the monotonically decreasing of $\{L_S(\mathbf{W}_t)\}$. The proofs are given in Appendix A.2.

*Excess population risk.* Combining stability bounds, optimization bounds and approximation error together, and noting that $L(\mathbf{W}^*_{\frac{1}{\eta T}}) + \frac{1}{2\eta T}\|\mathbf{W}^*_{\frac{1}{\eta T}} - \mathbf{W}_0\|_2^2 \leq L(\mathbf{W}^*) + \frac{1}{2\eta T}\|\mathbf{W}^* - \mathbf{W}_0\|_2^2$, one can get the final error bound. The detailed proof can be found in Appendix A.3.

**Three-layer Neural Networks.** One of our technical contributions is to show that this almost co-coercivity still holds true along the GD's trajectory by a novel induction strategy fully exploring the over-parameterization. In particular, we develop stability bounds and control the generalization error by estimating the smoothness and curvature of the loss function:

$$\lambda_{\max}(\nabla^2\ell(\mathbf{W};z)) \leq \rho_{\mathbf{W}} \text{ and } \lambda_{\min}(\nabla^2\ell(\mathbf{W};z))\geq-C_{\mathbf{W}}\big(\frac{2B_{\sigma'}B_\sigma}{m^{2c-1}}\|\mathbf{W}^{(2)}\|_2 + \sqrt{2c_0}\big),$$

where $\rho_{\mathbf{W}}$ and $C_{\mathbf{W}}$ depend on $\|\mathbf{W}\|_2$. The specific forms of $\rho_{\mathbf{W}}$ and $C_{\mathbf{W}}$ are given in Appendix B.1. As mentioned in Remark 4, it is not easy to estimate $\|\mathbf{W}_t\|_2$ since the smoothness of the loss relies on the norm of $\mathbf{W}$ in this case. We address this difficulty by first giving a rough bound of $\|\mathbf{W}_t - \mathbf{W}_0\|_2$ by induction, i.e., $\|\mathbf{W}_t - \mathbf{W}_0\|_2 \leq \eta t m^{2c-1}$. Then, for any $\mathbf{W}$ produced by GD iterates, we can control $\rho_{\mathbf{W}}$ by a constant $\hat{\rho}$ if $m$ is large enough. Finally, by assuming $\eta\hat{\rho} \leq 1/2$, we prove that $\|\mathbf{W}_t - \mathbf{W}_0\|_2 \leq \sqrt{2c_0\eta t}$ and further get $\|\mathbf{W}_t\|_2 \leq \|\mathbf{W}_0\|_2 + \sqrt{2c_0\eta t}$.

After estimating $\|\mathbf{W}_t\|_2$, we can develop the following almost co-coercivity of the gradient operator:

$$\langle\mathbf{W}_t - \mathbf{W}_t^{(i)}, \nabla L_{S\backslash i}(\mathbf{W}_t) - \nabla L_{S\backslash i}(\mathbf{W}_t^{(i)})\rangle \geq 2\eta\big(1 - 4\eta\hat{\rho}\big)\big\|\nabla L_{S\backslash i}(\mathbf{W}_t) - \nabla L_{S\backslash i}(\mathbf{W}_t^{(i)})\big\|_2^2$$
$$- \tilde{\epsilon}_t\big\|\mathbf{W}_t - \mathbf{W}_t^{(i)} - \eta\big(\nabla L_{S\backslash i}(\mathbf{W}_t) - \nabla L_{S\backslash i}(\mathbf{W}_t^{(i)})\big)\big\|_2^2.$$

It helps to establish the uniform stability bound $\big\|\mathbf{W}_t-\mathbf{W}_t^{(i)}\big\|_2 \leq 2e\eta T\sqrt{2c_0\hat{\rho}(\hat{\rho}\eta T+2)}/n$. Based on stability bounds, we get generalization bounds by Lemma A.2. The proofs are given in Appendix B.1.

Similar to two-layer NNs, to estimate the optimization error for three-layer NNs, we first control $\mathbb{E}[\|\mathbf{W}^*_{\frac{1}{\eta T}} - \mathbf{W}_t\|_2^2]$ by using the smoothness and weak convexity of the loss. Then the desired bound can be obtained by the monotonically decreasing property of $\{L_S(\mathbf{W}_t)\}$. The final error bounds (Theorem 7 and Corollary 8) can be derived by plugging generalization and optimization bounds back into (3). The detailed proof can be found in Appendix B.3.

## 5 CONCLUSION

In this paper, we extend the previous work (Lei et al., 2022; Richards & Kuzborskij, 2021) by establishing the stability and generalization of GD for NNs with generic network scaling factors. Our results provide a quantitative condition that relates the scaling factor and network complexity to the achievement of the desired excess risk rate in two-layer and three-layer NNs.

There are several remaining questions for further study. Firstly, can our analysis of GD for NNs be extended to SGD with reduced computation cost? The main challenge lies in the non-monotonicity of objective functions in SGD, which is crucial for the stability analysis. Secondly, our analysis for three-layer NNs does not apply to the case of $c = 1/2$. It would be interesting to develop a result specifically for this setting. Lastly, extending our results to multiple layers would be interesting. The main difficulty lies in estimating the maximum and minimum eigenvalues of a Hessian matrix of the empirical risk, which rely on the norm of the coupled weights of different layers.

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

# Appendix for "Generalization Guarantees of Gradient Descent for Shallow Neural Networks"

## A    PROOFS OF TWO-LAYER NEURAL NETWORKS

### A.1    PROOFS OF GENERALIZATION BOUNDS

We first introduce the self-bounding property of smooth functions (Srebro et al., 2010).

**Lemma A.1** (Self-bounding property). *Suppose for all $z \in \mathcal{Z}$, the function $\mathbf{W} \mapsto \ell(\mathbf{W}; z)$ is nonnegative and $\rho$-smooth. Then $\|\nabla \ell(\mathbf{W}; z)\|_2^2 \leq 2\rho \ell(\mathbf{W}; z)$.*

The following lemma establishes the connection (Lei & Ying, 2020a) between the on-average argument stability and the generalization error.

**Lemma A.2.** *Let $\mathcal{A}$ be an algorithm. If for any $z$, the map $\mathbf{W} \mapsto \ell(\mathbf{W}; z)$ is $\rho$-smooth, then*

$$\mathbb{E}[L(\mathcal{A}(S)) - L_S(\mathcal{A}(S))] \leq \frac{\rho}{2n} \sum_{i=1}^n \mathbb{E}[\|\mathcal{A}(S) - \mathcal{A}(S^{(i)})\|_2^2] + \left(\frac{2\rho \mathbb{E}[L_S(\mathcal{A}(S))]}{n} \sum_{i=1}^n \mathbb{E}[\|\mathcal{A}(S) - \mathcal{A}(S^{(i)})\|_2^2]\right)^{\frac{1}{2}}.$$

We work with vectorized quantities so $\mathbf{W} \in \mathbb{R}^{md}$. Then $\nabla f_{\mathbf{W}}(\mathbf{x}) \in \mathbb{R}^{md}$ and $\nabla^2 f_{\mathbf{W}}(\mathbf{x}) \in \mathbb{R}^{md \times md}$. Denote by $\|\mathbf{W}\|_{op}$ the spectral norm of a matrix $\mathbf{W}$. We first introduce the following lemma, which shows that the loss function is smooth and weakly convex.

**Lemma A.3** (Smoothness and Curvature). *Suppose Assumptions 1 and 2 hold. Let $\mathbf{W}_0$ be the initial point of GD. For any fixed $\mathbf{W} \in \mathbb{R}^{m \times d}$ and any $z \in \mathcal{Z}$, there holds*

$$\lambda_{\max}(\nabla^2 \ell(\mathbf{W}; z)) \leq \rho \text{ with } \rho = c_{\mathbf{x}}^2 \left(\frac{B_{\sigma'}^2 + B_\sigma B_{\sigma''}}{m^{2c-1}} + \frac{B_{\sigma''} c_y}{m^c}\right),$$

$$\lambda_{\min}(\nabla^2 \ell(\mathbf{W}; z)) \geq -\left(\frac{c_{\mathbf{x}}^3 B_{\sigma'} B_{\sigma''}}{m^{2c-\frac{1}{2}}} \|\mathbf{W} - \mathbf{W}_0\|_2 + \frac{c_{\mathbf{x}}^2 B_\sigma \sqrt{2c_0}}{m^c}\right).$$

*Proof.* Recall that $f_{\mathbf{W}}(\mathbf{x}) = \frac{1}{m^c} \sum_{k=1}^m a_k \sigma(\mathbf{w}_k \mathbf{x})$. Let $\mathbf{v} = (\mathbf{v}_1, \dots, \mathbf{v}_m) \in \mathbb{R}^{dm}$ with $\mathbf{v}_k \in \mathbb{R}^d$ where $k \in [m]$. According to Assumption 1,2 and noting that $|a_k| = 1$, we can give the following estimations:

$$\|\nabla^2 f_{\mathbf{W}}(\mathbf{x})\|_{op} = \max_{\mathbf{v}: \|\mathbf{v}\|_2 \leq 1} \sum_{k=1}^m \frac{a_k}{m^c} \langle \mathbf{v}_k, \mathbf{x} \rangle^2 \sigma''(\mathbf{w}_k \mathbf{x})$$

$$\leq \frac{\|\mathbf{x}\|_2^2 B_{\sigma''}}{m^c} \max_{\mathbf{v}: \|\mathbf{v}\|_2 \leq 1} \sum_{k=1}^m \|\mathbf{v}_k\|_2^2$$

$$\leq \frac{c_{\mathbf{x}}^2 B_{\sigma''}}{m^c}, \tag{A.1}$$

$$\|\nabla f_{\mathbf{W}}(\mathbf{x})\|_2^2 = \sum_{k=1}^m \left\|\frac{a_k}{m^c} \mathbf{x} \sigma'(\mathbf{w}_k \mathbf{x})\right\|_2^2 \leq \frac{B_{\sigma'}^2 c_{\mathbf{x}}^2}{m^{2c-1}} \tag{A.2}$$

and

$$|f_{\mathbf{W}}(\mathbf{x}) - y| \leq |f_{\mathbf{W}}(\mathbf{x})| + c_y \leq m^{1-c} B_\sigma + c_y. \tag{A.3}$$

Note

$$\nabla^2 \ell(\mathbf{W}; z) = \nabla f_{\mathbf{W}}(\mathbf{x}) \nabla f_{\mathbf{W}}(\mathbf{x})^\top + \nabla^2 f_{\mathbf{W}}(\mathbf{x})(f_{\mathbf{W}}(\mathbf{x}) - y). \tag{A.4}$$

Then for any $\mathbf{W} \in \mathbb{R}^{md}$, we can upper bound the maximum eigenvalue of the Hessian as

$$\lambda_{\max}(\nabla^2 \ell(\mathbf{W}; z)) \leq \|\nabla f_{\mathbf{W}}(\mathbf{x})\|_2^2 + \|\nabla^2 f_{\mathbf{W}}(\mathbf{x})\|_{op} |f_{\mathbf{W}}(\mathbf{x}) - y|$$

$$\leq \frac{B_{\sigma'}^2 c_{\mathbf{x}}^2}{m^{2c-1}} + \frac{c_{\mathbf{x}}^2 B_{\sigma''}}{m^c} (m^{1-c} B_\sigma + c_y)$$

$$= c_{\mathbf{x}}^2 \left(\frac{B_{\sigma'}^2 + B_\sigma B_{\sigma''}}{m^{2c-1}} + \frac{B_{\sigma''} c_y}{m^c}\right),$$

which implies that the loss function is $\rho$-smooth with $\rho = c_{\mathbf{x}}^2 \left( \frac{B_{\sigma'}^2 + B_\sigma B_{\sigma''}}{m^{2c-1}} + \frac{B_{\sigma''} c_y}{m^c} \right)$.

For any $\mathbf{W}, \mathbf{W}' \in \mathbb{R}^{md}$, from Assumption 1 we know

$$\left| f_{\mathbf{W}}(\mathbf{x}) - f_{\mathbf{W}'}(\mathbf{x}) \right| \leq \frac{1}{m^c} \sum_{k=1}^m \left| \sigma(\mathbf{w}_k \mathbf{x}) - \sigma(\mathbf{w}_k' \mathbf{x}) \right| \leq \frac{B_{\sigma'}}{m^c} \sum_{k=1}^m \left| (\mathbf{w}_k - \mathbf{w}_k') \mathbf{x} \right| \leq \frac{c_{\mathbf{x}} B_{\sigma'}}{m^{c-1/2}} \|\mathbf{W} - \mathbf{W}'\|_2.$$
(A.5)

Combining (A.4) with the fact that $\nabla f_{\mathbf{W}}(\mathbf{x}) \nabla f_{\mathbf{W}}(\mathbf{x})^\top$ is positive semi-definite together, we obtain

$$\begin{aligned}
\lambda_{\min}(\nabla^2 \ell(\mathbf{W}; z)) &\geq -\|\nabla^2 f_{\mathbf{W}}(\mathbf{x})\|_{op} \left| f_{\mathbf{W}}(\mathbf{x}) - y \right| \\
&\geq -\frac{c_{\mathbf{x}}^2 B_{\sigma''}}{m^c} \left( \left| f_{\mathbf{W}}(\mathbf{x}) - f_{\mathbf{W}_0}(\mathbf{x}) \right| + \left| f_{\mathbf{W}_0}(\mathbf{x}) - y \right| \right) \\
&\geq -\frac{c_{\mathbf{x}}^2 B_{\sigma''}}{m^c} \left( \frac{c_{\mathbf{x}} B_{\sigma'}}{m^{c-1/2}} \|\mathbf{W} - \mathbf{W}_0\|_2 + \sqrt{2\ell(\mathbf{W}_0; z)} \right) \\
&\geq -\frac{c_{\mathbf{x}}^2 B_{\sigma''}}{m^c} \left( \frac{c_{\mathbf{x}} B_{\sigma'}}{m^{c-1/2}} \|\mathbf{W} - \mathbf{W}_0\|_2 + \sqrt{2c_0} \right),
\end{aligned}$$
(A.6)

where in the third inequality we used (A.5) with $\mathbf{W}' = \mathbf{W}_0$ and in the last inequality we used Assumption 2. The proof is completed. $\qquad\square$

To give an upper bound of the uniform stability, we need the following lemma which shows how the GD iterate will deviate from the initial point.

**Lemma A.4** (Richards & Kuzborskij (2021)). *Suppose the loss is $\rho$-smooth and $\eta \leq 1/(2\rho)$. Then for any $t \geq 0$, $i \in [n]$,*

$$\|\mathbf{W}_t - \mathbf{W}_0\|_2 \leq \sqrt{2\eta t L_S(\mathbf{W}_0)},$$

$$\|\mathbf{W}_t^{(i)} - \mathbf{W}_0\|_2 \leq \sqrt{2\eta t L_{S^{(i)}}(\mathbf{W}_0)}.$$

The following lemma shows an almost co-coercivity of the gradient operator associated with shallow neural networks. For any $i \in [n]$, define $S^{(i)} = \{z_1, \ldots, z_{i-1}, z_i', z_{i+1}, \ldots, z_n\}$ as the set formed from $S$ by replacing the $i$-th element with $z_i'$. For any $\mathbf{W} \in \mathcal{W}$,

$$L_{S \setminus i}(\mathbf{W}) = L_S(\mathbf{W}) - \frac{1}{n} \ell(\mathbf{W}; z_i) = L_{S^{(i)}}(\mathbf{W}) - \frac{1}{n} \ell(\mathbf{W}; z_i').$$

Let $\{\mathbf{W}_t\}$ and $\{\mathbf{W}_t^{(i)}\}$ be the sequence produced by GD based on $S$ and $S^{(i)}$, respectively.

**Lemma A.5** (Almost Co-coercivity of the Gradient Operator). *Suppose the loss is $\rho$-smooth and $\eta \leq 1/(2\rho)$. Then*

$$\begin{aligned}
\langle \mathbf{W}_t - \mathbf{W}_t^{(i)}, \nabla L_{S \setminus i}(\mathbf{W}_t) - \nabla L_{S \setminus i}(\mathbf{W}_t^{(i)}) \rangle &\geq 2\eta \left( 1 - \frac{\eta\rho}{2} \right) \left\| \nabla L_{S \setminus i}(\mathbf{W}_t) - \nabla L_{S \setminus i}(\mathbf{W}_t^{(i)}) \right\|_2^2 \\
&\quad - \epsilon_t \left\| \mathbf{W}_t - \mathbf{W}_t^{(i)} - \eta \left( \nabla L_{S \setminus i}(\mathbf{W}_t) - \nabla L_{S \setminus i}(\mathbf{W}_t^{(i)}) \right) \right\|_2^2,
\end{aligned}$$

*where $\epsilon_t = \frac{c_{\mathbf{x}}^2 B_{\sigma''}}{m^c} \left( \frac{c_{\mathbf{x}} B_{\sigma'}}{m^{c-1/2}} (1 + \eta\rho) \left\| \mathbf{W}_t - \mathbf{W}_t^{(i)} \right\|_2 + \frac{c_{\mathbf{x}} B_{\sigma'} \sqrt{2\eta T c_0}}{m^{c-1/2}} + \sqrt{2c_0} \right)$.*

*Proof.* This lemma can be proved in a similar way as Lemma 5 in Richards & Kuzborskij (2021) except the estimation of the eigenvalue of Hessian matrix. Specifically, for $\alpha \in [0, 1]$, let $\mathbf{W}(\alpha) = \alpha \mathbf{W}_t + (1 - \alpha) \mathbf{W}_t^{(i)} - \alpha\eta \left( \nabla \ell(\mathbf{W}_t; z) - \nabla \ell(\mathbf{W}_t^{(i)}; z) \right)$. According to (A.1) and (A.4), for any $\mathbf{W} \in \mathcal{W}$, we know

$$\lambda_{\min}\left( \nabla^2 L_{S \setminus i}(\mathbf{W}) \right) \geq -\frac{c_{\mathbf{x}}^2 B_{\sigma''}}{m^c} \left( \frac{1}{n} \sum_{j \in [n], j \neq i} \left| f_{\mathbf{W}}(\mathbf{x}_j) - y_j \right| \right).$$

Let $\mathbf{W} = \mathbf{W}(\alpha)$. Note Lemma A.3 shows that the loss is $\rho$-smooth with $\rho = c_{\mathbf{x}}^2 \big( \frac{B_{\sigma'}^2 + B_\sigma B_{\sigma''}}{m^{2c-1}} + \frac{B_{\sigma''} c_y}{m^c} \big)$. Then from (A.5) and the smoothness of $\ell$ we can get

$$\frac{1}{n} \sum_{j \in [n], j \neq i} |f_{\mathbf{W}(\alpha)}(\mathbf{x}_j) - y_j|$$

$$\leq \frac{1}{n} \sum_{j \in [n], j \neq i} |(f_{\mathbf{W}(\alpha)}(\mathbf{x}_j) - f_{\mathbf{W}_t^{(i)}}(\mathbf{x}_j)) + (f_{\mathbf{W}_t^{(i)}}(\mathbf{x}_j) - f_{\mathbf{W}_0}(\mathbf{x}_j)) + (f_{\mathbf{W}_0}(\mathbf{x}_j) - y_j)|$$

$$\leq \frac{c_{\mathbf{x}} B_{\sigma'}}{m^{c-1/2}} \|\mathbf{W}(\alpha) - \mathbf{W}_t^{(i)}\|_2 + \frac{c_{\mathbf{x}} B_{\sigma'}}{m^{c-1/2}} \|\mathbf{W}_t^{(i)} - \mathbf{W}_0\|_2 + \sqrt{2 L_{S \setminus i}(\mathbf{W}_0)}$$

$$\leq \frac{c_{\mathbf{x}} B_{\sigma'} \alpha}{m^{c-1/2}} \big( \|\mathbf{W}_t - \mathbf{W}_t^{(i)}\|_2 + \eta \|\nabla \ell(\mathbf{W}_t; z) - \nabla \ell(\mathbf{W}_t^{(i)}; z)\|_2 \big) + \frac{c_{\mathbf{x}} B_{\sigma'} \sqrt{2 \eta T c_0}}{m^{c-1/2}} + \sqrt{2 c_0}$$

$$\leq \frac{c_{\mathbf{x}} B_{\sigma'} (1 + \eta \rho)}{m^{c-1/2}} \|\mathbf{W}_t - \mathbf{W}_t^{(i)}\|_2 + \frac{c_{\mathbf{x}} B_{\sigma'} \sqrt{2 \eta T c_0}}{m^{c-1/2}} + \sqrt{2 c_0},$$

where in the third inequality we used Lemma A.4 and $\ell(\mathbf{W}_0; z) \leq c_0$.

Combining the above two inequalities together, we get

$$\lambda_{\min}\big(\nabla^2 L_{S \setminus i}(\mathbf{W}(\alpha))\big) \geq -\frac{c_{\mathbf{x}}^2 B_{\sigma''}}{m^c} \Big( \frac{c_{\mathbf{x}} B_{\sigma'} (1 + \eta \rho)}{m^{c-1/2}} \|\mathbf{W}_t - \mathbf{W}_t^{(i)}\|_2 + \frac{c_{\mathbf{x}} B_{\sigma'} \sqrt{2 \eta T c_0}}{m^{c-1/2}} + \sqrt{2 c_0} \Big).$$

Similarly, let $\widetilde{\mathbf{W}}(\alpha) = \alpha \mathbf{W}_t^{(i)} + (1 - \alpha) \mathbf{W}_t - \alpha \eta \big( \nabla \ell(\mathbf{W}_t^{(i)}; z)) - \nabla \ell(\mathbf{W}_t; z) \big)$, we can prove that

$$\lambda_{\min}\big(\nabla^2 L_{S \setminus i}(\widetilde{\mathbf{W}}(\alpha))\big) \geq -\frac{c_{\mathbf{x}}^2 B_{\sigma''}}{m^c} \Big( \frac{c_{\mathbf{x}} B_{\sigma'} (1 + \eta \rho)}{m^{c-1/2}} \|\mathbf{W}_t - \mathbf{W}_t^{(i)}\|_2 + \frac{c_{\mathbf{x}} B_{\sigma'} \sqrt{2 \eta T c_0}}{m^{c-1/2}} + \sqrt{2 c_0} \Big).$$

The remaining arguments in proving the lemma are the same as Lemma 5 in Richards & Kuzborskij (2021). We omit the proof for simplicity. $\qquad\square$

Based on the almost co-coercivity property of the gradient operator, we give the following uniform stability theorem.

**Theorem A.6** (Uniform Stability). *Suppose Assumptions 1 and 2 hold. Let $S, S^{(i)}$ be constructed in Definition 2. Let $\{\mathbf{W}_t\}$ and $\{\mathbf{W}_t^{(i)}\}$ be produced by (2) with $\eta \leq 1/(2\rho)$ based on $S$ and $S^{(i)}$, respectively. Assume (4) holds. For any $t \in [T]$, there holds*

$$\|\mathbf{W}_t - \mathbf{W}_t^{(i)}\|_2 \leq \frac{2 \eta e T \sqrt{2 c_0 \rho (\rho \eta T + 2)}}{n}.$$

*Proof.* Recall that

$$L_{S \setminus i}(\mathbf{W}) = L_S(\mathbf{W}) - \frac{1}{n} \ell(\mathbf{W}; z_i) = L_{S^{(i)}}(\mathbf{W}) - \frac{1}{n} \ell(\mathbf{W}; z_i').$$

Note $\mathcal{W} = \mathbb{R}^{m \times d}$. Then by the update rule $\mathbf{W}_{t+1} = \mathbf{W}_t - \eta \nabla L_S(\mathbf{W}_t)$, there holds

$$\big\| \mathbf{W}_{t+1} - \mathbf{W}_{t+1}^{(i)} \big\|_2^2$$

$$= \big\| \mathbf{W}_t - \mathbf{W}_t^{(i)} - \eta \big( \nabla L_{S \setminus i}(\mathbf{W}_t) - \nabla L_{S \setminus i}(\mathbf{W}_t^{(i)}) \big) - \frac{\eta}{n} \big( \nabla \ell(\mathbf{W}_t; z_i) - \nabla \ell(\mathbf{W}_t^{(i)}; z_i') \big) \big\|_2^2$$

$$\leq (1 + p) \big\| \mathbf{W}_t - \mathbf{W}_t^{(i)} - \eta \big( \nabla L_{S \setminus i}(\mathbf{W}_t) - \nabla L_{S \setminus i}(\mathbf{W}_t^{(i)}) \big) \big\|_2^2$$

$$\quad + \frac{\eta^2 (1 + 1/p)}{n^2} \big\| \nabla \ell(\mathbf{W}_t; z_i) - \nabla \ell(\mathbf{W}_t^{(i)}; z_i') \big\|_2^2$$

$$\leq (1 + p) \big\| \mathbf{W}_t - \mathbf{W}_t^{(i)} - \eta \big( \nabla L_{S \setminus i}(\mathbf{W}_t) - \nabla L_{S \setminus i}(\mathbf{W}_t^{(i)}) \big) \big\|_2^2$$

$$\quad + \frac{2 \eta^2 (1 + 1/p)}{n^2} \Big( \big\| \nabla \ell(\mathbf{W}_t; z_i) \big\|_2^2 + \big\| \nabla \ell(\mathbf{W}_t^{(i)}; z_i') \big\|_2^2 \Big), \tag{A.7}$$

where in the first inequality we used $(a + b)^2 \leq (1 + p) a^2 + (1 + 1/p) b^2$.

According to Lemma A.5 we can get

$$\big\|\mathbf{W}_t - \mathbf{W}_t^{(i)} - \eta\big(\nabla L_{S\setminus i}(\mathbf{W}_t) - \nabla L_{S\setminus i}(\mathbf{W}_t^{(i)})\big)\big\|_2^2$$

$$= \big\|\mathbf{W}_t - \mathbf{W}_t^{(i)}\big\|_2^2 + \eta^2\big\|\nabla L_{S\setminus i}(\mathbf{W}_t) - \nabla L_{S\setminus i}(\mathbf{W}_t^{(i)})\big\|_2^2 - 2\eta\Big\langle \mathbf{W}_t - \mathbf{W}_t^{(i)}, \nabla L_{S\setminus i}(\mathbf{W}_t) - \nabla L_{S\setminus i}(\mathbf{W}_t^{(i)})\Big\rangle$$

$$\leq \big\|\mathbf{W}_t - \mathbf{W}_t^{(i)}\big\|_2^2 + \eta^2\big\|\nabla L_{S\setminus i}(\mathbf{W}_t) - \nabla L_{S\setminus i}(\mathbf{W}_t^{(i)})\big\|_2^2 - 4\eta^2\Big(1 - \frac{\eta\rho}{2}\Big)\big\|\nabla L_{S\setminus i}(\mathbf{W}_t) - \nabla L_{S\setminus i}(\mathbf{W}_t^{(i)})\big\|_2^2$$

$$+ 2\eta\epsilon_t\big\|\mathbf{W}_t - \mathbf{W}_t^{(i)} - \eta\big(\nabla L_{S\setminus i}(\mathbf{W}_t) - \nabla L_{S\setminus i}(\mathbf{W}_t^{(i)})\big)\big\|_2^2,$$

where $\epsilon_t = \frac{c_{\mathbf{x}}B_{\sigma''}}{m^c}\big(\frac{c_{\mathbf{x}}B_{\sigma'}}{m^{c-1/2}}(1 + \eta\rho)\big\|\mathbf{W}_t - \mathbf{W}_t^{(i)}\big\|_2 + \frac{c_{\mathbf{x}}B_{\sigma'}\sqrt{2\eta Tc_0}}{m^{c-1/2}} + \sqrt{2c_0}\big)$.

Rearranging the above inequality and noting that $\eta\rho \leq 1/2$, we obtain

$$(1 - 2\eta\epsilon_t)\big\|\mathbf{W}_t - \mathbf{W}_t^{(i)} - \eta\big(\nabla L_{S\setminus i}(\mathbf{W}_t) - \nabla L_{S\setminus i}(\mathbf{W}_t^{(i)})\big)\big\|_2^2$$

$$\leq \big\|\mathbf{W}_t - \mathbf{W}_t^{(i)}\big\|_2^2 + \eta^2(2\eta\rho - 3)\big\|\nabla L_{S\setminus i}(\mathbf{W}_t) - \nabla L_{S\setminus i}(\mathbf{W}_t^{(i)})\big\|_2^2 \leq \big\|\mathbf{W}_t - \mathbf{W}_t^{(i)}\big\|_2^2.$$

We can choose $m$ large enough to ensure $2\eta\epsilon_t < 1$ holds for any $t \in [T]$. Indeed, $2\eta\epsilon_t < 1$ holds as long as condition (4) holds. We will discuss it at the end of the proof. Now, plugging the above inequality back into (A.7) yields

$$\big\|\mathbf{W}_{t+1} - \mathbf{W}_{t+1}^{(i)}\big\|_2^2$$

$$\leq \frac{1+p}{1 - 2\eta\epsilon_t}\big\|\mathbf{W}_t - \mathbf{W}_t^{(i)}\big\|_2^2 + \frac{2\eta^2(1 + 1/p)}{n^2}\Big(\big\|\nabla\ell(\mathbf{W}_t; z_i)\big\|_2^2 + \big\|\nabla\ell(\mathbf{W}_t^{(i)}; z_i')\big\|_2^2\Big). \quad \text{(A.8)}$$

We can apply (A.8) recursively and derive

$$\big\|\mathbf{W}_{t+1} - \mathbf{W}_{t+1}^{(i)}\big\|_2^2 \leq \frac{2\eta^2(1 + 1/p)}{n^2}\sum_{j=0}^{t}\Big(\big\|\nabla\ell(\mathbf{W}_j; z_i)\big\|_2^2 + \big\|\nabla\ell(\mathbf{W}_j^{(i)}; z_i')\big\|_2^2\Big)\prod_{\tilde{j}=j+1}^{t}\frac{1+p}{1 - 2\eta\epsilon_{\tilde{j}}},$$

$$\text{(A.9)}$$

where we used $\mathbf{W}_0 = \mathbf{W}_0^{(i)}$.

According to Lemma A.1 and Lemma A.4, we know

$$\|\nabla\ell(\mathbf{W}_j; z)\|_2^2 \leq 2\|\nabla\ell(\mathbf{W}_j; z) - \nabla\ell(\mathbf{W}_0; z)\|_2^2 + 2\|\nabla\ell(\mathbf{W}_0; z)\|_2^2$$

$$\leq 2\rho^2\|\mathbf{W}_j - \mathbf{W}_0\|_2^2 + 4\rho\ell(\mathbf{W}_0; z) \leq 4\rho^2\eta j L_S(\mathbf{W}_0) + 4\rho\ell(\mathbf{W}_0; z).$$

Similarly, we can show that

$$\|\nabla\ell(\mathbf{W}_j^{(i)}; z)\|_2^2 \leq 4\rho^2\eta j L_{S^{(i)}}(\mathbf{W}_0) + 4\rho\ell(\mathbf{W}_0; z).$$

Combining the above three inequalities together, we get

$$\big\|\mathbf{W}_{t+1} - \mathbf{W}_{t+1}^{(i)}\big\|_2^2$$

$$\leq \frac{8\rho\eta^2(1 + 1/p)}{n^2}\sum_{j=0}^{t}\Big(\rho\eta j L_S(\mathbf{W}_0) + \ell(\mathbf{W}_0; z_i) + \rho\eta j L_{S^{(i)}}(\mathbf{W}_0) + \ell(\mathbf{W}_0; z_i')\Big)\prod_{\tilde{j}=j+1}^{t}\frac{1+p}{1 - 2\eta\epsilon_{\tilde{j}}}$$

$$\leq \frac{8\rho\eta^2(1 + 1/p)}{n^2}\prod_{\tilde{j}=1}^{t}\frac{1+p}{1 - 2\eta\epsilon_{\tilde{j}}}\sum_{j=0}^{t}\Big(\rho\eta j L_S(\mathbf{W}_0) + \ell(\mathbf{W}_0; z_i) + \rho\eta j L_{S^{(i)}}(\mathbf{W}_0) + \ell(\mathbf{W}_0; z_i')\Big)$$

$$= \frac{4\rho\eta^2(1 + 1/p)}{n^2}\prod_{\tilde{j}=1}^{t}\frac{1+p}{1 - 2\eta\epsilon_{\tilde{j}}}\Big(\rho\eta t(t+1)\big(L_S(\mathbf{W}_0) + L_{S^{(i)}}(\mathbf{W}_0)\big) + 2(t+1)\big(\ell(\mathbf{W}_0; z_i)$$

$$+ \ell(\mathbf{W}_0; z_i')\big)\Big)$$

$$\leq \frac{8\rho\eta^2 c_0(1 + 1/p)(1 + t)\big(\rho\eta t + 2\big)}{n^2}\prod_{\tilde{j}=1}^{t}\frac{1+p}{1 - 2\eta\epsilon_{\tilde{j}}},$$

where we used $\ell(\mathbf{W}_0; z) \leq c_0$ for any $z \in \mathcal{Z}$. If we further choose $p = 1/t$, then there holds

$$\left\|\mathbf{W}_{t+1} - \mathbf{W}_{t+1}^{(i)}\right\|_2^2 \leq \frac{8\rho\eta^2 c_0 e(1+t)^2(\rho\eta t + 2)}{n^2} \prod_{\tilde{j}=1}^{t} \frac{1}{1 - 2\eta\epsilon_{\tilde{j}}}, \tag{A.10}$$

where we used $(1 + 1/t)^t \leq e$.

Now, we prove by induction to show

$$\left\|\mathbf{W}_{t+1} - \mathbf{W}_{t+1}^{(i)}\right\|_2 \leq \frac{2\eta e T \sqrt{2c_0\rho(\rho\eta T + 2)}}{n}. \tag{A.11}$$

(A.11) with $k = 0$ holds trivially. Assume (A.11) holds with all $k \leq t$, i.e., for all $k \leq t$

$$\left\|\mathbf{W}_k - \mathbf{W}_k^{(i)}\right\|_2 \leq \frac{2\eta e T \sqrt{2c_0\rho(\rho\eta T + 2)}}{n}. \tag{A.12}$$

and we want to show it holds with $k = t + 1 \leq T$. Recall that $\epsilon_k = \frac{c_{\mathbf{x}} B_{\sigma''}}{m^c} \left( \frac{c_{\mathbf{x}} B_{\sigma'}}{m^{c-1/2}} (1 + \eta\rho) \|\mathbf{W}_k - \mathbf{W}_k^{(i)}\|_2 + \frac{c_{\mathbf{x}} B_{\sigma'} \sqrt{2\eta T c_0}}{m^{c-1/2}} + \sqrt{2c_0} \right)$. From (A.12), for any $\tilde{j} \leq t$, we know

$$\epsilon_{\tilde{j}} \leq \epsilon' := \frac{c_{\mathbf{x}} B_{\sigma''}}{m^c} \left( \frac{2\sqrt{2c_0\rho(\rho\eta T + 2)}\eta e T (1 + \eta\rho) c_{\mathbf{x}} B_{\sigma'}}{n m^{c-1/2}} + \frac{c_{\mathbf{x}} B_{\sigma'} \sqrt{2\eta T c_0}}{m^{c-1/2}} + \sqrt{2c_0} \right).$$

Putting the above inequality back into (A.10), we get

$$\left\|\mathbf{W}_{t+1} - \mathbf{W}_{t+1}^{(i)}\right\|_2^2 \leq \frac{8\rho\eta^2 c_0 e(1+t)^2(\rho\eta t + 2)}{n^2} \left( \frac{1}{1 - 2\eta\epsilon'} \right)^t.$$

If $m$ is large enough such that $2\eta\epsilon' \leq 1/(t+1)$, then we can show

$$\left( \frac{1}{1 - 2\eta\epsilon'} \right)^t \leq \left( \frac{1}{1 - 1/(t+1)} \right)^t \leq e. \tag{A.13}$$

Then there holds

$$\left\|\mathbf{W}_{t+1} - \mathbf{W}_{t+1}^{(i)}\right\|_2 \leq \frac{2\eta e(1+t)\sqrt{2\rho c_0(\rho\eta t + 2)}}{n} \leq \frac{2\eta e T \sqrt{2\rho c_0(\rho\eta T + 2)}}{n}. \tag{A.14}$$

Now, we discuss the conditions on $m$. Suppose $m$ satisfies the following conditions

$$m \geq C_1 \left( \frac{(\eta T)^2(1 + \eta\rho)\sqrt{\rho(\rho\eta T + 2)}}{n} \right)^{\frac{2}{4c-1}}, \, m \geq C_2(\eta T)^{\frac{3}{4c-1}} \text{ and } m \geq C_3 \left( \eta T \right)^{\frac{1}{c}},$$

where $C_1 = (8ec_{\mathbf{x}}^2 B_{\sigma'} B_{\sigma''}\sqrt{2c_0})^{\frac{2}{4c-1}}$, $C_2 = \left( 4\sqrt{2c_0} c_{\mathbf{x}}^2 B_{\sigma'} B_{\sigma''} \right)^{\frac{3}{4c-1}}$ and $C_3 = (8\sqrt{2c_0} c_{\mathbf{x}} B_{\sigma''})^{1/c}$. Then it is easy to verify that

$$\frac{2\eta c_{\mathbf{x}} B_{\sigma''}}{m^c} \left( \frac{2\sqrt{2c_0\rho(\rho\eta T + 2)}\eta e T(1 + \eta\rho)c_{\mathbf{x}} B_{\sigma'}}{n m^{c-1/2}} + \frac{c_{\mathbf{x}} B_{\sigma'}\sqrt{2\eta T c_0}}{m^{c-1/2}} + \sqrt{2c_0} \right) \leq \frac{1}{T} \leq \frac{1}{1+t},$$

which ensures that $2\eta\epsilon' \leq 1/(t+1)$, and then (A.14) holds. The proof is completed. $\square$

We can combine Theorem A.6 and Lemma A.2 together to get the upper bound of the generalization error.

*Proof of Theorem 1.* Eq.(A.9) with $p = 1/t$ and Eq.(A.13) imply

$$\left\|\mathbf{W}_{t+1} - \mathbf{W}_{t+1}^{(i)}\right\|_2^2 \leq \frac{2e^2\eta^2(1+t)}{n^2} \sum_{j=0}^{t} \left( \left\|\nabla\ell(\mathbf{W}_j; z_i)\right\|_2^2 + \left\|\nabla\ell(\mathbf{W}_j^{(i)}; z_i')\right\|_2^2 \right)$$

$$\leq \frac{4e^2\eta^2\rho(1+t)}{n^2} \sum_{j=0}^{t} \left( \ell(\mathbf{W}_j; z_i) + \ell(\mathbf{W}_j^{(i)}; z_i') \right),$$

where in the last inequality we used self-bounding property of the smooth loss (Lemma A.1). Now, taking an average over $i \in [n]$ and using $\mathbb{E}[\ell(\mathbf{W}_j; z_i)] = \mathbb{E}[\ell(\mathbf{W}_j^{(i)}; z_i')]$, we have

$$
\frac{1}{n} \sum_{i=1}^{n} \mathbb{E}\|\mathbf{W}_{t+1} - \mathbf{W}_{t+1}^{(i)}\|_2^2 \leq \frac{4e^2\eta^2\rho(1+t)}{n^3} \sum_{j=0}^{t} \Big( \sum_{i=1}^{n} \mathbb{E}[\ell(\mathbf{W}_j; z_i)] + \mathbb{E}[\ell(\mathbf{W}_j^{(i)}; z_i')] \Big)
$$

$$
= \frac{8e^2\eta^2\rho(1+t)}{n^3} \sum_{j=0}^{t} \sum_{i=1}^{n} \mathbb{E}[\ell(\mathbf{W}_j; z_i)]
$$

$$
= \frac{8e^2\eta^2\rho(1+t)}{n^2} \sum_{j=0}^{t} \mathbb{E}[L_S(\mathbf{W}_j)].
$$

Combining the above stability bounds with Lemma A.2 together, we get

$$
\mathbb{E}[L(\mathbf{W}_t) - L_S(\mathbf{W}_t)] \leq \frac{4e^2\eta^2\rho^2 t}{n^2} \sum_{j=0}^{t-1} \mathbb{E}[L_S(\mathbf{W}_j)] + \Big( \frac{16e^2\eta^2\rho^2 t \mathbb{E}[L_S(\mathbf{W}_t)]}{n^2} \sum_{j=0}^{t-1} \mathbb{E}[L_S(\mathbf{W}_j)] \Big)^{\frac{1}{2}}
$$

$$
\leq \frac{4e^2\eta^2\rho^2 t}{n^2} \sum_{j=0}^{t-1} \mathbb{E}[L_S(\mathbf{W}_j)] + \frac{4e\eta\rho}{n} \sum_{j=0}^{t-1} \mathbb{E}[L_S(\mathbf{W}_j)],
$$

where in the last inequality we used $L_S(\mathbf{W}_t) \leq \frac{1}{t} \sum_{j=1}^{t-1} L_S(\mathbf{W}_j)$ (Richards & Kuzborskij, 2021). The proof is completed. □

## A.2 PROOFS OF OPTIMIZATION BOUNDS

Before giving the proofs of optimization error bounds, we first introduce the following lemma on the bound of GD iterates.

**Lemma A.7.** *Suppose Assumptions 1 and 2 hold, and $\eta \leq 1/(2\rho)$. Assume (4) and (5) hold. Then for any $t \in [T]$, there holds*

$$
1 \vee \mathbb{E}[\|\mathbf{W}_{\frac{1}{\eta T}}^* - \mathbf{W}_t\|_2^2]
$$

$$
\leq \Big( \frac{8e^2\eta^3\rho^2 t^2}{n^2} + \frac{8e\eta^2 t\rho}{n} \Big) \sum_{s=0}^{t-1} \mathbb{E}[L_S(\mathbf{W}_s)] + 2\mathbb{E}[\|\mathbf{W}_{\frac{1}{\eta T}}^* - \mathbf{W}_0\|_2^2] + 2\eta T [L(\mathbf{W}_{\frac{1}{\eta T}}^*) - L(\mathbf{W}^*)].
$$

*Proof.* For any $\mathbf{W}, \widetilde{\mathbf{W}} \in \mathbb{R}^{md}$ and $\alpha \in [0,1]$, define $\mathbf{W}(\alpha) := \widetilde{\mathbf{W}} + \alpha(\mathbf{W} - \widetilde{\mathbf{W}})$. Note that

$$
\lambda_{\min}(\nabla^2 L_S(\mathbf{W}(\alpha)))
$$

$$
\geq -\max_i\{\|\nabla^2 f(\mathbf{x}_i)\|_2\} \Big( \frac{1}{n} \sum_{i=1}^{n} |f_{\mathbf{W}(\alpha)}(\mathbf{x}_i) - y_i| \Big)
$$

$$
\geq -\frac{c_x^2 B_{\sigma''}}{m^c} \frac{1}{n} \Big( \sum_{i=1}^{n} \big( |f_{\mathbf{W}(\alpha)}(\mathbf{x}_i) - f_{\widetilde{\mathbf{W}}}(\mathbf{x}_i)| + |f_{\widetilde{\mathbf{W}}}(\mathbf{x}_i) - f_{\mathbf{W}_0}(\mathbf{x}_i)| + |f_{\mathbf{W}_0}(\mathbf{x}_i) - y_i| \big) \Big)
$$

$$
\geq -\frac{c_x^2 B_{\sigma''}}{m^c} \Big( \frac{B_{\sigma'} c_{\mathbf{x}}}{m^{c-1/2}} \|\mathbf{W}(\alpha) - \widetilde{\mathbf{W}}\|_2 + \frac{B_{\sigma'} c_{\mathbf{x}}}{m^{c-1/2}} \|\widetilde{\mathbf{W}} - \mathbf{W}_0\|_2 + \sqrt{2L_S(\mathbf{W}_0)} \Big)
$$

$$
\geq -\frac{c_x^2 B_{\sigma''}}{m^c} \Big( \frac{B_{\sigma'} c_{\mathbf{x}}}{m^{c-1/2}} \|\mathbf{W} - \widetilde{\mathbf{W}}\|_2 + \frac{B_{\sigma'} c_{\mathbf{x}}}{m^{c-1/2}} \|\widetilde{\mathbf{W}} - \mathbf{W}_0\|_2 + \sqrt{2c_0} \Big).
$$

Then for any $t \in [T]$, let $\widetilde{\mathbf{W}} = \mathbf{W}_t$, and define

$$
g(\alpha) := L_S(\mathbf{W}(\alpha)) + \frac{c_{\mathbf{x}}^2 B_{\sigma''}}{m^c} \frac{\alpha^2}{2} \Big( \frac{B_{\sigma'} c_{\mathbf{x}}}{m^{c-1/2}} \|\mathbf{W} - \mathbf{W}_t\|_2 + \frac{B_{\sigma'} c_{\mathbf{x}}}{m^{c-1/2}} \|\mathbf{W}_t - \mathbf{W}_0\|_2 + \sqrt{2c_0} \Big)
$$

$$
\times (1 \vee \mathbb{E}[\|\mathbf{W} - \mathbf{W}_t\|_2^2]).
$$

It is obvious that $g''(\alpha) \geq 0$. Then $g(\alpha)$ is convex in $\alpha \in [0, 1]$. Now, by convexity we know

$$g(1) - g(0) = L_S(\mathbf{W}) + \frac{c_{\mathbf{x}}^2 B_{\sigma''}}{2m^c} \Big( \frac{B_{\sigma'} c_{\mathbf{x}}}{m^{c-1/2}} \|\mathbf{W} - \mathbf{W}_t\|_2 + \frac{B_{\sigma'} c_{\mathbf{x}}}{m^{c-1/2}} \|\mathbf{W}_t - \mathbf{W}_0\|_2 + \sqrt{2c_0} \Big)$$
$$\times (1 \vee \mathbb{E}[\|\mathbf{W} - \mathbf{W}_t\|_2^2]) - L_S(\mathbf{W}_t)$$
$$\geq \langle \mathbf{W} - \mathbf{W}_t, \nabla L_S(\mathbf{W}_t) \rangle = g'(0).$$

Rearranging the above inequality we get

$$L_S(\mathbf{W}_t) \leq L_S(\mathbf{W}) + \frac{c_{\mathbf{x}}^2 B_{\sigma''}}{2m^c} \Big( \frac{B_{\sigma'} c_{\mathbf{x}}}{m^{c-1/2}} \|\mathbf{W} - \mathbf{W}_t\|_2 + \frac{B_{\sigma'} c_{\mathbf{x}}}{m^{c-1/2}} \|\mathbf{W}_t - \mathbf{W}_0\|_2 + \sqrt{2c_0} \Big)$$
$$\times (1 \vee \mathbb{E}[\|\mathbf{W} - \mathbf{W}_t\|_2^2]) - \langle \mathbf{W} - \mathbf{W}_t, \nabla L_S(\mathbf{W}_t) \rangle. \tag{A.15}$$

Combining (A.15) with the smoothness of the loss we can get

$$L_S(\mathbf{W}_{t+1}) \leq L_S(\mathbf{W}_t) + \langle \nabla L_S(\mathbf{W}_t), \mathbf{W}_{t+1} - \mathbf{W}_t \rangle + \frac{\rho}{2} \|\mathbf{W}_{t+1} - \mathbf{W}_t\|_2^2$$
$$\leq L_S(\mathbf{W}_t) - \eta \langle \nabla L_S(\mathbf{W}_t), \nabla L_S(\mathbf{W}_t) \rangle + \frac{\rho}{2} \|\mathbf{W}_{t+1} - \mathbf{W}_t\|_2^2$$
$$\leq L_S(\mathbf{W}_t) - \eta (1 - \frac{\eta \rho}{2}) \|\nabla L_S(\mathbf{W}_t)\|_2^2$$
$$\leq L_S(\mathbf{W}_t) - \frac{\eta}{2} \|\nabla L_S(\mathbf{W}_t)\|_2^2$$
$$\leq L_S(\mathbf{W}) + \frac{c_{\mathbf{x}}^2 B_{\sigma''}}{2m^c} \Big( \frac{B_{\sigma'} c_{\mathbf{x}}}{m^{c-1/2}} \|\mathbf{W} - \mathbf{W}_t\|_2 + \frac{B_{\sigma'} c_{\mathbf{x}}}{m^{c-1/2}} \|\mathbf{W}_t - \mathbf{W}_0\|_2 + \sqrt{2c_0} \Big)$$
$$\times (1 \vee \mathbb{E}[\|\mathbf{W} - \mathbf{W}_t\|_2^2]) - \langle \mathbf{W} - \mathbf{W}_t, \nabla L_S(\mathbf{W}_t) \rangle - \frac{\eta}{2} \|\nabla L_S(\mathbf{W}_t)\|_2^2,$$

where in the third inequality we used the update rule (2) and $\eta \rho \leq 1$.

According to the equality $2\langle x - y, x - z \rangle = \|x - y\|_2^2 + \|x - z\|_2^2 - \|y - z\|_2^2$, we know

$$-\langle \mathbf{W} - \mathbf{W}_t, \nabla L_S(\mathbf{W}_t) \rangle - \frac{\eta}{2} \|\nabla L_S(\mathbf{W}_t)\|_2^2 = \frac{1}{\eta} \langle \mathbf{W} - \mathbf{W}_t, \mathbf{W}_{t+1} - \mathbf{W}_t \rangle - \frac{1}{2\eta} \|\mathbf{W}_{t+1} - \mathbf{W}_t\|_2^2$$
$$= \frac{1}{2\eta} \big( \|\mathbf{W} - \mathbf{W}_t\|_2^2 - \|\mathbf{W}_{t+1} - \mathbf{W}\|_2^2 \big).$$

Then there holds

$$L_S(\mathbf{W}_{t+1}) \leq L_S(\mathbf{W}) + \frac{c_{\mathbf{x}}^2 B_{\sigma''}}{2m^c} \Big( \frac{B_{\sigma'} c_{\mathbf{x}}}{m^{c-1/2}} \|\mathbf{W} - \mathbf{W}_t\|_2 + \frac{B_{\sigma'} c_{\mathbf{x}} \sqrt{2\eta T c_0}}{m^{c-1/2}} + \sqrt{2c_0} \Big)$$
$$\times (1 \vee \mathbb{E}[\|\mathbf{W} - \mathbf{W}_t\|_2^2]) + \frac{1}{2\eta} \Big( \|\mathbf{W} - \mathbf{W}_t\|_2^2 - \|\mathbf{W}_{t+1} - \mathbf{W}\|_2^2 \Big). \tag{A.16}$$

The above inequality with $\mathbf{W} = \mathbf{W}^*_{\frac{1}{\eta T}}$ implies

$$\frac{1}{t} \sum_{s=0}^{t-1} \mathbb{E}[L_S(\mathbf{W}_s)] + \frac{\mathbb{E}\big[\|\mathbf{W}_t - \mathbf{W}^*_{\frac{1}{\eta T}}\|_2^2\big]}{2\eta t}$$

$$\leq \mathbb{E}[L(\mathbf{W}^*_{\frac{1}{\eta T}})] + \frac{\mathbb{E}\big[\|\mathbf{W}^*_{\frac{1}{\eta T}} - \mathbf{W}_0\|_2^2\big]}{2\eta t} + \frac{c_{\mathbf{x}}^2 B_{\sigma''}}{2m^c t} \sum_{s=0}^{t-1} \Big( \frac{B_{\sigma'} c_{\mathbf{x}}}{m^{c-1/2}} \mathbb{E}[\|\mathbf{W}^*_{\frac{1}{\eta T}} - \mathbf{W}_s\|_2]$$

$$+ \frac{B_{\sigma'} c_{\mathbf{x}} \sqrt{2\eta T c_0}}{m^{c-1/2}} + \sqrt{2c_0} \Big) \times (1 \vee \mathbb{E}[\|\mathbf{W}^*_{\frac{1}{\eta T}} - \mathbf{W}_s\|_2^2]).$$

Combined the above inequality with Theorem 1 implies

$$
\frac{\mathbb{E}\big[\|\mathbf{W}_t - \mathbf{W}^*_{\frac{1}{\eta T}}\|_2^2\big]}{2\eta t}
$$

$$
\leq \frac{1}{t}\sum_{s=0}^{t-1}\big[L(\mathbf{W}^*_{\frac{1}{\eta T}}) - \mathbb{E}[L(\mathbf{W}_s)]\big] + \frac{\mathbb{E}\big[\|\mathbf{W}^*_{\frac{1}{\eta T}} - \mathbf{W}_0\|_2^2\big]}{2\eta t} + \Big(\frac{4e^2\eta^2\rho^2 t}{n^2} + \frac{4e\eta\rho}{n}\Big)\sum_{s=0}^{t-1}\mathbb{E}\big[L_S(\mathbf{W}_s)\big]
$$

$$
+ \frac{c_{\mathbf{x}}^2 B_{\sigma''}}{2m^c t}\sum_{s=0}^{t-1}\Big(\frac{B_{\sigma'}c_{\mathbf{x}}}{m^{c-1/2}}\mathbb{E}[\|\mathbf{W}^*_{\frac{1}{\eta T}} - \mathbf{W}_s\|_2] + \frac{B_{\sigma'}c_{\mathbf{x}}\sqrt{2\eta T c_0}}{m^{c-1/2}} + \sqrt{2c_0}\Big)(1 \vee \mathbb{E}[\|\mathbf{W}^*_{\frac{1}{\eta T}} - \mathbf{W}_s\|_2^2])
$$

$$
\leq L(\mathbf{W}^*_{\frac{1}{\eta T}}) - L(\mathbf{W}^*) + \frac{\mathbb{E}\big[\|\mathbf{W}^*_{\frac{1}{\eta T}} - \mathbf{W}_0\|_2^2\big]}{2\eta t} + \Big(\frac{4e^2\eta^2\rho^2 t}{n^2} + \frac{4e\eta\rho}{n}\Big)\sum_{s=0}^{t-1}\mathbb{E}\big[L_S(\mathbf{W}_s)\big]
$$

$$
+ \frac{c_{\mathbf{x}}^2 B_{\sigma''}}{2m^c t}\sum_{s=0}^{t}\Big(\frac{B_{\sigma'}c_{\mathbf{x}}}{m^{c-1/2}}\mathbb{E}[\|\mathbf{W}^*_{\frac{1}{\eta T}} - \mathbf{W}_s\|_2] + \frac{B_{\sigma'}c_{\mathbf{x}}\sqrt{2\eta T c_0}}{m^{c-1/2}} + \sqrt{2c_0}\Big)(1 \vee \mathbb{E}[\|\mathbf{W}^*_{\frac{1}{\eta T}} - \mathbf{W}_s\|_2^2]),
$$

$$(A.17)$$

where in the second inequality we used $L(\mathbf{W}^*_{\frac{1}{\eta T}}) - L(\mathbf{W}_s) \leq L(\mathbf{W}^*_{\frac{1}{\eta T}}) - L(\mathbf{W}^*)$ since $L(\mathbf{W}_s) \geq L(\mathbf{W}^*)$ for any $s \in [t-1]$.

On the other hand, using Lemma A.4 we can obtain

$$
\|\mathbf{W}^*_{\frac{1}{\eta T}} - \mathbf{W}_s\|_2 \leq \|\mathbf{W}^*_{\frac{1}{\eta T}} - \mathbf{W}_0\|_2 + \|\mathbf{W}_s - \mathbf{W}_0\|_2 \leq \sqrt{2\eta T c_0} + \|\mathbf{W}^*_{\frac{1}{\eta T}} - \mathbf{W}_0\|_2. \quad (A.18)
$$

Then we know

$$
\Big(\frac{B_{\sigma'}c_{\mathbf{x}}}{m^{c-1/2}}\mathbb{E}[\|\mathbf{W}^*_{\frac{1}{\eta T}} - \mathbf{W}_s\|_2] + \frac{B_{\sigma'}c_{\mathbf{x}}\sqrt{2\eta T c_0}}{m^{c-1/2}} + \sqrt{2c_0}\Big)(1 \vee \mathbb{E}[\|\mathbf{W}^*_{\frac{1}{\eta T}} - \mathbf{W}_s\|_2^2])
$$

$$
\leq \Big(\frac{B_{\sigma'}c_{\mathbf{x}}}{m^{c-1/2}}(2\sqrt{2\eta T c_0} + \|\mathbf{W}^*_{\frac{1}{\eta T}} - \mathbf{W}_0\|_2) + \sqrt{2c_0}\Big)(1 \vee \mathbb{E}[\|\mathbf{W}^*_{\frac{1}{\eta T}} - \mathbf{W}_s\|_2^2]).
$$

Plugging the above inequality back into (A.17) yields

$$
\frac{\mathbb{E}\big[\|\mathbf{W}_t - \mathbf{W}^*_{\frac{1}{\eta T}}\|_2^2\big]}{2\eta t}
$$

$$
\leq \frac{\mathbb{E}\big[\|\mathbf{W}^*_{\frac{1}{\eta T}} - \mathbf{W}_0\|_2^2\big]}{2\eta t} + \frac{\tilde{b}(\sqrt{2\eta T c_0} + \|\mathbf{W}^*_{\frac{1}{\eta T}} - \mathbf{W}_0\|_2)}{m^c t}\sum_{s=0}^{t}(1 \vee \mathbb{E}[\|\mathbf{W}^*_{\frac{1}{\eta T}} - \mathbf{W}_s\|_2^2])
$$

$$
+ \Big(\frac{4e^2\eta^2\rho^2 t}{n^2} + \frac{4e\eta\rho}{n}\Big)\sum_{s=0}^{t-1}\mathbb{E}\big[L_S(\mathbf{W}_s)\big] + L(\mathbf{W}^*_{\frac{1}{\eta T}}) - L(\mathbf{W}^*),
$$

where $\tilde{b} = \frac{c_{\mathbf{x}}^2 B_{\sigma''}}{2}\big(\frac{2B_{\sigma'}c_{\mathbf{x}}}{m^{c-1/2}} + \sqrt{2c_0}\big)$.

Multiplying both sides by $2\eta t$ yields

$$
\mathbb{E}\big[\|\mathbf{W}_t - \mathbf{W}^*_{\frac{1}{\eta T}}\|_2^2\big]
$$

$$
\leq \mathbb{E}\big[\|\mathbf{W}^*_{\frac{1}{\eta T}} - \mathbf{W}_0\|_2^2\big] + \frac{2\tilde{b}\eta(\sqrt{2\eta T c_0} + \|\mathbf{W}^*_{\frac{1}{\eta T}} - \mathbf{W}_0\|_2)}{m^c}\sum_{s=0}^{t}(1 \vee \mathbb{E}[\|\mathbf{W}^*_{\frac{1}{\eta T}} - \mathbf{W}_s\|_2^2])
$$

$$
+ \Big(\frac{8e^2\eta^3\rho^2 t^2}{n^2} + \frac{8e\eta^2\rho t}{n}\Big)\sum_{s=0}^{t-1}\mathbb{E}\big[L_S(\mathbf{W}_s)\big] + 2\eta T\big[L(\mathbf{W}^*_{\frac{1}{\eta T}}) - L(\mathbf{W}^*)\big].
$$

Let $x = \max_{s \in [T]} \mathbb{E}[\|\mathbf{W}^*_{\frac{1}{\eta T}} - \mathbf{W}_s\|_2^2] \vee 1$. Then the above inequality implies

$$x \leq \mathbb{E}\big[\|\mathbf{W}^*_{\frac{1}{\eta T}} - \mathbf{W}_0\|_2^2\big] + \frac{2\tilde{b}\eta T(\sqrt{2\eta T c_0} + \|\mathbf{W}^*_{\frac{1}{\eta T}} - \mathbf{W}_0\|_2)}{m^c} x$$
$$+ \Big(\frac{8e^2\eta^3\rho^2 t^2}{n^2} + \frac{8e\eta^2\rho t}{n}\Big) \sum_{s=0}^{t-1} \mathbb{E}\big[L_S(\mathbf{W}_s)\big] + 2\eta T\big[L(\mathbf{W}^*_{\frac{1}{\eta T}}) - L(\mathbf{W}^*)\big].$$

Without loss of generality, we assume $\eta \leq 1$. Condition (5) implies $m \geq \big(4\tilde{b}\eta T(\sqrt{2\eta T c_0} + \|\mathbf{W}^*_{\frac{1}{\eta T}} -$

$\mathbf{W}_0\|_2)\big)^{\frac{1}{c}}$. Then there holds $\frac{2\tilde{b}\eta T(\sqrt{2\eta T c_0} + \|\mathbf{W}^*_{\frac{1}{\eta T}} - \mathbf{W}_0\|_2)}{m^c} \leq \frac{1}{2}$. Hence

$$x \leq \Big(\frac{16e^2\eta^3\rho^2 t^2}{n^2} + \frac{16e\eta^2 t\rho}{n}\Big) \sum_{s=0}^{t-1} \mathbb{E}\big[L_S(\mathbf{W}_s)\big] + 2\mathbb{E}\big[\|\mathbf{W}^*_{\frac{1}{\eta T}} - \mathbf{W}_0\|_2^2\big] + 2\eta T\big[L(\mathbf{W}^*_{\frac{1}{\eta T}}) - L(\mathbf{W}^*)\big].$$

It then follows that

$$1 \vee \mathbb{E}[\|\mathbf{W}^*_{\frac{1}{\eta T}} - \mathbf{W}_t\|_2^2]$$
$$\leq \Big(\frac{16e^2\eta^3\rho^2 t^2}{n^2} + \frac{16e\eta^2 t\rho}{n}\Big) \sum_{s=0}^{t-1} \mathbb{E}\big[L_S(\mathbf{W}_s)\big] + 2\mathbb{E}\big[\|\mathbf{W}^*_{\frac{1}{\eta T}} - \mathbf{W}_0\|_2^2\big] + 2\eta T\big[L(\mathbf{W}^*_{\frac{1}{\eta T}}) - L(\mathbf{W}^*)\big].$$

This completes the proof. $\qquad\square$

Now, we can give the proof of Theorem 2.

*Proof of Theorem 2.* Recall that $\tilde{b} = \frac{c_\mathbf{x}^2 B_{\sigma''}}{2}\big(\frac{2B_{\sigma'}c_\mathbf{x}}{m^{c-1/2}} + \sqrt{2c_0}\big)$. Eq. (A.16) with $\mathbf{W} = \mathbf{W}^*_{\frac{1}{\eta T}}$ implies

$$\frac{1}{T}\sum_{s=0}^{T-1} \mathbb{E}[L_S(\mathbf{W}_s)]$$
$$\leq \mathbb{E}[L_S(\mathbf{W}^*_{\frac{1}{\eta T}})] + \frac{\tilde{b}(\sqrt{2\eta T c_0} + \|\mathbf{W}^*_{\frac{1}{\eta T}} - \mathbf{W}_0\|_2)}{m^c T} \sum_{s=0}^{T-1} 1 \vee \mathbb{E}[\|\mathbf{W}^*_{\frac{1}{\eta T}} - \mathbf{W}_s\|_2^2] + \frac{\|\mathbf{W}^*_{\frac{1}{\eta T}} - \mathbf{W}_0\|_2^2}{2\eta T},$$
$$\text{(A.19)}$$

where in the last inequality we used (A.18).

Further, by monotonic decrease of $\{L_S(\mathbf{W}_t)\}$, we know

$$\mathbb{E}[L_S(\mathbf{W}_T)]$$
$$\leq \mathbb{E}[L_S(\mathbf{W}^*_{\frac{1}{\eta T}})] + \frac{\tilde{b}(\sqrt{2\eta T c_0} + \|\mathbf{W}^*_{\frac{1}{\eta T}} - \mathbf{W}_0\|_2)}{m^c T} \sum_{s=0}^{T-1} 1 \vee \mathbb{E}[\|\mathbf{W}^*_{\frac{1}{\eta T}} - \mathbf{W}_s\|_2^2] + \frac{\|\mathbf{W}^*_{\frac{1}{\eta T}} - \mathbf{W}_0\|_2^2}{2\eta T}.$$

Note that Lemma A.7 shows

$$1 \vee \mathbb{E}[\|\mathbf{W}^*_{\frac{1}{\eta T}} - \mathbf{W}_t\|_2^2]$$
$$\leq \Big(\frac{16e^2\eta^3\rho^2 t^2}{n^2} + \frac{16e\eta^2 t\rho}{n}\Big) \sum_{s=0}^{t-1} \mathbb{E}\big[L_S(\mathbf{W}_s)\big] + 2\mathbb{E}\big[\|\mathbf{W}^*_{\frac{1}{\eta T}} - \mathbf{W}_0\|_2^2\big] + 2\eta T\big[L(\mathbf{W}^*_{\frac{1}{\eta T}}) - L(\mathbf{W}^*)\big].$$

Combining the above two inequalities together, we get

$$\mathbb{E}[L_S(\mathbf{W}_T)]$$
$$\leq \mathbb{E}[L_S(\mathbf{W}^*_{\frac{1}{\eta T}})] + \frac{2\tilde{b}(\sqrt{2\eta T c_0} + \|\mathbf{W}^*_{\frac{1}{\eta T}} - \mathbf{W}_0\|_2)}{m^c}\Big(\Big(\frac{8e^2\eta^3\rho^2 T^2}{n^2} + \frac{8e\eta^2 T\rho}{n}\Big) \sum_{s=0}^{T-1} \mathbb{E}\big[L_S(\mathbf{W}_s)\big]$$
$$+ \|\mathbf{W}^*_{\frac{1}{\eta T}} - \mathbf{W}_0\|_2^2 + \eta T\big[L(\mathbf{W}^*_{\frac{1}{\eta T}}) - L(\mathbf{W}^*)\big]\Big) + \frac{\|\mathbf{W}^*_{\frac{1}{\eta T}} - \mathbf{W}_0\|_2^2}{2\eta T}.$$

The theorem is proved. $\qquad\square$

**Lemma A.8.** *Suppose Assumptions 1 and 2 hold. Let $\{\mathbf{W}_t\}$ be produced by (2) with $\eta \le 1/(2\rho)$. Assume (4) and (5) hold. Then*

$$\sum_{s=0}^{T-1} \mathbb{E}[L_S(\mathbf{W}_s)]$$

$$\le 4TL(\mathbf{W}^*_{\frac{1}{\eta T}}) - 2\eta TL(\mathbf{W}^*) + \Big(\frac{2\tilde{b}T(\sqrt{2\eta Tc_0} + \|\mathbf{W}^*_{\frac{1}{\eta T}} - \mathbf{W}_0\|_2)}{m^c} + \frac{1}{2\eta}\Big)\|\mathbf{W}^*_{\frac{1}{\eta T}} - \mathbf{W}_0\|_2^2.$$

*Proof.* Multiplying $T$ over both sides of (A.19) and using Lemma A.7 we get

$$\sum_{s=0}^{T-1} \mathbb{E}[L_S(\mathbf{W}_s)]$$

$$\le TL(\mathbf{W}^*_{\frac{1}{\eta T}}) + \frac{\tilde{b}(\sqrt{2\eta Tc_0} + \|\mathbf{W}^*_{\frac{1}{\eta T}} - \mathbf{W}_0\|_2)}{m^c} \sum_{s=0}^{T-1} 1 \vee \mathbb{E}[\|\mathbf{W}^*_{\frac{1}{\eta T}} - \mathbf{W}_s\|_2^2] + \frac{\|\mathbf{W}^*_{\frac{1}{\eta T}} - \mathbf{W}_0\|_2^2}{2\eta}$$

$$\le TL(\mathbf{W}^*_{\frac{1}{\eta T}}) + \frac{\tilde{b}T(\sqrt{2\eta Tc_0} + \|\mathbf{W}^*_{\frac{1}{\eta T}} - \mathbf{W}_0\|_2)}{m^c}\Big(\Big(\frac{16e^2\eta^3\rho^2T^2}{n^2} + \frac{16e\eta^2T\rho}{n}\Big) \sum_{s=0}^{T-1} \mathbb{E}[L_S(\mathbf{W}_s)]$$

$$+ 2\|\mathbf{W}^*_{\frac{1}{\eta T}} - \mathbf{W}_0\|_2^2 + 2\eta T[L(\mathbf{W}^*_{\frac{1}{\eta T}}) - L(\mathbf{W}^*)]\Big) + \frac{\|\mathbf{W}^*_{\frac{1}{\eta T}} - \mathbf{W}_0\|_2^2}{2\eta}.$$

Condition (5) implies $m \ge \big(2\tilde{b}T(\sqrt{2\eta Tc_0} + \|\mathbf{W}^*_{\frac{1}{\eta T}} - \mathbf{W}_0\|_2)\big(\frac{16e^2\eta^3\rho^2T^2}{n^2} + \frac{16e\eta^2T\rho}{n}\big)\big)^{1/c}$ and $m \ge \big(2\tilde{b}T(\sqrt{2\eta Tc_0} + \|\mathbf{W}^*_{\frac{1}{\eta T}} - \mathbf{W}_0\|_2)\big)^{1/c}$, there holds

$$\sum_{s=0}^{T-1} \mathbb{E}[L_S(\mathbf{W}_s)]$$

$$\le 4TL(\mathbf{W}^*_{\frac{1}{\eta T}}) - 2\eta TL(\mathbf{W}^*) + \Big(\frac{2\tilde{b}T(\sqrt{2\eta Tc_0} + \|\mathbf{W}^*_{\frac{1}{\eta T}} - \mathbf{W}_0\|_2)}{m^c} + \frac{1}{2\eta}\Big)\|\mathbf{W}^*_{\frac{1}{\eta T}} - \mathbf{W}_0\|_2^2,$$

which completes the proof. $\qquad\square$

### A.3 PROOFS OF EXCESS RISK BOUNDS

*Proof of Theorem 3.* According to Lemma A.8 and noting that $\tilde{b} = \mathcal{O}(1)$, we know

$$\sum_{s=0}^{T-1} \mathbb{E}[L_S(\mathbf{W}_s)] = \mathcal{O}\Big(TL(\mathbf{W}^*_{\frac{1}{\eta T}}) + \Big(\frac{T(\sqrt{\eta T} + \|\mathbf{W}^*_{\frac{1}{\eta T}} - \mathbf{W}_0\|_2)}{m^c} + \frac{1}{\eta}\Big)\|\mathbf{W}^*_{\frac{1}{\eta T}} - \mathbf{W}_0\|_2^2\Big). \quad \text{(A.20)}$$

The upper bound of the generalization error can be controlled by plugging (A.20) back into Theorem 1

$$\mathbb{E}[L(\mathbf{W}_T) - L_S(\mathbf{W}_T)]$$

$$= \mathcal{O}\Big(\Big(\frac{\eta^2\rho^2T}{n^2} + \frac{\eta\rho}{n}\Big) \sum_{s=0}^{T-1} \mathbb{E}[L_S(\mathbf{W}_s)]\Big)$$

$$= \mathcal{O}\Big(\Big(\frac{\eta^2\rho^2T^2}{n^2} + \frac{\eta T\rho}{n}\Big)\Big(L(\mathbf{W}^*_{\frac{1}{\eta T}}) + \Big(\frac{\sqrt{\eta T} + \|\mathbf{W}^*_{\frac{1}{\eta T}} - \mathbf{W}_0\|_2}{m^c} + \frac{1}{\eta T}\Big)\|\mathbf{W}^*_{\frac{1}{\eta T}} - \mathbf{W}_0\|_2^2\Big).$$

$$\text{(A.21)}$$

The estimation of the optimization error is given by plugging (A.20) back into Theorem 2

$$
\mathbb{E}[L_S(\mathbf{W}_T) - L_S(\mathbf{W}^*_{\frac{1}{\eta T}}) - \frac{1}{2\eta T}\|\mathbf{W}^*_{\frac{1}{2\eta T}} - \mathbf{W}_0\|_2^2]
$$

$$
= \mathcal{O}\Big(\frac{(\sqrt{\eta T} + \|\mathbf{W}^*_{\frac{1}{\eta T}} - \mathbf{W}_0\|_2)}{m^c}\Big[\Big(\frac{\eta^3\rho^2 T^2}{n^2} + \frac{\eta^2 T\rho}{n}\Big)\sum_{s=0}^{T-1}\mathbb{E}[L_S(\mathbf{W}_s)] + \|\mathbf{W}^*_{\frac{1}{\eta T}} - \mathbf{W}_0\|_2^2
$$

$$
+ \eta T\big[L(\mathbf{W}^*_{\frac{1}{\eta T}}) - L(\mathbf{W}^*)\big]\Big]\Big)
$$

$$
= \mathcal{O}\Big(\frac{(\sqrt{\eta T}+\|\mathbf{W}^*_{\frac{1}{\eta T}}-\mathbf{W}_0\|_2)}{m^c}\Big(\frac{\eta^3\rho^2 T^2}{n^2}+\frac{\eta^2 T\rho}{n}\Big)\Big(\frac{T(\sqrt{\eta T}+\|\mathbf{W}^*_{\frac{1}{\eta T}}-\mathbf{W}_0\|_2)}{m^c}+\frac{1}{\eta}\Big)\|\mathbf{W}^*_{\frac{1}{\eta T}}-\mathbf{W}_0\|_2^2
$$

$$
+ \frac{\eta T(\sqrt{\eta T}+\|\mathbf{W}^*_{\frac{1}{\eta T}}-\mathbf{W}_0\|_2)}{m^c}\Big(\frac{\eta^2\rho^2 T^2}{n^2}+\frac{\eta T\rho}{n}\Big)L(\mathbf{W}^*_{\frac{1}{\eta T}})
$$

$$
+ \frac{(\sqrt{\eta T}+\|\mathbf{W}^*_{\frac{1}{\eta T}}-\mathbf{W}_0\|_2)}{m^c}\big(\|\mathbf{W}^*_{\frac{1}{\eta T}}-\mathbf{W}_0\|_2^2+\eta T\Lambda_{\frac{1}{\eta T}}\big)\Big), \tag{A.22}
$$

where we used the fact that $L(\mathbf{W}^*_{\frac{1}{\eta T}}) - L(\mathbf{W}^*) \le \Lambda_{\frac{1}{\eta T}}$.

Combining (A.21) and (A.22) together and noting that the approximation error $\Lambda_{\frac{1}{\eta T}} = L(\mathbf{W}^*_{\frac{1}{\eta T}}) + \frac{1}{2\eta T}\|\mathbf{W}^*_{\frac{1}{\eta T}} - \mathbf{W}_0\|_2^2 - L(\mathbf{W}^*)$ we get

$$
\mathbb{E}[L(\mathbf{W}_T) - L(\mathbf{W}^*)]
$$

$$
= \Big[\mathbb{E}[L(\mathbf{W}_T) - L_S(\mathbf{W}_T)] + \mathbb{E}\Big[L_S(\mathbf{W}_T) - \big(L_S(\mathbf{W}^*_{\frac{1}{\eta T}}) + \frac{1}{2\eta T}\|\mathbf{W}^*_{\frac{1}{\eta T}} - \mathbf{W}_0\|_2^2\big)\Big]\Big]
$$

$$
+ \Big[L(\mathbf{W}^*_{\frac{1}{\eta T}}) + \frac{1}{2\eta T}\|\mathbf{W}^*_{\frac{1}{\eta T}} - \mathbf{W}_0\|_2^2 - L(\mathbf{W}^*)\Big]
$$

$$
= \mathcal{O}\Big(\frac{\eta T\rho}{n}\Big(\frac{\eta\rho T}{n}+1\Big)\Big(1+\frac{\eta T(\sqrt{\eta T}+\|\mathbf{W}^*_{\frac{1}{\eta T}}-\mathbf{W}_0\|_2)}{m^c}\Big)\Big[L(\mathbf{W}^*_{\frac{1}{\eta T}})+\Big(\frac{1}{2\eta T}+\frac{\sqrt{\eta T}+\|\mathbf{W}^*_{\frac{1}{\eta T}}-\mathbf{W}_0\|_2}{m^c}\Big)
$$

$$
\times \|\mathbf{W}^*_{\frac{1}{\eta T}}-\mathbf{W}_0\|_2^2\Big] + \frac{(\sqrt{\eta T}+\|\mathbf{W}^*_{\frac{1}{\eta T}}-\mathbf{W}_0\|_2)}{m^c}\big(\|\mathbf{W}^*_{\frac{1}{\eta T}}-\mathbf{W}_0\|_2^2+\eta T\Lambda_{\frac{1}{\eta T}}\big)+\Lambda_{\frac{1}{\eta T}}\Big).
$$

Recalling that $\rho = \mathcal{O}(m^{1-2c})$. If $\eta T m^{1-2c} = \mathcal{O}(n)$ and $\eta T(\sqrt{\eta T} + \|\mathbf{W}^* - \mathbf{W}_0\|_2) = \mathcal{O}(m^c)$, there holds $\eta T\rho = \mathcal{O}(n)$ and $\eta T(\sqrt{\eta T} + \|\mathbf{W}^*_{\frac{1}{\eta T}} - \mathbf{W}_0\|_2)/m^c = \mathcal{O}(1)$. Then from the above bound we can get

$$
\mathbb{E}[L(\mathbf{W}_T) - L(\mathbf{W}^*)] = \mathcal{O}\Big(\frac{\eta T\rho}{n}\Big[L(\mathbf{W}^*_{\frac{1}{\eta T}}) + \frac{1}{2\eta T}\|\mathbf{W}^*_{\frac{1}{\eta T}}-\mathbf{W}_0\|_2^2\Big] + \frac{1}{\eta T}\|\mathbf{W}^*_{\frac{1}{\eta T}}-\mathbf{W}_0\|_2^2 + \Lambda_{\frac{1}{\eta T}}\Big).
$$

Combining the above bound with the facts $L(\mathbf{W}^*_{\frac{1}{\eta T}}) + \frac{1}{2\eta T}\|\mathbf{W}^*_{\frac{1}{\eta T}} - \mathbf{W}_0\|_2^2 = L(\mathbf{W}^*) + \Lambda_{\frac{1}{\eta T}}$ and $\|\mathbf{W}^*_{\frac{1}{\eta T}} - \mathbf{W}_0\|_2 \le \sqrt{\eta T\Lambda_{\frac{1}{\eta T}}}$ together we get

$$
\mathbb{E}[L(\mathbf{W}_T) - L(\mathbf{W}^*)] = \mathcal{O}\Big(\frac{\eta T\rho}{n}L(\mathbf{W}^*) + \Lambda_{\frac{1}{\eta T}}\Big).
$$

The proof is completed. $\qquad\square$

*Proof of Corollary 4.* **Part (a). Case 1.** From the definition of the approximation error $\Lambda_{\frac{1}{\eta T}}$, we know that $\Lambda_{\frac{1}{\eta T}} \le \frac{1}{2\eta T}\|\mathbf{W}^* - \mathbf{W}_0\|_2^2$. Combining this with Theorem 3, we have

$$
\mathbb{E}[L(\mathbf{W}_T) - L(\mathbf{W}^*)] = \mathcal{O}\Big(\frac{\eta T\rho}{n}L(\mathbf{W}^*) + \frac{1}{\eta T}\|\mathbf{W}^* - \mathbf{W}_0\|_2^2\Big).
$$

Without loss of generality, we consider $\|\mathbf{W}_0\|_2$ as a constant. To obtain the excess risk rate, we discuss the following two cases: $2c + 6\mu - 3 > 0$ and $2c + 6\mu - 3 \le 0$.

For the case $2c+6\mu-3>0$, to ensure conditions (4), (5) and $\eta T(\sqrt{\eta T}+\|\mathbf{W}^*_{\frac{1}{\eta T}}-\mathbf{W}_0\|_2)=\mathcal{O}(m^c)$ hold, we set $m\asymp(\eta T)^{\frac{3}{2c}}$ for this case. Then according to Theorem 3 and Assumption 3 we know

$$\mathbb{E}[L(\mathbf{W}_T)-L(\mathbf{W}^*)]=\mathcal{O}\Big((\eta T)^{\frac{3-4c}{2c}}n^{-1}+(\eta T)^{\frac{3-6\mu-2c}{2c}}\Big).$$

If $c<3/4$, under the condition $\eta T\lesssim n^{\frac{c}{3-4c}}$ and $n^{\frac{c}{6\mu+2c-3}}\lesssim\eta T$, there holds $(\eta T)^{\frac{3-4c}{2c}}n^{-1}+(\eta T)^{\frac{3-6\mu-2c}{2c}}=\mathcal{O}(1/\sqrt{n})$. To ensure the above-mentioned conditions hold simultaneously, we further require $c+\mu\geq 1$ such that $n^{\frac{c}{3-4c}}\gtrsim n^{\frac{c}{6\mu+2c-3}}$. Therefore, if $c\in[1-\mu,3/4]$ and $c+3\mu>3/2$, we can obtain

$$\mathbb{E}[L(\mathbf{W}_T)-L(\mathbf{W}^*)]=\mathcal{O}\Big(\frac{1}{\sqrt{n}}\Big)$$

with $m\asymp(\eta T)^{\frac{3}{2c}}$ and $n^{\frac{c}{6\mu+2c-3}}\lesssim\eta T\lesssim n^{\frac{c}{3-4c}}$.

If $c\geq 3/4$, for any $\eta T\geq 1$ and $n\geq 1$, there holds $(\eta T)^{\frac{3-4c}{2c}}n^{-1}=\mathcal{O}(1/\sqrt{n})$. Similar to before, if $\eta T\gtrsim n^{\frac{c}{6\mu+2c-3}}$, there holds $(\eta T)^{\frac{3-6\mu-2c}{2c}}=\mathcal{O}(1/\sqrt{n})$. Then we can obtain the excess population bound $\mathcal{O}(1/\sqrt{n})$ with $m\asymp(\eta T)^{\frac{3}{2c}}$ and $\eta T\gtrsim n^{\frac{c}{6\mu+2c-3}}$.

**Case 2.** For the case $2c+6\mu-3\leq 0$, we can choose $m\asymp(\eta T)^{\frac{1}{c+\mu-1/2}}$ to ensure conditions (4), (5) and $\eta T(\sqrt{\eta T}+\|\mathbf{W}^*_{\frac{1}{\eta T}}-\mathbf{W}_0\|_2)=\mathcal{O}(m^c)$ hold. From Theorem 3 we know

$$\mathbb{E}[L(\mathbf{W}_T)-L(\mathbf{W}^*)]=\mathcal{O}\big((\eta T)^{\frac{1-2c+2\mu}{2c+2\mu-1}}n^{-1}+(\eta T)^{\frac{3-6\mu-2c}{2c+2\mu-1}}\big).$$

Note $3-6\mu-2c\geq 0$ and $2c+2\mu-1>0$. Then the term $(\eta T)^{\frac{3-6\mu-2c}{2c+2\mu-1}}$ will not converge for any choice of $\eta T$ in this case. The proof of Part (a) is completed.

**Part (b).** Now, we consider the low noise case, i.e., $L(\mathbf{W}^*)=0$. Combining the fact $\Lambda_{\frac{1}{\eta T}}\leq\frac{1}{2\eta T}\|\mathbf{W}^*-\mathbf{W}_0\|_2^2$, Assumption 3 and Theorem 3 with $L(\mathbf{W}^*)=0$, we can get

$$\mathbb{E}[L(\mathbf{W}_T)-L(\mathbf{W}^*)]=\mathcal{O}\Big(\frac{m^{1-2\mu}}{\eta T}\Big).$$

Similar to part (a), we can set $m\asymp(\eta T)^{\frac{3}{2c}}$, $\eta T\gtrsim n^{\frac{2c}{6\mu+2c-3}}$ and obtain

$$\mathbb{E}[L(\mathbf{W}_T)-L(\mathbf{W}^*)]=\mathcal{O}\Big(\frac{1}{n}\Big).$$

We can check that this choice of $m$ and $\eta T$ satisfies conditions (4) and (5). The proof is completed. $\square$

**Remark A.1.** Several works Charles & Papailiopoulos (2018); Hardt et al. (2016); Lei & Ying (2020b); Zhou et al. (2022); Mou et al. (2018) studied the stability behavior of stochastic gradient methods for non-convex losses, which can be applied to two-layer networks. Specifically, to obtain meaningful stability bounds, Hardt et al. (2016) required a time-dependent step size $\eta_t=1/t$, which is insufficient to get a good convergence rate for optimization error. Charles & Papailiopoulos (2018); Lei & Ying (2020b); Zhou et al. (2022) established generalization bounds by introducing the Polyak-Łojasiewicz condition, which depends on a problem-dependent number. This number might be large in practice and results in a worse generalization bound. It is hard to provide a direct comparison with their results since the learning settings are different.

# B  PROOFS OF THREE-LAYER NEURAL NETWORKS

## B.1  PROOFS OF GENERALIZATION BOUNDS

For a matrix $\mathbf{W}$, let $\mathbf{W}_{s;\cdot}$ and $\mathbf{W}_{is}$ denote the $s$-th row and the $(i,s)$-th entry of $\mathbf{W}$, respectively.

**Lemma B.1** (Smoothness and Curvature). *Suppose Assumptions 1 and 2 hold. For any fixed* $\mathbf{W}=(\mathbf{W}^{(1)},\mathbf{W}^{(2)})\in\mathbb{R}^{m\times d}\times\mathbb{R}^{m\times m}$ *and any* $z\in\mathcal{Z}$, *there holds*

$$\lambda_{\max}\big(\nabla^2\ell(\mathbf{W};z)\big)\leq\rho_{\mathbf{W}}\ \text{with}$$

$$\rho_{\mathbf{W}} = \frac{B_{\sigma'}^4 c_{\mathbf{x}}^2}{m^{4c-1}} \big\| \mathbf{W}^{(2)} \big\|_2^2 + \frac{B_{\sigma'}^2 B_\sigma^2}{m^{4c-2}} + C_{\mathbf{W}} \Big( \frac{B_{\sigma'} B_\sigma}{m^{2c-1}} \big\| \mathbf{W}^{(2)} \big\|_2 + \sqrt{2c_0} \Big),$$

$$\lambda_{\min} \big( \nabla^2 \ell(\mathbf{W}; z) \big) \geq -C_{\mathbf{W}} \Big( \frac{B_{\sigma'} B_\sigma}{m^{2c-1}} \big\| \mathbf{W}^{(2)} \big\|_2 + \sqrt{2c_0} \Big),$$

*where* $C_{\mathbf{W}} = \frac{B_{\sigma'}^2 B_{\sigma''} c_{\mathbf{x}}^2}{m^{3c}} \big\| \mathbf{W}^{(2)} \big\|_2^2 + \Big( \frac{B_{\sigma'} B_{\sigma''} c_{\mathbf{x}}^2}{m^{2c-\frac{1}{2}}} + \frac{2 B_{\sigma''} B_{\sigma'} B_\sigma c_{\mathbf{x}}}{m^{3c-\frac{1}{2}}} \Big) \big\| \mathbf{W}^{(2)} \big\|_2 + \frac{B_{\sigma''} B_\sigma^2}{m^{3c-1}} + \frac{2 B_{\sigma'}^2 c_{\mathbf{x}}}{m^{2c-\frac{1}{2}}}.$

*Proof.* Let $A_{\mathbf{W}^{(1)}} = [\sigma(\mathbf{W}_{1;:}^{(1)} \mathbf{x}), \ldots, \sigma(\mathbf{W}_{m;:}^{(1)} \mathbf{x})]^\top \in \mathbb{R}^m$. Let $\tilde{\mathbf{w}} = \mathbf{w}^\top$. We first estimate the upper bound of $\| \nabla f_{\mathbf{W}}(\mathbf{x}) \|_2$. Note that for any $k = 1, \ldots, m$

$$\nabla_{\tilde{\mathbf{w}}_k^{(1)}} f_{\mathbf{W}}(\mathbf{x}) = \frac{\partial f_{\mathbf{W}}(\mathbf{x})}{\partial \tilde{\mathbf{w}}_k^{(1)}} = \frac{1}{m^{2c}} \sum_{i=1}^m a_i \sigma' \Big( \frac{1}{m^c} \sum_{s=1}^m \mathbf{W}_{is}^{(2)} \sigma(\mathbf{W}_{s;:}^{(1)} \mathbf{x}) \Big) \mathbf{W}_{ik}^{(2)} \sigma'(\mathbf{W}_{k;:}^{(1)} \mathbf{x}) \mathbf{x}$$

and

$$\nabla_{\tilde{\mathbf{w}}_k^{(2)}} f_{\mathbf{W}}(\mathbf{x}) = \frac{\partial f_{\mathbf{W}}(\mathbf{x})}{\partial \tilde{\mathbf{w}}_k^{(2)}} = \frac{a_k}{m^{2c}} \sigma' \Big( \frac{1}{m^c} \sum_{s=1}^m \mathbf{W}_{ks}^{(2)} \sigma(\mathbf{W}_{s;:}^{(1)} \mathbf{x}) \Big) A_{\mathbf{W}^{(1)}}.$$

According to Assumptions 1 and 2, the upper bound of the gradient can be controlled as follows

$$\begin{aligned}
\| \nabla f_{\mathbf{W}}(\mathbf{x}) \|_2^2 &= \sum_{k=1}^m \Big( \big\| \nabla_{\tilde{\mathbf{w}}_k^{(1)}} f_{\mathbf{W}}(\mathbf{x}) \big\|_2^2 + \big\| \nabla_{\tilde{\mathbf{w}}_k^{(2)}} f_{\mathbf{W}}(\mathbf{x}) \big\|_2^2 \Big) \\
&\leq \frac{B_{\sigma'}^4 c_{\mathbf{x}}^2}{m^{4c-1}} \sum_{k=1}^m \sum_{i=1}^m |\mathbf{W}_{ik}^{(2)}|^2 + \frac{B_{\sigma'}^2 B_\sigma^2}{m^{4c-2}} \\
&\leq \frac{B_{\sigma'}^4 c_{\mathbf{x}}^2}{m^{4c-1}} \big\| \mathbf{W}^{(2)} \big\|_2^2 + \frac{B_{\sigma'}^2 B_\sigma^2}{m^{4c-2}}.
\end{aligned} \tag{B.1}$$

For any $k, j \in [m]$, we know

$$\begin{aligned}
\frac{\partial^2 f_{\mathbf{W}}(\mathbf{x})}{\big( \partial \tilde{\mathbf{w}}_k^{(1)} \big)^2} =& \frac{1}{m^{3c}} \sum_{i=1}^m a_i \sigma'' \Big( \frac{1}{m^c} \sum_{s=1}^m \mathbf{w}_{is}^2 \sigma(\mathbf{W}_{s;:}^{(1)} \mathbf{x}) \Big) \big( \mathbf{W}_{ik}^{(2)} \big)^2 \big( \sigma'(\mathbf{W}_{k;:}^{(1)} \mathbf{x}) \big)^2 \mathbf{x} \mathbf{x}^\top \\
&+ \frac{1}{m^{2c}} \sum_{i=1}^m a_i \sigma' \Big( \frac{1}{m^c} \sum_{s=1}^m \mathbf{W}_{is}^2 \sigma(\mathbf{W}_{s;:}^{(1)} \mathbf{x}) \Big) \mathbf{W}_{ik}^{(2)} \sigma''(\mathbf{W}_{k;:}^{(1)} \mathbf{x}) \mathbf{x} \mathbf{x}^\top,
\end{aligned}$$

$$\frac{\partial^2 f_{\mathbf{W}}(\mathbf{x})}{\big( \partial \tilde{\mathbf{w}}_k^{(2)} \big)^2} = \frac{a_k}{m^{3c}} \sigma'' \Big( \frac{1}{m^c} \sum_{s=1}^m \mathbf{W}_{ks}^{(2)} \sigma(\mathbf{W}_{s;:}^{(1)} \mathbf{x}) \Big) A_{\mathbf{W}^{(1)}} A_{\mathbf{W}^{(1)}}^\top$$

and

$$\begin{aligned}
\frac{\partial^2 f_{\mathbf{W}}(\mathbf{x})}{\partial \tilde{\mathbf{w}}_k^{(1)} \partial \tilde{\mathbf{w}}_j^{(2)}} =& \frac{1}{m^{3c}} a_j \sigma'' \Big( \frac{1}{m^c} \sum_{s=1}^m \mathbf{W}_{js}^{(2)} \sigma(\mathbf{W}_{s;:}^{(1)} \mathbf{x}) \Big) \mathbf{W}_{jk}^{(2)} \sigma'(\mathbf{W}_{k;:}^{(1)} \mathbf{x}) \mathbf{x} A_{\mathbf{W}^{(1)}}^\top \\
&+ \frac{1}{m^{2c}} a_j \sigma' \Big( \frac{1}{m^c} \sum_{s=1}^m \mathbf{W}_{js}^{(2)} \sigma(\mathbf{W}_{s;:}^{(1)} \mathbf{x}) \Big) \sigma'(\mathbf{W}_{k;:}^{(1)} \mathbf{x}) \mathbf{x} B_k,
\end{aligned}$$

where $B_k \in \mathbb{R}^{1 \times m}$ with $k$-th element is 1 and others are 0. Let the vector $\mathbf{u} \in \mathbb{R}^{md+m^2}$ have unit norm $\| \mathbf{u} \|_2 = 1$ and be composed in a manner matching the parameter $\mathbf{W} = (\mathbf{W}^{(1)}, \mathbf{W}^{(2)})$ so that $\mathbf{u} = (\mathbf{u}^{(1)}, \mathbf{u}^{(2)})$, where $\mathbf{u}^{(1)} \in \mathbb{R}^{m \times d}$ and $\mathbf{u}^{(2)} \in \mathbb{R}^{m \times m}$ have been vectorised in a row-major manner with $\mathbf{u}_k^{(1)} \in \mathbb{R}^{1 \times d}$ and $\mathbf{u}_k^{(2)} \in \mathbb{R}^{1 \times m}$. Then

$$\mathbf{u}^\top \nabla^2 f_{\mathbf{W}}(\mathbf{x}) \mathbf{u}$$
$$= \sum_{k=1}^m \Big( \mathbf{u}_k^{(1)} \frac{\partial^2 f_{\mathbf{W}}(\mathbf{x})}{\big( \partial \tilde{\mathbf{w}}_k^{(1)} \big)^2} \big( \mathbf{u}_k^{(1)} \big)^\top + \mathbf{u}_k^{(2)} \frac{\partial^2 f_{\mathbf{W}}(\mathbf{x})}{\big( \partial \tilde{\mathbf{w}}_k^{(2)} \big)^2} \big( \mathbf{u}_k^{(2)} \big)^\top + 2 \sum_{j=1}^m \mathbf{u}_k^{(1)} \frac{\partial^2 f_{\mathbf{W}}(\mathbf{x})}{\partial \tilde{\mathbf{w}}_k^{(1)} \partial \tilde{\mathbf{w}}_j^{(2)}} \big( \mathbf{u}_j^{(2)} \big)^\top \Big). \tag{B.2}$$

We estimate the above three terms separately. Let $\mathbf{W}^{(2)}_{\cdot k}$ denote the $k$-th column of $\mathbf{W}^{(2)}$.

$$\sum_{k=1}^{m} \mathbf{u}_k^{(1)} \frac{\partial^2 f_{\mathbf{W}}(\mathbf{x})}{\left(\partial \tilde{\mathbf{w}}_k^{(1)}\right)^2} \left(\mathbf{u}_k^{(1)}\right)^\top$$

$$\leq \frac{B_{\sigma''} B_{\sigma'}^2}{m^{3c}} \sum_{k=1}^{m} \mathbf{u}_k^{(1)} \sum_{i=1}^{m} \left(\mathbf{W}_{ik}^{(2)}\right)^2 \mathbf{x}\mathbf{x}^\top \left(\mathbf{u}_k^{(1)}\right)^\top + \frac{B_{\sigma'} B_{\sigma''}}{m^{2c}} \sum_{k=1}^{m} \mathbf{u}_k^{(1)} \sum_{i=1}^{m} \mathbf{W}_{ik}^{(2)} \mathbf{x}\mathbf{x}^\top \left(\mathbf{u}_k^{(1)}\right)^\top$$

$$\leq \frac{B_{\sigma''} B_{\sigma'}^2}{m^{3c}} \sum_{i=1}^{m} \sum_{k=1}^{m} \left(\mathbf{W}_{ik}^{(2)}\right)^2 \left(\mathbf{u}_k^{(1)}\mathbf{x}\right)^2 + \frac{B_{\sigma'} B_{\sigma''}}{m^{2c}} \sqrt{\sum_{k=1}^{m} \left(\sum_{i=1}^{m} \mathbf{W}_{ik}^{(2)}\right)^2} \sqrt{\sum_{k=1}^{m} \left(\mathbf{u}_k^{(1)}\mathbf{x}\mathbf{x}^\top \left(\mathbf{u}_k^{(1)}\right)^\top\right)^2}$$

$$\leq \frac{B_{\sigma''} B_{\sigma'}^2 c_{\mathbf{x}}^2}{m^{3c}} \sum_{k=1}^{m} \sum_{i=1}^{m} \left(\mathbf{W}_{ik}^{(2)}\right)^2 \|\mathbf{u}_k^{(1)}\|_2^2 + \frac{B_{\sigma'} B_{\sigma''}}{m^{2c}} \sqrt{m \sum_{k=1}^{m} \sum_{i=1}^{m} \left(\mathbf{W}_{ik}^{(2)}\right)^2 \sum_{k=1}^{m} \left(\mathbf{u}_k^{(1)}\mathbf{x}\mathbf{x}^\top \left(\mathbf{u}_k^{(1)}\right)^\top\right)}$$

$$\leq \frac{B_{\sigma''} B_{\sigma'}^2 c_{\mathbf{x}}^2}{m^{3c}} \|\mathbf{W}^{(2)}\|_2^2 + \frac{B_{\sigma'} B_{\sigma''} c_{\mathbf{x}}^2}{m^{2c-\frac{1}{2}}} \|\mathbf{W}^{(2)}\|_2, \tag{B.3}$$

where in the third inequality we used $\left(\sum_{i=1}^{m} \mathbf{W}_{ik}^{(2)}\right)^2 \leq m \sum_{i=1}^{m} \left(\mathbf{W}_{ik}^{(2)}\right)^2$ and $\sum_{k=1}^{m} \left(\mathbf{u}_k^{(1)}\mathbf{x}\mathbf{x}^\top \left(\mathbf{u}_k^{(1)}\right)^\top\right)^2 \leq \left(\sum_{k=1}^{m} \mathbf{u}_k^{(1)}\mathbf{x}\mathbf{x}^\top \left(\mathbf{u}_k^{(1)}\right)^\top\right)^2$, and the last inequality follows from $\mathbf{u}_k^{(1)}\mathbf{x}\mathbf{x}^\top \left(\mathbf{u}_k^{(1)}\right)^\top = \left(\mathbf{u}_k^{(1)}\mathbf{x}\right)^2 \leq c_{\mathbf{x}}^2 \|\mathbf{u}_k^{(1)}\|_2^2$ and $\|\mathbf{u}_k^{(1)}\|_2^2 \leq 1$.

For the second term in (B.2), we control it by

$$\sum_{k=1}^{m} \mathbf{u}_k^{(2)} \frac{\partial^2 f_{\mathbf{W}}(\mathbf{x})}{\left(\partial \tilde{\mathbf{w}}_k^{(2)}\right)^2} \left(\mathbf{u}_k^{(2)}\right)^\top \leq \frac{B_{\sigma''}}{m^{3c}} \sum_{k=1}^{m} \left(\mathbf{u}_k^{(2)} A_{\mathbf{W}^{(1)}}\right)^2 \leq \frac{B_{\sigma''} B_{\sigma}}{m^{3c-1}} \|\mathbf{u}^{(2)}\|_2^2 \leq \frac{B_{\sigma''} B_{\sigma}}{m^{3c-1}}. \tag{B.4}$$

Further, according to Cauchy-Schwarz inequality, we can get

$$\sum_{k=1}^{m} \sum_{j=1}^{m} \mathbf{u}_k^{(1)} \frac{\partial^2 f_{\mathbf{W}}(\mathbf{x})}{\partial \tilde{\mathbf{w}}_k^{(1)} \partial \tilde{\mathbf{w}}_j^{(2)}} \left(\mathbf{u}_j^{(2)}\right)^\top$$

$$\leq \frac{B_{\sigma''} B_{\sigma'} B_{\sigma}}{m^{3c}} \sum_{j=1}^{m} \mathbf{W}_{j;\cdot}^{(2)} \mathbf{u}^{(1)}\mathbf{x} \|\mathbf{u}_j^{(2)}\|_1 + \frac{B_{\sigma'}^2}{m^{2c}} \sum_{k=1}^{m} \mathbf{u}_k^{(1)}\mathbf{x}\left(\sum_{j=1}^{m} \mathbf{u}_{jk}^{(2)}\right)$$

$$\leq \frac{B_{\sigma''} B_{\sigma'} B_{\sigma}}{m^{3c}} \sqrt{\sum_{j=1}^{m} \|\mathbf{u}_\ell^{(2)}\|_1^2} \sqrt{\sum_{j=1}^{m} \left(\mathbf{W}_{j;\cdot}^{(2)} \mathbf{u}^{(1)}\mathbf{x}\right)^2} + \frac{B_{\sigma'}^2}{m^{2c}} \sqrt{\sum_{k=1}^{m} \left(\mathbf{u}_k^{(1)}\mathbf{x}\right)^2} \sqrt{\sum_{k=1}^{m} \|\mathbf{u}_{\cdot k}^{(2)}\|_1^2}$$

$$\leq \frac{B_{\sigma''} B_{\sigma'} B_{\sigma}}{m^{3c}} \sqrt{m \sum_{j=1}^{m} \|\mathbf{u}_j^{(2)}\|_2^2} \sqrt{\sum_{j=1}^{m} \|\mathbf{W}_{j;\cdot}^{(2)}\|_2^2 \|\mathbf{u}^{(1)}\mathbf{x}\|_2^2} + \frac{B_{\sigma'}^2 c_{\mathbf{x}}}{m^{2c}} \sqrt{\sum_{k=1}^{m} \|\mathbf{u}_k^{(1)}\|_2^2} \sqrt{m \sum_{k=1}^{m} \|\mathbf{u}_{\cdot k}^{(2)}\|_2^2}$$

$$\leq \frac{B_{\sigma''} B_{\sigma'} B_{\sigma} c_{\mathbf{x}}}{m^{3c-\frac{1}{2}}} \|\mathbf{u}^{(1)}\|_2 \|\mathbf{u}^{(2)}\|_2 \|\mathbf{W}^{(2)}\|_2 + \frac{B_{\sigma'}^2 c_{\mathbf{x}}}{m^{2c-\frac{1}{2}}} \|\mathbf{u}^{(1)}\|_2 \|\mathbf{u}^{(2)}\|_2$$

$$\leq \frac{B_{\sigma''} B_{\sigma'} B_{\sigma} c_{\mathbf{x}}}{m^{3c-\frac{1}{2}}} \|\mathbf{W}^{(2)}\|_2 + \frac{B_{\sigma'}^2 c_{\mathbf{x}}}{m^{2c-\frac{1}{2}}}, \tag{B.5}$$

where in the first equality we used $\sum_{k=1}^{m} \mathbf{u}_k^{(1)} \mathbf{W}_{jk}^{(2)}\mathbf{x} = \mathbf{W}_{j;\cdot}^{(2)} \mathbf{u}^{(1)}\mathbf{x}$, the second inequality follows from $\left(\sum_{j=1}^{m} \mathbf{u}_{jk}^{(2)}\right) \leq \|\mathbf{u}_{\cdot k}^{(2)}\|_1$. Here $\mathbf{u}_{\cdot k}^{(2)}$ denotes the $k$-th column of $\mathbf{u}^{(2)}$.

Plugging (B.3), (B.4) and (B.5) back into (B.2) we can get

$$\left\|\nabla^2 f_{\mathbf{W}}(\mathbf{x})\right\|_{op}$$

$$\leq \frac{B_{\sigma'}^2 B_{\sigma''} c_{\mathbf{x}}^2}{m^{3c}} \|\mathbf{W}^{(2)}\|_2^2 + \left(\frac{B_{\sigma'} B_{\sigma''} c_{\mathbf{x}}^2}{m^{2c-\frac{1}{2}}} + \frac{2 B_{\sigma''} B_{\sigma'} B_{\sigma} c_{\mathbf{x}}}{m^{3c-\frac{1}{2}}}\right) \|\mathbf{W}^{(2)}\|_2 + \frac{B_{\sigma''} B_{\sigma}^2}{m^{3c-1}} + \frac{2 B_{\sigma'}^2 c_{\mathbf{x}}}{m^{2c-\frac{1}{2}}} := C_{\mathbf{W}}. \tag{B.6}$$

For any $\mathbf{W}, \widetilde{\mathbf{W}}$, according to Assumptions 1 ans 2 we can get

$$
\begin{aligned}
& \big| f_{\mathbf{W}}(\mathbf{x}) - f_{\widetilde{\mathbf{W}}}(\mathbf{x}) \big| \\
& \leq \frac{1}{m^c} \sum_{k=1}^m \Big| \sigma\Big(\frac{1}{m^c} \sum_{s=1}^m \mathbf{W}_{ks}^{(2)} \sigma(\mathbf{W}_s^{(1)}\mathbf{x})\Big) - \sigma\Big(\frac{1}{m^c} \sum_{s=1}^m \widetilde{\mathbf{W}}_{ks}^{(2)} \sigma(\widetilde{\mathbf{W}}_{s;\cdot}^{(1)}\mathbf{x})\Big)\Big| \\
& \leq \frac{B_{\sigma'}}{m^{2c}} \sum_{k=1}^m \sum_{s=1}^m \big| \mathbf{W}_{ks}^{(2)} \sigma(\mathbf{W}_{s;\cdot}^{(1)}\mathbf{x}) - \widetilde{\mathbf{W}}_{ks}^{(2)} \sigma(\mathbf{W}_{s;\cdot}^{(1)}\mathbf{x}) + \widetilde{\mathbf{W}}_{ks}^{(2)} \sigma(\mathbf{W}_{s;\cdot}^{(1)}\mathbf{x}) - \widetilde{\mathbf{W}}_{ks}^{(2)} \sigma(\widetilde{\mathbf{W}}_{s;\cdot}^{(1)}\mathbf{x}) \big| \\
& \leq \frac{B_{\sigma'} B_\sigma}{m^{2c}} \sum_{k=1}^m \sum_{s=1}^m \big| \mathbf{W}_{ks}^{(2)} - \widetilde{\mathbf{W}}_{ks}^{(2)} \big| + \frac{B_{\sigma'}^2}{m^{2c}} \sum_{k=1}^m \sum_{s=1}^m \big| \widetilde{\mathbf{W}}_{ks}^{(2)} \big| \cdot \big| (\mathbf{W}_{s;\cdot}^{(1)} - \widetilde{\mathbf{W}}_{s;\cdot}^{(1)})\mathbf{x} \big| \\
& \leq \frac{B_{\sigma'} B_\sigma}{m^{2c-1}} \big\| \mathbf{W}^{(2)} - \widetilde{\mathbf{W}}^{(2)} \big\|_2 + \frac{B_{\sigma'}^2 c_{\mathbf{x}} \| \widetilde{\mathbf{W}}^{(2)} \|_\infty}{m^{2c-\frac{3}{2}}} \big\| \mathbf{W}^{(1)} - \widetilde{\mathbf{W}}^{(1)} \big\|_2.
\end{aligned} \tag{B.7}
$$

Since

$$
\nabla^2 \ell(\mathbf{W}; z) = \nabla f_{\mathbf{W}}(\mathbf{x}) \nabla f_{\mathbf{W}}(\mathbf{x})^\top + \nabla^2 f_{\mathbf{W}}(\mathbf{x})\big(f_{\mathbf{W}}(\mathbf{x}) - y\big). \tag{B.8}
$$

Then for any $\mathbf{W} \in \mathbb{R}^{md+m^2}$, we can upper bound the maximal eigenvalue of the Hessian by combining (B.1), (B.6) and (B.7) with $\widetilde{\mathbf{W}} = \mathbf{0}$ together

$$
\begin{aligned}
\lambda_{\max}(\nabla^2 \ell(\mathbf{W}; z)) & \leq \| \nabla f_{\mathbf{W}}(\mathbf{x}) \|_2^2 + \| \nabla^2 f_{\mathbf{W}}(\mathbf{x}) \|_{op} | f_{\mathbf{W}}(\mathbf{x}) - y | \\
& \leq \frac{B_{\sigma'}^4 c_{\mathbf{x}}^2}{m^{4c-1}} \| \mathbf{W}^{(2)} \|_2^2 + \frac{B_{\sigma'}^2 B_\sigma^2}{m^{4c-2}} + C_{\mathbf{W}}\big( | f_{\mathbf{W}}(\mathbf{x}) - f_{\mathbf{0}}(\mathbf{x}) | + | f_{\mathbf{0}}(\mathbf{x}) - y | \big) \\
& \leq \frac{B_{\sigma'}^4 c_{\mathbf{x}}^2}{m^{4c-1}} \| \mathbf{W}^{(2)} \|_2^2 + \frac{B_{\sigma'}^2 B_\sigma^2}{m^{4c-2}} + C_{\mathbf{W}}\Big( \frac{B_{\sigma'} B_\sigma}{m^{2c-1}} \| \mathbf{W}^{(2)} \|_2 + \sqrt{2\ell(\mathbf{0}; z)} \Big) \\
& \leq \frac{B_{\sigma'}^4 c_{\mathbf{x}}^2}{m^{4c-1}} \| \mathbf{W}^{(2)} \|_2^2 + \frac{B_{\sigma'}^2 B_\sigma^2}{m^{4c-2}} + C_{\mathbf{W}}\Big( \frac{B_{\sigma'} B_\sigma}{m^{2c-1}} \| \mathbf{W}^{(2)} \|_2 + \sqrt{2c_0} \Big).
\end{aligned}
$$

Note that $\nabla f_{\mathbf{W}}(\mathbf{x}) \nabla f_{\mathbf{W}}(\mathbf{x})^\top$ is positive semi-definite, then from (B.6), (B.8) and (B.7) with $\widetilde{\mathbf{W}} = \mathbf{0}$ we can get

$$
\begin{aligned}
\lambda_{\min}(\nabla^2 \ell(\mathbf{W}; z)) & \geq -\| \nabla^2 f_{\mathbf{W}}(\mathbf{x}) \|_{op} | f_{\mathbf{W}}(\mathbf{x}) - y | \\
& \geq -C_{\mathbf{W}}\big( | f_{\mathbf{W}}(\mathbf{x}) - f_{\mathbf{0}}(\mathbf{x}) | + | f_{\mathbf{0}}(\mathbf{x}) - y | \big) \\
& \geq -C_{\mathbf{W}}\Big( \frac{B_{\sigma'} B_\sigma}{m^{2c-1}} \| \mathbf{W}^{(2)} \|_2 + \sqrt{2\ell(\mathbf{W}_0; z)} \Big) \\
& \geq -C_{\mathbf{W}}\Big( \frac{B_{\sigma'} B_\sigma}{m^{2c-1}} \| \mathbf{W}^{(2)} \|_2 + \sqrt{2c_0} \Big).
\end{aligned}
$$

The proof is completed. $\qquad \square$

Let $B_1 = \max\big\{ B_{\sigma'}^2 B_{\sigma''} c_{\mathbf{x}}^2, B_{\sigma'} B_{\sigma''} c_{\mathbf{x}}^2, 2B_{\sigma''} B_\sigma c_{\mathbf{x}}, B_{\sigma''} B_\sigma^2, 2B_{\sigma''}^2 c_{\mathbf{x}} \big\}$ and $B_2 = \max\big\{ B_{\sigma'}^4 c_{\mathbf{x}}^2, B_{\sigma'}^2 B_{\sigma''}^2, B_{\sigma''} B_\sigma, \sqrt{2c_0} \big\}$.

**Lemma B.2.** *Suppose Assumptions 1 and 2 hold. Let $\{\mathbf{W}_t\}$ and $\{\mathbf{W}_t'\}$ be produced by GD iterates with $T$ iterations based on $S$ and $S'$, respectively. Let $C > 0$ be a constant. Assume $\eta \leq 1/(2\hat{\rho})$ and (6) hold. Then for any $c \in (1/2, 1]$ and any $t = 0, \ldots, T$ there holds*

$$
\| \mathbf{W}_t - \mathbf{W}_0 \|_2 \leq \sqrt{2c_0 \eta t}
$$

*and*

$$
\| \nabla \ell(\mathbf{W}_t; z) - \nabla \ell(\mathbf{W}_t'; z) \|_2 \leq \hat{\rho} \| \mathbf{W}_t - \mathbf{W}_t' \|_2.
$$

*Proof.* We will prove by induction to show $\| \mathbf{W}_t - \mathbf{W}_0 \|_2 \leq \eta t m^{2c-1}$. Further, we can show that $\rho_{\mathbf{W}} = \mathcal{O}(1)$ for any $\mathbf{W}$ produced by GD iterates if $m$ satisfies (6). Then by assuming $\eta \leq 1/(2\hat{\rho})$ we can prove that $\| \mathbf{W}_t - \mathbf{W}_0 \|_2 \leq \sqrt{2c_0 \eta t}$.

It's obvious that $\|\mathbf{W}_t - \mathbf{W}_0\|_2 \le \eta t m^{2c-1}$ with $t = 0$ holds trivially. Assume $\|\mathbf{W}_t - \mathbf{W}_0\|_2 \le \eta t m^{2c-1}$. According to the update rule (2) we know

$$
\begin{aligned}
&\|\mathbf{W}_{t+1} - \mathbf{W}_0\|_2 \\
&\le \|\mathbf{W}_t - \mathbf{W}_0\|_2 + \eta \|\nabla L_S(\mathbf{W}_t)\|_2 \\
&\le \|\mathbf{W}_t - \mathbf{W}_0\|_2 + \eta \max_{i\in[n]} \|\nabla f_{\mathbf{W}_t}(\mathbf{x}_i)\|_2 \big| f_{\mathbf{W}_t}(\mathbf{x}_i) - y_i \big| \\
&\le \|\mathbf{W}_t - \mathbf{W}_0\|_2 + \eta \Big( \frac{B_{\sigma'}^2 c_{\mathbf{x}}}{m^{2c-\frac{1}{2}}} (\|\mathbf{W}_t - \mathbf{W}_0\|_2 + \|\mathbf{W}_0\|_2) + \frac{B_{\sigma'} B_\sigma}{m^{2c-1}} \Big) \Big( \big| f_{\mathbf{W}_t}(\mathbf{x}_i) - f_0(\mathbf{W}_t) \big| \\
&\quad + \big| f_0(\mathbf{W}_t) - y_i \big| \Big) \\
&\le \|\mathbf{W}_t - \mathbf{W}_0\|_2 + \eta \Big( \frac{B_{\sigma'}^2 c_{\mathbf{x}}}{m^{2c-\frac{1}{2}}} (\|\mathbf{W}_t - \mathbf{W}_0\|_2 + \|\mathbf{W}_0\|_2) + \frac{B_{\sigma'} B_\sigma}{m^{2c-1}} \Big) \Big( \frac{B_{\sigma'} B_\sigma}{m^{2c-1}} (\|\mathbf{W}_t - \mathbf{W}_0\|_2 \\
&\quad + \|\mathbf{W}_0\|_2) + \sqrt{2c_0} \Big),
\end{aligned}
\tag{B.9}
$$

where in the third inequality we used (B.1), the last inequality used (B.7) with $\widetilde{\mathbf{W}} = \mathbf{0}$. If $m$ is large enough such that

$$
\begin{aligned}
&\Big( \frac{B_{\sigma'}^2 c_{\mathbf{x}}}{m^{2c-\frac{1}{2}}} (\|\mathbf{W}_t - \mathbf{W}_0\|_2 + \|\mathbf{W}_0\|_2) + \frac{B_{\sigma'} B_\sigma}{m^{2c-1}} \Big) \Big( \frac{B_{\sigma'} B_\sigma}{m^{2c-1}} (\|\mathbf{W}_t - \mathbf{W}_0\|_2 + \|\mathbf{W}_0\|_2) + \sqrt{2c_0} \Big) \\
&\le m^{2c-1},
\end{aligned}
\tag{B.10}
$$

then from (B.9) we know that $\|\mathbf{W}_{t+1} - \mathbf{W}_0\|_2 \le \eta(t+1) m^{2c-1}$. The first part of the lemma can be proved. Now, we discuss the conditions on $m$ such that (B.10) holds. Let $x = \|\mathbf{W}_t - \mathbf{W}_0\|_2 + \|\mathbf{W}_0\|_2$. To guarantee (B.10), it suffices that the following three inequalities hold

$$
\frac{B_{\sigma'}^3 B_\sigma c_{\mathbf{x}}}{m^{4c-\frac{3}{2}}} x^2 \le \frac{m^{2c-1}}{3}, \quad \Big( \frac{B_{\sigma'}^2 B_\sigma^2}{m^{4c-2}} + \frac{\sqrt{2c_0} B_{\sigma'}^2 c_{\mathbf{x}}}{m^{2c-\frac{1}{2}}} \Big) x \le \frac{m^{2c-1}}{3}, \quad \frac{\sqrt{2c_0} B_{\sigma'} B_\sigma}{m^{2c-1}} \le \frac{m^{2c-1}}{3}. \tag{B.11}
$$

It's easy to verify that (B.11) holds if $m \gtrsim \max\{(\eta T)^{\frac{4}{4c-1}}, (\eta T)^{\frac{1}{4c-2}}, \|\mathbf{W}_0\|_2^{\frac{1}{3c-\frac{5}{4}}}, \|\mathbf{W}_0\|_2^{\frac{1}{6c-3}}\}$, which can be ensured by (6). Hence, if $c \in (1/2, 1]$ and (6) holds, we have $\|\mathbf{W}_t - \mathbf{W}_0\|_2 \le \eta t m^{2c-1}$ for all $t = 0, 1, \ldots, T$.

Recall that

$$
B_1 = \max \big\{ B_{\sigma'}^2 B_{\sigma''} c_{\mathbf{x}}^2, B_{\sigma'} B_{\sigma''} c_{\mathbf{x}}^2, 2B_{\sigma''} B_\sigma c_{\mathbf{x}}, B_{s\sigma''} B_\sigma^2, 2B_{s\sigma''}^2 c_{\mathbf{x}} \big\}
$$

and

$$
B_2 = \max \big\{ B_{\sigma'}^4 c_{\mathbf{x}}^2, B_{\sigma'}^2 B_{\sigma''}^2, B_{s\sigma''} B_\sigma, \sqrt{2c_0} \big\}.
$$

Then from Lemma B.1 we know

$$
C_{\mathbf{W}} \le B_1 \Big( m^{-3c} \big\| \mathbf{W}^{(2)} \big\|_2^2 + \big( m^{\frac{1}{2}-2c} + m^{\frac{1}{2}-3c} \big) \|\mathbf{W}^{(2)}\|_2 + m^{1-3c} + m^{\frac{1}{2}-2c} \Big)
$$

and

$$
\rho_{\mathbf{W}} \le B_2 \Big( m^{1-4c} \big\| \mathbf{W} \big\|_2^2 + m^{2-4c} + C_{\mathbf{W}} \big( m^{1-2c} \big\| \mathbf{W} \big\|_2 + 1 \big) \Big).
$$

Note that (6) implies $m \gtrsim (\eta T)^4 + \|\mathbf{W}_0\|_2^{\frac{4}{8c-3}}$. By using $\|\mathbf{W}_t\|_2 \le \eta t m^{2c-1} + \|\mathbf{W}_0\|_2$ we can verify that

$$
\rho_{\mathbf{W}_t} \le 4B_2(1 + 2B_1) := \hat{\rho} \quad \text{for any} \quad t = 0, \ldots, T.
$$

Hence, we know that $\ell$ is $\hat{\rho}$-smooth when the parameter space is the trajectory of GD. Then for any $t = 0, \ldots, T$ and any $\mathbf{W}_t, \mathbf{W}_t'$ produced by GD iterates, there holds

$$
\|\nabla \ell(\mathbf{W}_t; z) - \nabla \ell(\mathbf{W}_t'; z)\|_2 \le \hat{\rho} \|\mathbf{W}_t - \mathbf{W}_t'\|_2. \tag{B.12}
$$

In addition, by the smoothness of $\ell$ we can get for any $j = 0, \ldots, T-1$

$$
L_S(\mathbf{W}_{j+1}) \le L_S(\mathbf{W}_j) - \eta \Big( 1 - \frac{\eta \hat{\rho}}{2} \Big) \|\nabla L_S(\mathbf{W}_j)\|_2^2.
$$

Rearranging and summing over $j$ yields

$$
\eta \Big( 1 - \frac{\eta \hat{\rho}}{2} \Big) \sum_{j=0}^t \|\nabla L_S(\mathbf{W}_j)\|_2^2 \le \sum_{j=0}^t L_S(\mathbf{W}_j) - L_S(\mathbf{W}_{j+1}) \le L_S(\mathbf{W}_0).
$$

Note that the update rule of GD (2) implies

$$\mathbf{W}_{t+1} = \mathbf{W}_0 - \eta \sum_{j=0}^{t} \nabla L_S(\mathbf{W}_j).$$

Combining the above two equations together and noting that $\eta\hat{\rho} \leq 1/2$, we obtain

$$\|\mathbf{W}_{t+1} - \mathbf{W}_0\|_2^2 = \eta^2 \big\| \sum_{j=0}^{t} \nabla L_S(\mathbf{W}_j) \big\|_2^2 \leq \eta^2 t \sum_{j=0}^{t} \|\nabla L_S(\mathbf{W}_j)\|_2^2 \leq 2\eta t L_S(\mathbf{W}_0) \leq 2c_0 \eta t.$$

The proof is completed. $\qquad\square$

The almost co-coercivity of the gradient operator for three-layer neural networks is given as follows. Recall that $S^{(i)} = \{z_1, \ldots, z_{i-1}, z_i', z_{i+1}, z_n\}$ is the set formed from $S$ by replacing the $i$-th element with $z_i'$. For any $\mathbf{W}$, we have

$$L_{S\backslash i}(\mathbf{W}) = L_S(\mathbf{W}) - \frac{1}{n}\ell(\mathbf{W}; z_i) = L_{S^{(i)}}(\mathbf{W}) - \frac{1}{n}\ell(\mathbf{W}; z_i').$$

Let $\{\mathbf{W}_t\}$ and $\{\mathbf{W}_t^{(i)}\}$ be the sequence produced by GD based on $S$ and $S^{(i)}$, respectively. Let $C_{3,T} = 4B_1\big(2c_0\eta T m^{-3c} + \big(m^{\frac{1}{2}-2c} + m^{\frac{1}{2}-3c}\big)\sqrt{2c_0\eta T} + m^{1-3c} + m^{\frac{1}{2}-2c}\big)$.

**Lemma B.3.** *Suppose $\eta \leq 1/(8\hat{\rho})$ where $\hat{\rho} = 4B_2(1 + 2B_1)$. Assume (6) holds. Then*

$$\langle \mathbf{W}_t - \mathbf{W}_t^{(i)}, \nabla L_{S\backslash i}(\mathbf{W}_t) - \nabla L_{S\backslash i}(\mathbf{W}_t^{(i)}) \rangle \geq 2\eta\big(1 - 4\eta\hat{\rho}\big)\big\|\nabla L_{S\backslash i}(\mathbf{W}_t) - \nabla L_{S\backslash i}(\mathbf{W}_t^{(i)})\big\|_2^2$$

$$- \tilde{\epsilon}_t \big\|\mathbf{W}_t - \mathbf{W}_t^{(i)} - \eta\big(\nabla L_{S\backslash i}(\mathbf{W}_t) - \nabla L_{S\backslash i}(\mathbf{W}_t^{(i)})\big)\big\|_2^2,$$

*where $\tilde{\epsilon}_t = C_{3,T}\big(B_3\big(\frac{\sqrt{\eta T} + \|\mathbf{W}_0\|_2}{m^{2c-\frac{1}{2}}} + m^{1-2c}\big)\big((1+\eta\hat{\rho})\|\mathbf{W}_t - \mathbf{W}_t^{(i)}\|_2 + 2\big(\sqrt{2c_0\eta T} + \|\mathbf{W}_0\|_2\big) + \sqrt{2c_0}\big).*

*Proof.* For any $\mathbf{W} \in \mathcal{B}\big(0, 2(\sqrt{2c_0\eta T} + \|\mathbf{W}_0\|_2)\big)$, defining the following two functions

$$G_1(\mathbf{W}) = L_{S\backslash i}(\mathbf{W}) - \langle \nabla L_{S\backslash i}(\mathbf{W}_t^{(i)}), \mathbf{W}\rangle, \qquad G_2(\mathbf{W}) = L_{S\backslash i}(\mathbf{W}) - \langle \nabla L_{S\backslash i}(\mathbf{W}_t), \mathbf{W}\rangle.$$

Note that

$$\langle \mathbf{W}_t - \mathbf{W}_t^{(i)}, \nabla L_{S\backslash i}(\mathbf{W}_t) - \nabla L_{S\backslash i}(\mathbf{W}_t^{(i)})\rangle = \big(G_1\big(\mathbf{W}_t\big) - G_1\big(\mathbf{W}_t^{(i)}\big)\big) + \big(G_2\big(\mathbf{W}_t^{(i)}\big) - G_2\big(\mathbf{W}_t\big)\big). \tag{B.13}$$

Hence, it is enough to lower bound $G_1\big(\mathbf{W}_t\big) - G_1\big(\mathbf{W}_t^{(i)}\big)$ and $G_2\big(\mathbf{W}_t^{(i)}\big) - G_2\big(\mathbf{W}_t\big)$.

Note that for any $\mathbf{W}_t$, Lemma B.1 implies that $\|\mathbf{W}_t\|_2 \leq \sqrt{2c_0\eta t} + \|\mathbf{W}_0\|_2$. Then there holds

$$\big\|\mathbf{W}_t - \eta\nabla G_1(\mathbf{W}_t)\big\|_2 \leq \big\|\mathbf{W}_t\big\|_2 + \eta\big\|\nabla L_{S\backslash i}(\mathbf{W}_t) - \nabla L_{S\backslash i}(\mathbf{W}_t^{(i)})\big\|_2$$

$$\leq \big\|\mathbf{W}_t\big\|_2 + \eta\hat{\rho}\big\|\mathbf{W}_t - \mathbf{W}_t^{(i)}\big\|_2$$

$$\leq \sqrt{2c_0\eta t} + \|\mathbf{W}_0\|_2 + 2\eta\hat{\rho}\big(\sqrt{2c_0\eta t} + \|\mathbf{W}_0\|_2\big)$$

$$\leq 2\big(\sqrt{2c_0\eta t} + \|\mathbf{W}_0\|_2\big),$$

where in the last inequality we used $\eta\hat{\rho} \leq 1/8$. Similarly, we can show that $\big\|\mathbf{W}_t^{(i)} - \eta\nabla G_2(\mathbf{W}_t^{(i)})\big\|_2 \leq 2(\sqrt{2c_0\eta T} + \|\mathbf{W}_0\|_2)$. Hence, we know $\mathbf{W}_t - \eta\nabla G_1(\mathbf{W}_t) \in \mathcal{B}\big(0, 2(\sqrt{2c_0\eta T} + \|\mathbf{W}_0\|_2)\big)$ and $\mathbf{W}_t^{(i)} - \eta\nabla G_2(\mathbf{W}_t^{(i)}) \in \mathcal{B}\big(0, 2(\sqrt{2c_0\eta T} + \|\mathbf{W}_0\|_2)\big)$. On the other hand, similar to Lemma B.2, we can show that $G_1(\mathbf{W})$ and $G_2(\mathbf{W})$ is $8\hat{\rho}$-smooth for any $\mathbf{W} \in \mathcal{B}\big(0, 2(\sqrt{2c_0\eta T} + \|\mathbf{W}_0\|_2)\big)$. Combining the above results, we can get

$$G_1(\mathbf{W}_t - \eta\nabla G_1(\mathbf{W}_t)) \leq G_1(\mathbf{W}_t) - \eta\big(1 - 4\eta\hat{\rho}\big)\|\nabla G_1(\mathbf{W}_t)\|_2^2, \tag{B.14}$$

$$G_2(\mathbf{W}_t^{(i)} - \eta\nabla G_2(\mathbf{W}_t^{(i)})) \leq G_2(\mathbf{W}_t^{(i)}) - \eta\big(1 - 4\eta\hat{\rho}\big)\|\nabla G_2(\mathbf{W}_t^{(i)})\|_2^2. \tag{B.15}$$

If we can further show that

$$G_1(\mathbf{W}_t - \eta\nabla G_1(\mathbf{W}_t)) \geq G_1(\mathbf{W}_t^{(i)}) - \frac{\tilde{\epsilon}_t}{2}\|\mathbf{W}_t - \mathbf{W}_t^{(i)} - \eta\nabla G_1(\mathbf{W}_t)\|_2^2, \tag{B.16}$$

$$G_2(\mathbf{W}_t^{(i)} - \eta\nabla G_2(\mathbf{W}_t^{(i)})) \geq G_2(\mathbf{W}_t) - \frac{\tilde{\epsilon}_t}{2}\|\mathbf{W}_t^{(i)} - \mathbf{W}_t - \eta\nabla G_2(\mathbf{W}_t)\|_2^2, \tag{B.17}$$

with $\tilde{\epsilon}_t = C_{3,T}\big(B_3\big(\frac{\sqrt{\eta T}+\|\mathbf{W}_0\|_2}{m^{2c-\frac{1}{2}}}+m^{1-2c}\big)\big((1+\eta\hat{\rho})\|\mathbf{W}_t-\mathbf{W}_t^{(i)}\|_2+2\big(\sqrt{2c_0\eta T}+\|\mathbf{W}_0\|_2\big)+\sqrt{2c_0}\big)$, where $C_{3,T} = 4B_1\big(2c_0\eta T m^{-3c}+\big(m^{\frac{1}{2}-2c}+m^{\frac{1}{2}-3c}\big)\sqrt{2c_0\eta T}+m^{1-3c}+m^{\frac{1}{2}-2c}\big)$. Then combining (B.14), (B.15), (B.16) and (B.17) together yields

$$G_1(\mathbf{W}_t) - G_1(\mathbf{W}_t^{(i)}) \geq \eta\big(1-4\eta\hat{\rho}\big)\|\nabla G_1(\mathbf{W}_t)\|_2^2 - \frac{\tilde{\epsilon}_t}{2}\|\mathbf{W}_t - \mathbf{W}_t^{(i)} - \eta\nabla G_1(\mathbf{W}_t)\|_2^2,$$

$$G_2(\mathbf{W}_t^{(i)}) - G_2(\mathbf{W}_t) \geq \eta\big(1-4\eta\hat{\rho}\big)\|\nabla G_2(\mathbf{W}_t)\|_2^2 - \frac{\tilde{\epsilon}_t}{2}\|\mathbf{W}_t^{(i)} - \mathbf{W}_t - \eta\nabla G_2(\mathbf{W}_t)\|_2^2.$$

Plugging the above two inequalities back into (B.13) yields

$$\langle\mathbf{W}_t - \mathbf{W}_t^{(i)}, \nabla L_{S\setminus i}(\mathbf{W}_t) - \nabla L_{S\setminus i}(\mathbf{W}_t^{(i)})\rangle = G_1(\mathbf{W}_t) - G_1(\mathbf{W}_t^{(i)}) + G_2(\mathbf{W}_t^{(i)}) - G_2(\mathbf{W}_t)$$

$$\geq 2\eta\big(1-4\eta\hat{\rho}\big)\big\|\nabla L_{S\setminus i}(\mathbf{W}_t)-\nabla L_{S\setminus i}(\mathbf{W}_t^{(i)})\big\|_2^2 - \tilde{\epsilon}_t\big\|\mathbf{W}_t-\mathbf{W}_t^{(i)}-\eta\big(\nabla L_{S\setminus i}(\mathbf{W}_t)-\nabla L_{S\setminus i}(\mathbf{W}_t^{(i)})\big)\big\|_2^2.$$

The desired result has been proved.

Now, we give the proof of (B.16) and (B.17). For $\alpha \in [0,1]$, let $\mathbf{W}(\alpha) = \alpha\mathbf{W}_t + (1-\alpha)\mathbf{W}_t^{(i)} - \alpha\eta\big(\nabla L_{S\setminus i}(\mathbf{W}_t) - \nabla L_{S\setminus i}(\mathbf{W}_t^{(i)})\big)$. For any $\alpha \in [0,1]$, it's obvious that $\|\mathbf{W}(\alpha)\|_2 \leq 2\big(\sqrt{2c_0\eta t} + \|\mathbf{W}_0\|_2\big)$ by using Lemma B.2. Combining this observation with (B.8) we can obtain

$$\lambda_{\min}\big(\nabla^2 L_{S\setminus i}(\mathbf{W}(\alpha))\big)$$

$$\geq -\max_{j\in[n]}\|\nabla^2 f_{\mathbf{W}(\alpha)}(\mathbf{x}_j)\|_{op}\big|f_{\mathbf{W}(\alpha)}(\mathbf{x}_j)-y_j\big|$$

$$\geq -C_{\mathbf{W}(\alpha)}\max_{j\in[n]}\Big(\big|f_{\mathbf{W}(\alpha)}(\mathbf{x}_j)-f_{\mathbf{W}_t^{(i)}}(\mathbf{x}_j)\big|+\big|f_{\mathbf{W}_t^{(i)}}(\mathbf{x}_j)-f_{\mathbf{0}}(\mathbf{x}_j)\big|+\big|f_{\mathbf{0}}(\mathbf{x}_j)-y_j\big|\Big)$$

$$\geq -C_{\mathbf{W}(\alpha)}\Big(\Big(\frac{B_{\sigma'}^2 c_{\mathbf{x}}}{m^{2c-\frac{1}{2}}}\|\mathbf{W}(\alpha)\|_2+\frac{B_{\sigma'}B_{\sigma}}{m^{2c-1}}\Big)\big(\|\mathbf{W}(\alpha)-\mathbf{W}_t^{(i)}\|_2+\|\mathbf{W}_t^{(i)}\|_2\big)+\sqrt{2c_0}\Big)$$

$$\geq -C_{\mathbf{W}(\alpha)}\Big(B_3\big(\frac{\sqrt{\eta T}+\|\mathbf{W}_0\|_2}{m^{2c-\frac{1}{2}}}+m^{1-2c}\big)\big(\|\mathbf{W}_t-\mathbf{W}_t^{(i)}\|_2+\eta\|\nabla\ell(\mathbf{W}_t)-\ell(\mathbf{W}_t^{(i)})\|_2$$

$$\quad +\|\mathbf{W}_t^{(i)}\|_2\big)+\sqrt{2c_0}\Big)$$

$$\geq -\tilde{\epsilon}_t, \tag{B.18}$$

where the second inequality is due to (B.6), the third inequality is according to (B.1) and in the last inequality we used Lemma B.2. Similarly, let $\widetilde{\mathbf{W}}(\alpha) = \alpha\mathbf{W}_t^{(i)} + (1-\alpha)\mathbf{W}_t - \alpha\eta\big(\nabla L_{S\setminus i}(\mathbf{W}_t^{(i)}) - \nabla L_{S\setminus i}(\mathbf{W}_t)\big)$, we can also control $\lambda_{\min}\big(\nabla^2 L_{S\setminus i}(\widetilde{\mathbf{W}}(\alpha))\big)$ by $-\tilde{\epsilon}_t$.

Let $\Delta = \big\|\mathbf{W}_t - \mathbf{W}_t^{(i)} - \eta\big(\nabla L_{S\setminus i}(\mathbf{W}_t) - \nabla L_{S\setminus i}(\mathbf{W}_t^{(i)})\big)\big\|_2^2$. We define

$$g_1(\alpha) = G_1(\mathbf{W}(\alpha)) + \frac{\tilde{\epsilon}_t\alpha^2}{2}\Delta, \qquad g_2(\alpha) = G_2(\widetilde{\mathbf{W}}(\alpha)) + \frac{\tilde{\epsilon}_t\alpha^2}{2}\Delta.$$

From (B.18) we know that $g_1''(\alpha) \geq 0$ for any $\alpha \in [0,1]$. Hence, $g_1$ is convex on $[0,1]$. Then from the convexity of $g_1$ we can get that

$$0 = g_1'(0) \leq g_1(1) - g_1(0) \leq G_1(\mathbf{W}_t - \eta\nabla G_1(\mathbf{W}_t)) + \frac{\tilde{\epsilon}_t}{2}\Delta - G_1(\mathbf{W}_t^{(i)}),$$

which completes the proof of (B.16). We can also show $g_2(\alpha)$ is convex on $[0,1]$ and prove (B.17) in a similar way. The proof is completed. $\qquad\square$

Based on Lemma B.1, Lemma B.2 and Lemma B.3, we can establish the following uniform stability bounds for three-layer neural networks.

**Theorem B.4** (Uniform Stability). *Suppose Assumptions 1 and 2 hold. Let $S$ and $S^{(i)}$ be constructed in Definition 2. Let $\{\mathbf{W}_t\}$ and $\{\mathbf{W}_t^{(i)}\}$ be produced by* (2) *with $\eta \le 1/(8\hat{\rho})$ based on $S$ and $S^{(i)}$, respectively. Assume* (6) *holds. Then, for any $t \in [T]$, there holds*

$$\left\|\mathbf{W}_t - \mathbf{W}_t^{(i)}\right\|_2 \le \frac{2e\eta T\sqrt{2c_0\hat{\rho}(\hat{\rho}\eta T + 2)}}{n}.$$

*Proof.* Similar to (A.7), by the update rule $\mathbf{W}_{t+1} = \mathbf{W}_t - \eta\nabla L_S(\mathbf{W}_t)$ we know

$$\left\|\mathbf{W}_{t+1} - \mathbf{W}_{t+1}^{(i)}\right\|_2^2$$
$$\le (1+p)\left\|\mathbf{W}_t - \mathbf{W}_t^{(i)} - \eta\big(\nabla L_{S\backslash i}(\mathbf{W}_t) - \nabla L_{S\backslash i}(\mathbf{W}_t^{(i)})\big)\right\|_2^2$$
$$+ \frac{2\eta^2(1+1/p)}{n^2}\big(\left\|\nabla\ell(\mathbf{W}_t; z_i)\right\|_2^2 + \left\|\nabla\ell(\mathbf{W}_t^{(i)}; z_i')\right\|_2^2\big). \tag{B.19}$$

From Lemma B.3 we know

$$\left\|\mathbf{W}_t - \mathbf{W}_t^{(i)} - \eta\big(\nabla L_{S\backslash i}(\mathbf{W}_t) - \nabla L_{S\backslash i}(\mathbf{W}_t^{(i)})\big)\right\|_2^2$$
$$= \left\|\mathbf{W}_t - \mathbf{W}_t^{(i)}\right\|_2^2 + \eta^2\left\|\nabla L_{S\backslash i}(\mathbf{W}_t) - \nabla L_{S\backslash i}(\mathbf{W}_t^{(i)})\right\|_2^2$$
$$- 2\eta\Big\langle\mathbf{W}_t - \mathbf{W}_t^{(i)}, \nabla L_{S\backslash i}(\mathbf{W}_t) - \nabla L_{S\backslash i}(\mathbf{W}_t^{(i)})\Big\rangle$$
$$\le \left\|\mathbf{W}_t - \mathbf{W}_t^{(i)}\right\|_2^2 + \eta^2\left\|\nabla L_{S\backslash i}(\mathbf{W}_t) - \nabla L_{S\backslash i}(\mathbf{W}_t^{(i)})\right\|_2^2$$
$$- 4\eta^2\big(1 - 4\eta\hat{\rho}\big)\left\|\nabla L_{S\backslash i}(\mathbf{W}_t) - \nabla L_{S\backslash i}(\mathbf{W}_t^{(i)})\right\|_2^2$$
$$+ 2\eta\tilde{\epsilon}_t\left\|\mathbf{W}_t - \mathbf{W}_t^{(i)} - \eta\big(\nabla L_{S\backslash i}(\mathbf{W}_t) - \nabla L_{S\backslash i}(\mathbf{W}_t^{(i)})\big)\right\|_2^2.$$

Note $4\eta\hat{\rho} \le 1/2$ implies $1 - 4(1 - 4\eta\hat{\rho}) < 0$ and condition (6) ensures that $2\eta\tilde{\epsilon}_t < 1$ for any $t \in [T]$. Then from the above inequality we can get

$$(1 - 2\eta\tilde{\epsilon}_t)\left\|\mathbf{W}_t - \mathbf{W}_t^{(i)} - \eta\big(\nabla L_{S\backslash i}(\mathbf{W}_t) - \nabla L_{S\backslash i}(\mathbf{W}_t^{(i)})\big)\right\|_2^2 \le \left\|\mathbf{W}_t - \mathbf{W}_t^{(i)}\right\|_2^2. \tag{B.20}$$

Now, plugging (B.20) back into (B.19) we have

$$\left\|\mathbf{W}_{t+1} - \mathbf{W}_{t+1}^{(i)}\right\|_2^2 \le \frac{1+p}{1-2\eta\tilde{\epsilon}_t}\left\|\mathbf{W}_t - \mathbf{W}_t^{(i)}\right\|_2^2 + \frac{2\eta^2(1+1/p)}{n^2}\Big(\left\|\nabla\ell(\mathbf{W}_t; z_i)\right\|_2^2 + \left\|\nabla\ell(\mathbf{W}_t^{(i)}; z_i')\right\|_2^2\Big).$$

Applying the above inequality recursively and noting that $\mathbf{W}_0 = \mathbf{W}_0^{(i)}$ we get

$$\left\|\mathbf{W}_{t+1} - \mathbf{W}_{t+1}^{(i)}\right\|_2^2 \le \frac{2\eta^2(1+1/p)}{n^2}\sum_{j=0}^{t}\big(\left\|\nabla\ell(\mathbf{W}_j; z_i)\right\|_2^2 + \left\|\nabla\ell(\mathbf{W}_j^{(i)}; z_i')\right\|_2^2\big)\prod_{\tilde{j}=j+1}^{t}\frac{1+p}{1-2\eta\tilde{\epsilon}_{\tilde{j}}}$$

$$\le \frac{2\eta^2(1+1/p)(1+p)^t}{n^2(1-2\eta\tilde{\epsilon}_j)^t}\sum_{j=0}^{t}\big(\left\|\nabla\ell(\mathbf{W}_j; z_i)\right\|_2^2 + \left\|\nabla\ell(\mathbf{W}_j^{(i)}; z_i')\right\|_2^2\big).$$

Let $p = 1/t$ and note that $(1 + 1/t)^t \le e$, we have

$$\left\|\mathbf{W}_{t+1} - \mathbf{W}_{t+1}^{(i)}\right\|_2^2 \le \frac{2e\eta^2(1+t)}{n^2}\sum_{j=0}^{t}\big(\left\|\nabla\ell(\mathbf{W}_j; z_i)\right\|_2^2 + \left\|\nabla\ell(\mathbf{W}_j^{(i)}; z_i')\right\|_2^2\big)\prod_{\tilde{j}=j+1}^{t}\frac{1}{1-2\eta\tilde{\epsilon}_{\tilde{j}}}. \tag{B.21}$$

According to Lemma B.1 and Lemma A.1, Assumption 2 and noting that $\|\mathbf{W}_j - \mathbf{W}_0\|_2 \le \sqrt{2c_0\eta j}$ for any $j \le t$, we know

$$\|\nabla\ell(\mathbf{W}_j; z)\|_2^2 \le 2\|\nabla\ell(\mathbf{W}_j; z) - \nabla\ell(\mathbf{W}_0; z)\|_2^2 + 2\|\nabla\ell(\mathbf{W}_0; z)\|_2^2$$
$$\le 2\hat{\rho}^2\|\mathbf{W}_j - \mathbf{W}_0\|_2^2 + 4\hat{\rho}\ell(\mathbf{W}_0; z) \le 4c_0\hat{\rho}(\hat{\rho}\eta j + 1).$$

Similarly, we have

$$\|\nabla\ell(\mathbf{W}_j^{(i)}; z)\|_2^2 \le 4c_0\hat{\rho}(\hat{\rho}\eta j + 1).$$

Combining the above three inequalities together, we get

$$\left\| \mathbf{W}_{t+1} - \mathbf{W}_{t+1}^{(i)} \right\|_2^2 \leq \frac{8c_0 e \eta^2 (1+t)^2 \hat{\rho}\left(\hat{\rho}\eta t + 2\right)}{n^2} \prod_{\tilde{j}=j+1}^{t} \frac{1}{1 - 2\eta \tilde{\epsilon}_{\tilde{j}}}.$$

Similar to the proof of Theorem A.6, we can derive the following stability result by induction

$$\left\| \mathbf{W}_{t+1} - \mathbf{W}_{t+1}^{(i)} \right\|_2 \leq \frac{2e\eta T \sqrt{2c_0 \hat{\rho}(\hat{\rho}\eta T + 2)}}{n}.$$

Here, the condition $\frac{1}{(1-2\eta\tilde{\epsilon}_j)^t} \leq \left(\frac{1}{1-1/(t+1)}\right)^t \leq e$ is ensured by condition (6), i.e.,

$$m \gtrsim \left((\eta T \mathcal{B}_T)^2 + \frac{(\eta T)^{\frac{7}{2}} \mathcal{B}_T}{n}\right)^{\frac{1}{5c-\frac{1}{2}}} + \left((\eta T)^{\frac{3}{2}} \mathcal{B}_T^2 + \frac{(\eta T)^3 \mathcal{B}_T}{n}\right)^{\frac{1}{4c-1}} + \left((\eta T)^2 \mathcal{B}_T + \frac{(\eta T)^{\frac{7}{2}}}{n}\right)^{\frac{1}{5c-1}}$$

$$+ \left((\eta T)^{\frac{3}{2}} \mathcal{B}_T + \frac{(\eta T)^3}{n}\right)^{\frac{1}{4c-\frac{3}{2}}}$$

with $\mathcal{B}_T = \sqrt{\eta T} + \|\mathbf{W}_0\|_2$. The proof is the same as Theorem A.6. We omit it for simplicity. $\square$

We can combine Theorem B.4 and Lemma A.2 together to get the upper bound of the generalization error.

*Proof of Theorem 5.* The proof is similar to that of Theorem 1. From (B.21) we know that

$$\left\| \mathbf{W}_{t+1} - \mathbf{W}_{t+1}^{(i)} \right\|_2^2 \leq \frac{4e^2 \eta^2 \hat{\rho}(1+t)}{n^2} \sum_{j=0}^{t} \left(\ell(\mathbf{W}_j; z_i) + \ell(\mathbf{W}_j^{(i)}; z_i')\right),$$

where we used the self-bounding property of the smooth loss (Lemma A.1).

Then, taking an average over $i \in [n]$ and noting that $\mathbb{E}\left[\ell(\mathbf{W}_j; z_i)\right] = \mathbb{E}\left[\ell(\mathbf{W}_j^{(i)}; z_i')\right]$, we have

$$\frac{1}{n} \sum_{i=1}^{n} \mathbb{E}\left\| \mathbf{W}_{t+1} - \mathbf{W}_{t+1}^{(i)} \right\|_2^2 \leq \frac{8e^2 \eta^2 \hat{\rho}(1+t)}{n^2} \sum_{j=0}^{t} \mathbb{E}\left[L_S(\mathbf{W}_j)\right].$$

Combining the above stability bounds with Lemma A.2 together and noting that $L_S(\mathbf{W}_t) \leq \frac{1}{t} \sum_{j=1}^{t-1} L_S(\mathbf{W}_j)$ (Richards & Kuzborskij, 2021), the desired result is obtained. $\square$

### B.2 PROOFS OF OPTIMIZATION BOUNDS

To show optimization error bounds, we first introduce the following lemma on the bound of GD iterates.

**Lemma B.5.** *Suppose Assumptions 1 and 2 hold, and $\eta \leq 1/(8\hat{\rho})$. Assume (6) and (7) hold. Then for any $t \in [T]$, there holds*

$$1 \vee \mathbb{E}[\|\mathbf{W}_{\frac{1}{\eta T}}^* - \mathbf{W}_t\|_2^2]$$

$$\leq \left(\frac{16e^2 \eta^3 t^2 \hat{\rho}^2}{n^2} + \frac{16e\eta^2 t \hat{\rho}}{n}\right) \sum_{s=0}^{t-1} \mathbb{E}\left[L_S(\mathbf{W}_s)\right] + 2\|\mathbf{W}_{\frac{1}{\eta T}}^* - \mathbf{W}_0\|_2^2 + 2\eta T\left[L(\mathbf{W}_{\frac{1}{\eta T}}^*) - L(\mathbf{W}^*)\right].$$

*Proof.* For any $\mathbf{W} \in \mathbb{R}^{md}$ and $\alpha \in [0,1]$, define $\mathbf{W}(\alpha) := \mathbf{W}_t + \alpha(\mathbf{W} - \mathbf{W}_t)$. Similar to (B.18), according to Lemma B.1 we can show that

$$\lambda_{\min}\left(\nabla^2 L_S(\mathbf{W}(\alpha))\right) \geq -C_{\mathbf{W}(\alpha)}\left(\left(\frac{B_{\sigma'}^2 c_\mathbf{x}}{m^{2c-1/2}}\|\mathbf{W}(\alpha)\|_2 + \frac{B_{\sigma'} B_\sigma}{m^{2c-1}}\right)\left(\|\mathbf{W} - \mathbf{W}_t\|_2 + \|\mathbf{W}_t\|_2\right) + \sqrt{2c_0}\right),$$

where $C_{\mathbf{W}(\alpha)} = \frac{B_{\sigma'}^2 B_{\sigma''} c_\mathbf{x}^2}{m^{3c}}\|\mathbf{W}(\alpha)^{(2)}\|_2^2 + \left(\frac{B_{\sigma'} B_{\sigma''} c_\mathbf{x}^2}{m^{2c-\frac{1}{2}}} + \frac{2B_{\sigma''} B_{\sigma'} B_\sigma c_\mathbf{x}}{m^{3c-\frac{1}{2}}}\right)\|\mathbf{W}(\alpha)^{(2)}\|_2 + \frac{B_{\sigma''} B_\sigma^2}{m^{3c-1}} + \frac{2B_{\sigma'}^2 c_\mathbf{x}}{m^{2c-\frac{1}{2}}}$. Let $\hat{C}_\mathbf{W} = 4B_1\left(m^{-3c}(\|\mathbf{W}\|_2^2 + 4c_0\eta T + 2\|\mathbf{W}_0\|_2^2) + m^{\frac{1}{2}-2c}(\|\mathbf{W}\|_2 + \sqrt{2c_0\eta T} + \|\mathbf{W}_0\|_2)\right)$. According to Lemma B.2, we can verify that $C_{\mathbf{W}(\alpha)} \leq \hat{C}_\mathbf{W}$ for any $\alpha \in [0,1]$.

Now, let

$$g(\alpha):=L_S(\mathbf{W}(\alpha))+\frac{\alpha^2\hat{C}_{\mathbf{W}}}{2}\Big(\big(\frac{B_{\sigma'}^2 c_{\mathbf{x}}}{m^{2c-1/2}}\|\mathbf{W}(\alpha)\|_2+\frac{B_{\sigma'}B_\sigma}{m^{2c-1}}\big)\big(\|\mathbf{W}-\mathbf{W}_t\|_2+\|\mathbf{W}_t\|_2\big)+\sqrt{2c_0}\Big)$$
$$\times(1\vee\mathbb{E}[\|\mathbf{W}-\mathbf{W}_t\|_2^2]).$$

It is obvious that $g(\alpha)$ is convex in $\alpha\in[0,1]$. Similar to the proof of Lemma A.7, by convexity of $g$ and smoothness of the loss we can show that

$$\frac{1}{t}\sum_{s=0}^{t-1}\mathbb{E}[L_S(\mathbf{W}_s)]+\frac{\mathbb{E}\big[\|\mathbf{W}_t-\mathbf{W}\|_2^2\big]}{2\eta t}$$

$$\leq\mathbb{E}[L_S(\mathbf{W})]+\frac{\mathbb{E}\big[\|\mathbf{W}-\mathbf{W}_0\|_2^2\big]}{2\eta t}+\frac{\hat{C}_{\mathbf{W}}}{2t}\sum_{s=0}^{t-1}\Big(\big(\frac{B_{\sigma'}^2 c_{\mathbf{x}}}{m^{2c-1/2}}\|\mathbf{W}(\alpha)\|_2+\frac{B_{\sigma'}B_\sigma}{m^{2c-1}}\big)\big(\mathbb{E}[\|\mathbf{W}-\mathbf{W}_s\|_2]$$

$$+\sqrt{2c_0\eta T}+\|\mathbf{W}_0\|_2\big)+\sqrt{2c_0}\Big)(1\vee\mathbb{E}[\|\mathbf{W}-\mathbf{W}_s\|_2^2]).\tag{B.22}$$

Combining the above inequality with Theorem 5 and let $\mathbf{W}=\mathbf{W}^*_{\frac{1}{\eta T}}$, we have

$$\frac{\mathbb{E}\big[\|\mathbf{W}_t-\mathbf{W}^*_{\frac{1}{\eta T}}\|_2^2\big]}{2\eta t}$$

$$\leq\frac{1}{t}\sum_{s=1}^{t-1}\big[L(\mathbf{W}^*_{\frac{1}{\eta T}})-\mathbb{E}[L(\mathbf{W}_s)]\big]+\frac{\|\mathbf{W}^*_{\frac{1}{\eta T}}-\mathbf{W}_0\|_2^2}{2\eta t}+\Big(\frac{4e^2\eta^2 t\hat{\rho}^2}{n^2}+\frac{4e\eta\hat{\rho}}{n}\Big)\sum_{s=0}^{t-1}\mathbb{E}\big[L_S(\mathbf{W}_s)\big]$$

$$+\frac{1}{2t}\sum_{s=0}^{t-1}\hat{C}_{\mathbf{W}^*_{\frac{1}{\eta T}}}\Big(\big(\frac{B_{\sigma'}^2 c_{\mathbf{x}}}{m^{2c-1/2}}\|\mathbf{W}(\alpha)\|_2+\frac{B_{\sigma'}B_\sigma}{m^{2c-1}}\big)\big(\mathbb{E}[\|\mathbf{W}^*_{\frac{1}{\eta T}}-\mathbf{W}_s\|_2]+\sqrt{2c_0\eta T}+\|\mathbf{W}_0\|_2\big)+\sqrt{2c_0}\Big)$$

$$\times(1\vee\mathbb{E}[\|\mathbf{W}^*_{\frac{1}{\eta T}}-\mathbf{W}_s\|_2^2])$$

$$\leq L(\mathbf{W}^*_{\frac{1}{\eta T}})-L(\mathbf{W}^*)+\frac{\|\mathbf{W}^*_{\frac{1}{\eta T}}-\mathbf{W}_0\|_2^2}{2\eta t}+\Big(\frac{4e^2\eta^2 t\hat{\rho}^2}{n^2}+\frac{4e\eta\hat{\rho}}{n}\Big)\sum_{s=0}^{t-1}\mathbb{E}\big[L_S(\mathbf{W}_s)\big]$$

$$+\frac{1}{2t}\sum_{s=0}^{t-1}\hat{C}_{\mathbf{W}^*_{\frac{1}{\eta T}}}\Big(\big(\frac{B_{\sigma'}^2 c_{\mathbf{x}}}{m^{2c-1/2}}\|\mathbf{W}(\alpha)\|_2+\frac{B_{\sigma'}B_\sigma}{m^{2c-1}}\big)\big(\|\mathbf{W}^*_{\frac{1}{\eta T}}-\mathbf{W}_s\|_2+\sqrt{2c_0\eta T}+\|\mathbf{W}_0\|_2\big)+\sqrt{2c_0}\Big)$$

$$\times(1\vee\mathbb{E}[\|\mathbf{W}^*_{\frac{1}{\eta T}}-\mathbf{W}_s\|_2^2]),\tag{B.23}$$

where in the second inequality we used $L(\mathbf{W}^*_{\frac{1}{\eta T}})-L(\mathbf{W}_s)\leq L(\mathbf{W}^*_{\frac{1}{\eta T}})-L(\mathbf{W}^*)$ since $L(\mathbf{W}_s)\geq L(\mathbf{W}^*)$ for any $s\in[t-1]$.

According to Lemma B.2 we can get

$$\|\mathbf{W}^*_{\frac{1}{\eta T}}-\mathbf{W}_s\|_2\leq\|\mathbf{W}^*_{\frac{1}{\eta T}}-\mathbf{W}_0\|_2+\|\mathbf{W}_s-\mathbf{W}_0\|_2\leq\|\mathbf{W}^*_{\frac{1}{\eta T}}-\mathbf{W}_0\|_2+\sqrt{2c_0\eta T}.\tag{B.24}$$

Then there holds

$$\big(\frac{B_{\sigma'}^2 c_{\mathbf{x}}}{m^{2c-1/2}}\|\mathbf{W}(\alpha)\|_2+\frac{B_{\sigma'}B_\sigma}{m^{2c-1}}\big)\big(\|\mathbf{W}^*_{\frac{1}{\eta T}}-\mathbf{W}_s\|_2+\sqrt{2c_0\eta T}+\|\mathbf{W}_0\|_2\big)+\sqrt{2c_0}$$

$$\leq\Big(\frac{B_{\sigma'}^2 c_{\mathbf{x}}}{m^{2c-1/2}}(2\sqrt{2c_0\eta T}+2\|\mathbf{W}_0\|_2+\|\mathbf{W}^*_{\frac{1}{\eta T}}-\mathbf{W}_0\|_2)+\frac{B_{\sigma'}B_\sigma}{m^{2c-1}}\Big)(2\sqrt{2c_0\eta T}+\|\mathbf{W}_0\|_2$$

$$+\|\mathbf{W}^*_{\frac{1}{\eta T}}-\mathbf{W}_0\|_2)+\sqrt{2c_0}.\tag{B.25}$$

Plugging the above inequality back into (B.23) and multiplying both sides by $2\eta t$ yields

$$\mathbb{E}\big[\|\mathbf{W}_t - \mathbf{W}^*_{\frac{1}{\eta T}}\|_2^2\big]$$

$$\le \|\mathbf{W}^*_{\frac{1}{\eta T}} - \mathbf{W}_0\|_2^2 + \eta\hat{C}_{\mathbf{W}^*_{\frac{1}{\eta T}}}\bigg(\Big(\frac{B_{\sigma'}^2 c_{\mathbf{x}}}{m^{2c-1/2}}(2\sqrt{2c_0\eta T} + \|\mathbf{W}_0\|_2 + \|\mathbf{W}^*_{\frac{1}{\eta T}} - \mathbf{W}_0\|_2) + \frac{B_{\sigma'}B_\sigma}{m^{2c-1}}\Big)$$

$$\times (2\sqrt{2\eta T c_0} + \|\mathbf{W}_0\|_2 + \|\mathbf{W}^*_{\frac{1}{\eta T}} - \mathbf{W}_0\|_2) + \sqrt{2c_0}\bigg) \sum_{s=0}^{t-1}(1 \vee \mathbb{E}[\|\mathbf{W}^*_{\frac{1}{\eta T}} - \mathbf{W}_s\|_2^2])$$

$$+ \Big(\frac{8e^2\eta^3 t^2\hat{\rho}^2}{n^2} + \frac{8e\eta^2 t\hat{\rho}}{n}\Big)\sum_{s=0}^{t-1}\mathbb{E}[L_S(\mathbf{W}_s)] + 2\eta T\big[L(\mathbf{W}^*_{\frac{1}{\eta T}}) - L(\mathbf{W}^*)\big].$$

Let $x = \max_{s\in[T]}\mathbb{E}[\|\mathbf{W}^*_{\frac{1}{\eta T}} - \mathbf{W}_s\|_2^2] \vee 1$. Then the above inequality implies

$$x \le \|\mathbf{W}^*_{\frac{1}{\eta T}} - \mathbf{W}_0\|_2^2 + \eta T\hat{C}_{\mathbf{W}^*_{\frac{1}{\eta T}}}\bigg(\Big(\frac{B_{\sigma'}^2 c_{\mathbf{x}}}{m^{2c-1/2}}(2\sqrt{2c_0\eta T} + \|\mathbf{W}_0\|_2 + \|\mathbf{W}^*_{\frac{1}{\eta T}} - \mathbf{W}_0\|_2) + \frac{B_{\sigma'}B_\sigma}{m^{2c-1}}\Big)$$

$$\times (2\sqrt{2\eta T c_0} + \|\mathbf{W}_0\|_2 + \|\mathbf{W}^*_{\frac{1}{\eta T}} - \mathbf{W}_0\|_2) + \sqrt{2c_0}\bigg)x + \Big(\frac{8e^2\eta^3 t^2\hat{\rho}^2}{n^2} + \frac{8e\eta^2 t\hat{\rho}}{n}\Big)\sum_{s=0}^{t-1}\mathbb{E}[L_S(\mathbf{W}_s)]$$

$$+ 2\eta T\big[L(\mathbf{W}^*_{\frac{1}{\eta T}}) - L(\mathbf{W}^*)\big].$$

Note that condition (7) implies that $\eta T\hat{C}_{\mathbf{W}^*_{\frac{1}{\eta T}}}\big(\big(\frac{B_{\sigma'}^2 c_{\mathbf{x}}}{m^{2c-1/2}}(2\sqrt{2c_0\eta T} + \|\mathbf{W}_0\|_2 + \|\mathbf{W}^*_{\frac{1}{\eta T}} - \mathbf{W}_0\|_2) + \frac{B_{\sigma'}B_\sigma}{m^{2c-1}}\big)(2\sqrt{2\eta T c_0} + \|\mathbf{W}_0\|_2 + \|\mathbf{W}^*_{\frac{1}{\eta T}} - \mathbf{W}_0\|_2) + \sqrt{2c_0}\big) \le \frac{1}{2}$. Then there holds

$$x \le \Big(\frac{16e^2\eta^3 t^2\hat{\rho}^2}{n^2} + \frac{16e\eta^2 t\hat{\rho}}{n}\Big)\sum_{s=0}^{t-1}\mathbb{E}[L_S(\mathbf{W}_s)] + 2\|\mathbf{W}^*_{\frac{1}{\eta T}} - \mathbf{W}_0\|_2^2 + 2\eta T\big[L(\mathbf{W}^*_{\frac{1}{\eta T}}) - L(\mathbf{W}^*)\big].$$

It then follows that

$$1 \vee \mathbb{E}[\|\mathbf{W}^*_{\frac{1}{\eta T}} - \mathbf{W}_t\|_2^2]$$

$$\le \Big(\frac{16e^2\eta^3 t^2\hat{\rho}^2}{n^2} + \frac{16e\eta^2 t\hat{\rho}}{n}\Big)\sum_{s=0}^{t-1}\mathbb{E}[L_S(\mathbf{W}_s)] + 2\|\mathbf{W}^*_{\frac{1}{\eta T}} - \mathbf{W}_0\|_2^2 + 2\eta T\big[L(\mathbf{W}^*_{\frac{1}{\eta T}}) - L(\mathbf{W}^*)\big].$$

This completes the proof. $\qquad\square$

*Proof of Theorem 6.* Combining (B.25) and (B.22) with $\mathbf{W} = \mathbf{W}^*_{\frac{1}{\eta T}}$ together yields

$$\frac{1}{T}\sum_{s=0}^{T-1}\mathbb{E}[L_S(\mathbf{W}_s)]$$

$$\le \mathbb{E}[L_S(\mathbf{W}^*_{\frac{1}{\eta T}})] + \frac{\|\mathbf{W}^*_{\frac{1}{\eta T}} - \mathbf{W}_0\|_2^2}{2\eta T} + \frac{\hat{C}_{\mathbf{W}^*_{\frac{1}{\eta T}}}}{2T}\sum_{s=0}^{T-1}\bigg(\Big(\frac{B_{\sigma'}^2 c_{\mathbf{x}}}{m^{2c-1/2}}(2\sqrt{2c_0\eta T} + \|\mathbf{W}_0\|_2 + \|\mathbf{W}^*_{\frac{1}{\eta T}} - \mathbf{W}_0\|_2)$$

$$+ \frac{B_{\sigma'}B_\sigma}{m^{2c-1}}\Big)\big(\mathbb{E}[\|\mathbf{W}^*_{\frac{1}{\eta T}} - \mathbf{W}_s\|_2] + 2\sqrt{2c_0\eta T} + \|\mathbf{W}_0\|_2\big) + \sqrt{2c_0}\Big)(1 \vee \mathbb{E}[\|\mathbf{W}^*_{\frac{1}{\eta T}} - \mathbf{W}_s\|_2^2])$$

$$\le \mathbb{E}[L_S(\mathbf{W}^*_{\frac{1}{\eta T}})] + \frac{1}{2T}\hat{C}_{\mathbf{W}^*_{\frac{1}{\eta T}}}\hat{B}_{\mathbf{W}^*_{\frac{1}{\eta T}}}\sum_{s=0}^{T-1}1 \vee \mathbb{E}[\|\mathbf{W}^*_{\frac{1}{\eta T}} - \mathbf{W}_s\|_2^2] + \frac{\|\mathbf{W}^*_{\frac{1}{\eta T}} - \mathbf{W}_0\|_2^2}{2\eta T}, \quad \text{(B.26)}$$

where $\hat{B}_{\mathbf{W}^*_{\frac{1}{\eta T}}} = \big(\frac{B_{\sigma'}^2 c_{\mathbf{x}}}{m^{2c-1/2}}(2\sqrt{2c_0\eta T} + \|\mathbf{W}^*_{\frac{1}{\eta T}} - \mathbf{W}_0\|_2) + \frac{B_{\sigma'}B_\sigma}{m^{2c-1}}\big)(2\sqrt{2\eta T c_0} + \|\mathbf{W}_0\|_2 + \|\mathbf{W}^*_{\frac{1}{\eta T}} - \mathbf{W}_0\|_2) + \sqrt{2c_0}$ and in the last inequality we used (B.24).

By monotonic decrease of $\{L_S(\mathbf{W}_t)\}$ and Lemma B.5, we further know

$$\mathbb{E}[L_S(\mathbf{W}_T)] \leq \mathbb{E}[L_S(\mathbf{W}^*_{\frac{1}{\eta T}})] + \frac{\hat{C}_{\mathbf{W}^*_{\frac{1}{\eta T}}} \hat{B}_{\mathbf{W}^*_{\frac{1}{\eta T}}}}{2T} \sum_{s=0}^{T-1} 1 \vee \mathbb{E}[\|\mathbf{W}^*_{\frac{1}{\eta T}} - \mathbf{W}_s\|_2^2] + \frac{\|\mathbf{W}^*_{\frac{1}{\eta T}} - \mathbf{W}_0\|_2^2}{2\eta T}$$

$$\leq \mathbb{E}[L_S(\mathbf{W}^*_{\frac{1}{\eta T}})] + \hat{C}_{\mathbf{W}^*_{\frac{1}{\eta T}}} \hat{B}_{\mathbf{W}^*_{\frac{1}{\eta T}}} \left( \left( \frac{8e^2 \eta^3 T^2 \hat{\rho}^2}{n^2} + \frac{8e\eta^2 T \hat{\rho}}{n} \right) \sum_{s=0}^{T-1} \mathbb{E}[L_S(\mathbf{W}_s)] \right.$$

$$+ \|\mathbf{W}^*_{\frac{1}{\eta T}} - \mathbf{W}_0\|_2^2 + \eta T \left[ L(\mathbf{W}^*_{\frac{1}{\eta T}}) - L(\mathbf{W}^*) \right] \bigg) + \frac{\|\mathbf{W}^*_{\frac{1}{\eta T}} - \mathbf{W}_0\|_2^2}{2\eta T},$$

which completes the proof. $\qquad\square$

Recall that $\mathcal{C}_{T,n} = \eta T + \eta^3 T^2 / n^2$.

**Lemma B.6.** *Suppose Assumptions 1 and 2 hold. Let $\{\mathbf{W}_t\}$ be produced by (2) with $\eta \leq 1/(8\hat{\rho})$. Assume (6) and (7) hold. Then*

$$\sum_{s=0}^{T-1} \mathbb{E}[L_S(\mathbf{W}_s)] \leq 2TL(\mathbf{W}^*_{\frac{1}{\eta T}}) - \eta T L(\mathbf{W}^*) + \left( 1 + \frac{1}{2\eta} \right) \|\mathbf{W}^*_{\frac{1}{\eta T}} - \mathbf{W}_0\|_2^2.$$

*Proof.* Multiplying $T$ over both sides of (B.26) and using Lemma B.5 we get

$$\sum_{s=0}^{T-1} \mathbb{E}[L_S(\mathbf{W}_s)] \leq TL(\mathbf{W}^*_{\frac{1}{\eta T}}) + \hat{C}_{\mathbf{W}^*_{\frac{1}{\eta T}}} \hat{B}_{\mathbf{W}^*_{\frac{1}{\eta T}}} \left( \left( \frac{8e^2 \eta^3 T^2 \hat{\rho}^2}{n^2} + \frac{8e\eta^2 T \hat{\rho}}{n} \right) \sum_{s=0}^{T-1} \mathbb{E}[L_S(\mathbf{W}_s)] \right.$$

$$+ \|\mathbf{W}^*_{\frac{1}{\eta T}} - \mathbf{W}_0\|_2^2 + \eta T \left[ L(\mathbf{W}^*_{\frac{1}{\eta T}}) - L(\mathbf{W}^*) \right] \bigg) + \frac{\|\mathbf{W}^*_{\frac{1}{\eta T}} - \mathbf{W}_0\|_2^2}{2\eta}. \quad \text{(B.27)}$$

Condition (7) implies $\hat{C}_{\mathbf{W}^*_{\frac{1}{\eta T}}} \hat{B}_{\mathbf{W}^*_{\frac{1}{\eta T}}} \left( \frac{4e^2 \eta^3 T^2 \hat{\rho}^2}{n^2} + \frac{4e\eta^2 T \hat{\rho}}{n} \right) \leq 1/2$ and $\hat{C}_{\mathbf{W}^*_{\frac{1}{\eta T}}} \hat{B}_{\mathbf{W}^*_{\frac{1}{\eta T}}} \leq 1$. Then there holds

$$\sum_{s=0}^{T-1} \mathbb{E}[L_S(\mathbf{W}_s)] \leq 2TL(\mathbf{W}^*_{\frac{1}{\eta T}}) - \eta T L(\mathbf{W}^*) + \left( 1 + \frac{1}{2\eta} \right) \|\mathbf{W}^*_{\frac{1}{\eta T}} - \mathbf{W}_0\|_2^2,$$

which completes the proof. $\qquad\square$

## B.3 PROOFS OF EXCESS POPULATION BOUNDS

*Proof of Theorem 7.* Note $\hat{\rho} = \mathcal{O}(1)$. According to Theorem 5 and Lemma B.6 we know

$$\mathbb{E}[L(\mathbf{W}_T) - L_S(\mathbf{W}_T)] = \mathcal{O}\left( \left( \frac{\eta^2 T^2}{n^2} + \frac{\eta T}{n} \right) \left( L(\mathbf{W}^*_{\frac{1}{\eta T}}) + \frac{1}{\eta T} \|\mathbf{W}^*_{\frac{1}{\eta T}} - \mathbf{W}_0\|_2^2 \right) \right). \quad \text{(B.28)}$$

The estimation of the optimization error is given by combining Lemma B.6 and Theorem 6 together

$$\mathbb{E}[L_S(\mathbf{W}_T) - L_S(\mathbf{W}^*_{\frac{1}{\eta T}})] - \frac{1}{2\eta T} \|\mathbf{W}^*_{\frac{1}{\eta T}} - \mathbf{W}_0\|_2^2]$$

$$= \mathcal{O}\left( \left( \frac{\eta^2 T^2}{n^2} + \frac{\eta T}{n} \right) \left( L(\mathbf{W}^*_{\frac{1}{\eta T}}) + \frac{1}{2\eta T} \|\mathbf{W}^*_{\frac{1}{\eta T}} - \mathbf{W}_0\|_2^2 \right) + \frac{1}{\eta T} \|\mathbf{W}^*_{\frac{1}{\eta T}} - \mathbf{W}_0\|_2^2 + \Lambda_{\frac{1}{\eta T}} \right),$$

where we used the fact that $\hat{C}_{\mathbf{W}^*_{\frac{1}{\eta T}}} \hat{B}_{\mathbf{W}^*_{\frac{1}{\eta T}}} \leq 1/\eta T$ implied by condition (7) and $L(\mathbf{W}^*_{\frac{1}{\eta T}}) - L(\mathbf{W}^*) \leq \Lambda_{\frac{1}{\eta T}}$.

Combining the above two inequalities together we get

$$
\begin{aligned}
&\mathbb{E}[L(\mathbf{W}_T) - L(\mathbf{W}^*)] \\
&= \Big[ \mathbb{E}[L(\mathbf{W}_T) - L_S(\mathbf{W}_T)] \Big] + \mathbb{E}\Big[ L_S(\mathbf{W}_T) - \big( L_S(\mathbf{W}^*_{\frac{1}{\eta T}}) + \frac{1}{2\eta T}\|\mathbf{W}^*_{\frac{1}{\eta T}} - \mathbf{W}_0\|_2^2 \big) \Big] \\
&\quad + \Big[ L(\mathbf{W}^*_{\frac{1}{\eta T}}) + \frac{1}{2\eta T}\|\mathbf{W}^*_{\frac{1}{\eta T}} - \mathbf{W}_0\|_2^2 - L(\mathbf{W}^*) \Big] \\
&= \mathcal{O}\Big( \frac{\eta T}{n}\big(\frac{\eta T}{n}+1\big)\Big[ L(\mathbf{W}^*_{\frac{1}{\eta T}}) + \frac{1}{2\eta T}\|\mathbf{W}^*_{\frac{1}{\eta T}} - \mathbf{W}_0\|_2^2 \Big] + \frac{1}{\eta T}\|\mathbf{W}^*_{\frac{1}{\eta T}} - \mathbf{W}_0\|_2^2 + \Lambda_{\frac{1}{\eta T}} \Big).
\end{aligned}
$$

Since $n \gtrsim \eta T$, we further have

$$
\mathbb{E}[L(\mathbf{W}_T) - L(\mathbf{W}^*)] = \mathcal{O}\Big( \frac{\eta T}{n}\Big[ L(\mathbf{W}^*_{\frac{1}{\eta T}}) + \frac{1}{2\eta T}\|\mathbf{W}^*_{\frac{1}{\eta T}} - \mathbf{W}_0\|_2^2 \Big] + \frac{1}{\eta T}\|\mathbf{W}^*_{\frac{1}{\eta T}} - \mathbf{W}_0\|_2^2 + \Lambda_{\frac{1}{\eta T}} \Big).
$$

Finally, note that $L(\mathbf{W}^*_{\frac{1}{\eta T}}) + \frac{1}{2\eta T}\|\mathbf{W}^*_{\frac{1}{\eta T}} - \mathbf{W}_0\|_2^2 = L(\mathbf{W}^*) + \Lambda_{\frac{1}{\eta T}}$ and $\|\mathbf{W}^*_{\frac{1}{\eta T}} - \mathbf{W}_0\|_2^2 \le \eta T \Lambda_{\frac{1}{\eta T}}$, we have

$$
\mathbb{E}[L(\mathbf{W}_T) - L(\mathbf{W}^*)] = \mathcal{O}\Big( \frac{\eta T}{n} L(\mathbf{W}^*) + \Lambda_{\frac{1}{\eta T}} \Big).
$$

The proof is completed. $\qquad\square$

*Proof of Corollary 8.* **Part (a)**. From the definition of the approximation error $\Lambda_{\frac{1}{\eta T}}$ and Theorem 7 we can get

$$
\mathbb{E}[L(\mathbf{W}_T) - L(\mathbf{W}^*)] = \mathcal{O}\Big( \frac{\eta T}{n} L(\mathbf{W}^*) + \frac{1}{\eta T}\|\mathbf{W}^* - \mathbf{W}_0\|_2^2 \Big).
$$

For the case $c \in [9/16, 1]$, to ensure conditions (6) and (7) hold, we choose $m \asymp (\eta T)^4$ for this case. Then according to Theorem 7 and Assumption 3, there holds

$$
\mathbb{E}[L(\mathbf{W}_T) - L(\mathbf{W}^*)] = \mathcal{O}\Big( \frac{\eta T}{n} + (\eta T)^{3-8\mu} \Big).
$$

Here, the condition $\mu \ge 1/2$ ensures that $3 - 8\mu < 0$. Hence, the bound will vanish as $\eta T$ tends to 0. Further, if $n^{\frac{1}{2(8\mu-3)}} \lesssim \eta T \lesssim \sqrt{n}$ (the existence of $\eta T$ is ensured by $\mu \ge 1/2$), then there holds $\eta T/n = \mathcal{O}(n^{-1/2})$ and $(\eta T)^{3-8\mu} = \mathcal{O}(n^{-1/2})$. That is

$$
\mathbb{E}[L(\mathbf{W}_T) - L(\mathbf{W}^*)] = \mathcal{O}\Big( \frac{1}{\sqrt{n}} \Big).
$$

For the case $c \in (1/2, 9/16)$, we choose $m \asymp (\eta T)^{\frac{1}{4c-2}}$, and $n^{\frac{2c-1}{2\mu+4c-3}} \lesssim \eta T \lesssim \sqrt{n}$. From Theorem 7 and Assumption 3 we have

$$
\mathbb{E}[L(\mathbf{W}_T) - L(\mathbf{W}^*)] = \mathcal{O}\Big( \frac{1}{\sqrt{n}} \Big).
$$

The first part of the theorem is proved.

**Part (b).** For the case $c \in [9/16, 1]$, by choosing $m \asymp (\eta T)^4$ and $\eta T \gtrsim n^{\frac{1}{8\mu-3}}$, from Theorem 7 and Assumption 3 we have

$$
\mathbb{E}[L(\mathbf{W}_T) - L(\mathbf{W}^*)] = \mathcal{O}\Big( \frac{1}{n} \Big).
$$

For the case $c \in (1/2, 9/16)$, by choosing $m \asymp (\eta T)^{\frac{1}{4c-2}}$ and $\eta T \gtrsim n^{\frac{4c-2}{4c+2\mu-3}}$, from Theorem 7 and Assumption 3 we have

$$
\mathbb{E}[L(\mathbf{W}_T) - L(\mathbf{W}^*)] = \mathcal{O}\Big( \frac{1}{n} \Big).
$$

The proof is completed. $\qquad\square$

### B.4 MORE DISCUSSION ON RELATED WORKS

Richards & Rabbat (2021) derived a lower bound on the minimum eigenvalue of the Hessian for the three-layer NN, where the first layer activation is linear, and the second activation is smooth. They proved the weak convexity of the empirical risk scales with $m^{\frac{1}{2}-2c}$ when optimizing the first and third layers of weights with Lipschitz and convex losses. We train the first and the second layers of the network with general smooth activation functions for both layers. Our result shows that the weak convexity of the least square loss scales with $m^{\frac{1}{2}-2c}$.

