# OpenReview forum: "Generalization Guarantees of Gradient Descent for Multi-Layer Neural Networks"
_ICLR.cc/2024/Conference — Submitted to ICLR 2024_

### Official Review · Reviewer_PsiY · 2023-10-29

**Soundness:** 3 good
**Presentation:** 3 good
**Contribution:** 3 good
**Rating:** 6
**Confidence:** 3

**Summary:**

This paper studies the generalization of training overparameterized neural networks by gradient descent. This work follows the line of work of studying the generalization error via on-average argument stability. This work extends previous work by providing a relaxed condition for two-layer NNs and establishing an optimization and generalization result for 3-layer NNs. This work further shows that as the network scaling factor $c$ (in the $1/m^c$ scaling in front of NNs) increases, less overparameterization is needed in order to achieve the same generalization error.

**Strengths:**

This paper is technical and the results are novel in my opinion. The proofs are solid and nicely structured in the Appendix. On the technical side, I appreciate the authors are able to establish almost co-coercivity results for 3-layler NN and the authors are able to use this to establish the uniform stability bound on $W_t - W_t^{(i)}$. On the other hand, I also like the results on the relationship between the network width and the scaling parameter $c$ and that the authors are able to show that larger $c$ requires less over-parameterization.

**Weaknesses:**

One of the drawback of this analysis is that since the network width $m$ is usually chosen first in practice, this result (such as theorem 2,3) basically says the training can't be too long and $\eta T$ need to be upper bounded by some functions over the network width. Also, this work prove convergence via lower bounding the minimum eigenvalue of the Hessian and uses the Hessian at the initialization as a reference point. This limits how far the network weights can travel from the initialization value. For $L^2$ loss, I think this is OK but for losses like cross-entropy where the network achieve zero-loss at infinity, it is not clear how far this analysis can be generalized.

**Questions:**

1. How are the network weights initialized? It seems the only requirement on initialization is assumption 2.
2. When the authors are talking about network complexity, how is it defined? Especially, the paragraph before section 2 "...the larger the scaling parameter or the simpler the network complexity is...". It seems from assumption 3 that the network complexity is defined as the $L^2$ norm of the weight matrices. Also, why the larger the scaling parameter corresponds to the simpler the network complexity?
3. On page 2, the second bullet of contributions, the authors mentioned that "... due to the empirical risks' monotonically decreasing nature which no longer remains valid for three-layer NNs" Can't taking a smaller step in gradient descent steps help the empirical risks' monotonicity of decreasing?

**Details Of Ethics Concerns:**

None.

---

> ### Author Response · Authors · 2023-11-20
> **Thanks for your careful reading and constructive comments**
>
> >** Q1. One of the drawback of this analysis is that since the network width $m$ is usually chosen first in practice, this result (such as theorem 2,3) basically says the training can't be too long and $\eta T$ need to be upper bounded by some functions over the network width. Also, this work prove convergence via lower bounding the minimum eigenvalue of the Hessian and uses the Hessian at the initialization as a reference point. This limits how far the network weights can travel from the initialization value. For $L^2$ loss, I think this is OK but for losses like cross-entropy where the network achieve zero-loss at infinity, it is not clear how far this analysis can be generalized.
>
> Thanks for your thoughtful comment. In this work, we answer a theoretical question of how wide the network is required for GD to achieve the desired excess risk bound $O({1 / \sqrt{n}})$. Our results also shed light on how to choose the suitable $\eta T$ and $m$ related to data size $n$ to achieve the $O({1 / \sqrt{n}})$ risk rate under the different settings.
>
> In addition, if the network width is chosen first, our results (especially Corollaries 4 and 8 in the revised version) reflect the learning capability of this network. For example, let us consider the case $c=1$ and $\mu=1/2$. Corollary 4 implies, at the theoretical aspect, that if $n\lesssim m^{4/3}$, then GD for two-layer NNs with at least $\sqrt{n}$ iterations can achieve the desired error rate $O(1/\sqrt{n})$. While the estimation of the minimum eigenvalue of the Hessian $\lambda_{\min}(\ell(\mathbf{W};z))$ depends on $||\mathbf{W}-\mathbf{W}_0||_2$, we can show that along the trajectory of GD, $||\mathbf{W}_t-\mathbf{W}_0||_2\le c\sqrt{\eta T}$.  Hence, our analysis does not limit how far the network weights can travel from the initialization value.
> We mainly focus on the least square loss in this work. We agree with you that extending our results to the cross-entropy loss is very interesting and important, which we leave as an important future direction.
>
>
>
>
>
>
>
> >** Q2. How are the network weights initialized? It seems the only requirement on initialization is assumption 2.
>
>
> Yes, we can initialize the network weights randomly as the previous works did or use a fixed initialization as long as Assumption 2 holds.   Indeed, for sigmoid activation or hyperbolic tangent activation considered in our paper, it can be easily verified that Assumption 2 holds for any initialization if the inputs and labels are bounded. That is, for any network initialization $\mathbf{W}_0$ and any data $z=(\mathbf{x},y)$ with $||\mathbf{x}||_2\le c_x$ and $|y|\le c_y$, we can always find a constant $c_0>0$ such that the assumption $\ell(\mathbf{W}_0;z)\le c_0$ holds. We have added the clarification below Assumption 2 in the revised version.

---

> > ### Author Response · Authors · 2023-11-20
> >
> > >** Q3. When the authors are talking about network complexity, how is it defined? Especially, the paragraph before section 2 "...the larger the scaling parameter or the simpler the network complexity is...". It seems from assumption 3 that the network complexity is defined as the $L^2$ norm of the weight matrices. Also, why the larger the scaling parameter corresponds to the simpler the network complexity?
> >
> >
> > The network complexity is the Frobenius norm of the population risk minimizer $\mathbf{W}^*$ defined in Assumption 3 (following the same assumption in [1]). We did not mean that there is a relationship between the network complexity and scaling parameter. Our results imply that both the larger scaling parameter and the simpler network complexity yield less over-parameterization requirement for GD to achieve the desired error rates. We have deleted any misleading sentences that caused such confusion.
> >
> > Indeed, the network complexity assumption is used to control the approximation error which can interpret the complexity of the neural network, i.e., how well the neural network approximates the least population risk. To achieve guarantees on the excess population risk, the approximation error should be sufficiently small, which can be interpreted as a complexity assumption of the learning problem. To be precise, the approximation error
> >
> > $$L(\mathbf{W}\^\*\_{\frac{1}{\eta T}} )+\frac{1}{2\eta T} || \mathbf{W}\^\*\_{\frac{1}{\eta T}} -\mathbf{W}\_0  ||_2^2 -  L(\mathbf{W}\^\*)$$
> >
> > $$\le \inf_{\mathbf{W}} L(\mathbf{W} ) + \frac{1}{2\eta T} || \mathbf{W}  -\mathbf{W}_0  ||_2^2  -  L(\mathbf{W}\^\*),$$
> > which  is further bounded by $\frac{1}{2\eta T}|| \mathbf{W}\^* - \mathbf{W}_0 ||_2^2 .$ In the above estimation, controlling the approximation error requires placing assumptions on the Frobenius norm of the neural network weights $\mathbf{W}\^\*.$ Hence, this norm $|| \mathbf{W}\^\*||$ serves as an indicator of the network's complexity. Assumption 3 states that the population risk minimizer $\mathbf{W}\^\*$ satisfies $|| \mathbf{W}^{*}||  \le m^{\frac{1}{2}-\mu}.$ Therefore, a large $\mu$ aligns with a stronger assumption which corresponds to a simpler network, as the minimum risk can be attained by a set of weights with a smaller norm. In addition, we would like to emphasize that the main results of our paper are excess population risk bounds for two-layer and three-layer neural networks (Theorems 3 and 7 in the revised version), which do not require Assumption 3.  We have added more discussion in the revised version.
> >
> >
> > [1] Richards, D. and Rabbat, M. Learning with gradient descent and weakly convex losses. In International Conference on Artificial Intelligence and Statistics, pp. 1990–1998. PMLR, 2021.
> >
> >
> >
> >
> > >** Q4. On page 2, the second bullet of contributions, the authors mentioned that "... due to the empirical risks' monotonically decreasing nature which no longer remains valid for three-layer NNs". Can't taking a smaller step in gradient descent steps help the empirical risks' monotonicity of decreasing?
> >
> >
> > Thanks for your thoughtful comment. This question is exactly related to the main challenge of doing stability analysis of GD for neural networks. Indeed, the monotonically decreasing of the empirical risks relies deeply on the smoothness of the loss. If the loss is $\rho$-smooth, according to the smoothness and the update rule of GD we can show that
> >
> > $L_S(\mathbf{W}\_\{t+1\})-L\_S(\mathbf{W}\_t)\le -\eta (1-\frac{\eta\rho}{2})||\nabla L\_S(\mathbf{W}\_t)||_2^2\le 0$
> >
> > with $\eta\le \rho/2$. It implies that the empirical risks decrease monotonically along the trajectory of GD. For two-layer neural networks, the smoothness of the loss can be shown by uniformly bounding the maximum eigenvalue of a Hessian matrix of the empirical risk $\lambda_{\max}(\nabla \ell(\mathbf{W};z))$. While for three-layer neural networks, the maximum eigenvalue $\lambda_{\max}(\nabla \ell(\mathbf{W};z))$ depends on $||\mathbf{W}||\_2$. We cannot find a uniform bound of $\lambda\_\{\max\}(\nabla \ell(\mathbf{W};z))$ and show the smoothness of the loss even though the stepsize is very small. Our key technical contribution lies in employing a novel induction strategy that fully explores the effects of over-parameterization to give an estimation of the norm of $\mathbf{W}$, and then providing an upper bound of $\lambda_{\max}(\nabla \ell(\mathbf{W};z))$ to show the smoothness of the loss for three-layer neural networks.

---

### Official Review · Reviewer_khQG · 2023-11-01

**Soundness:** 3 good
**Presentation:** 2 fair
**Contribution:** 3 good
**Rating:** 5
**Confidence:** 3

**Summary:**

This paper gives guarantees to the excess risk of two-layer and three-layer neural networks. In particular, this paper compares the generalization error of the gradient descent algorithm for minimizing the empirical risk of the network and provides bounds on the generalization error.

The setup assumes that the activation functions are twice-differentiable, have bounded derivatives, and the inputs all lie within a bounded range.

The main result requires the network width to grow in a polynomial fashion with the number of epochs, the step size, and a certain scaling parameter used in the empirical loss.

Additionally, it is assumed that the population risk minimizer satisfies a certain norm-bounded assumption under $\mu$.

Given these conditions, the paper then proves that the excess risk will converge by a rate as something like $O(T / n)$, where $T$ is the total number of iterations. [However, there is an additional dependence between $T$ and $m$ (the width of the network).

At a high-level, the proof proceeds by arguing that the gradient descent satisfies an "on-average" stability condition, and this condition implies that the loss is strongly smooth and weakly convex. This condition allows one to prove that the gradient descent optimizer will converge to the global minimum.

The scaling of $m^{-c}$ on the empirical loss is then used in a manner similar to the NTK analysis to get the required smoothness condition.

**Strengths:**

- A nontrivial result that expands on the literature of generalization guarantees for deep neural networks.

- The paper is generally written with sufficient clarity for readers to follow.

- A nice diagram to visualize the dependencies in the problem constants.

**Weaknesses:**

- The proof thus far only works for three layers. [Although it is believable that with more work, one may be able to extend the analysis for multiple layers, e.g., by arguing a similar weak convexity condition on the loss function.]

- It is by now widely accepted that the NTK analysis is more of an analogy rather than a realistic characterization of what happens in reality. Thus, the practical relevance of this paper, which follows similar lines of arguments as the NTK (e.g., by rescaling the loss and expanding the network width), is very limited.

- The abstract (and the paper's title) mentions multi-layer neural networks, but the analysis currently holds for three layers. This is a mismatch.

**Questions:**

- How much more difficulty would it take to extend the analysis to multiple layers?

- Do you see a way to argue that the dependence on width (in equation (4)) is tight/necessary?

---

> ### Author Response · Authors · 2023-11-20
> **Thank you for your invaluable and constructive comments**
>
> >** Q1. The proof thus far only works for three layers. [Although it is believable that with more work, one may be able to extend the analysis for multiple layers, e.g., by arguing a similar weak convexity condition on the loss function.] / The abstract (and the paper's title) mentions multi-layer neural networks, but the analysis currently holds for three layers. This is a mismatch.
>
> Thanks for your constructive comment. Following your excellent suggestion, we have changed our title to "Generalization guarantees of gradient descent for shallow neural networks", and have revised the abstract accordingly.
>
>
>
> >** Q2. It is by now widely accepted that the NTK analysis is more of an analogy rather than a realistic characterization of what happens in reality. Thus, the practical relevance of this paper, which follows similar lines of arguments as the NTK (e.g., by rescaling the loss and expanding the network width), is very limited.
>
> Thank you for your comment. In this work, we provide an alternative approach in a kernel-free regime to study the generalization guarantees of GD with neural networks, using the concept of algorithmic stability.  *One noteworthy merit of our stability analysis lies in its ability to bypass tailored assumptions frequently necessitated by prior works [1,2].* Notably, these assumptions encompass aspects like random initialization and data distribution/kernel matrices which have a positive eigenvalue. In this regard, we consider our stability analysis perspective to be both complementary to existing research and at the same time, sheds new light on understanding GD’s generalization in neural networks.
>
> Moreover, our generalization results show that for both two-layer and three-layer neural networks, *GD can achieve the dimension-independent generalization bound with only very milder overparameterization* (e.g., $c=1/2$ and $\mu=1/2$, $m\asymp n^{3/2}$,  $c=1$ and $\mu=1/3$, $m\asymp n^{3/2}$ for two-layer neural networks. More details can be found in Corollaries 4 and 8).
>
> For instance, Theorem 5.1 of [1] established generalization bounds of GD for two-layer ReLU neural networks with random initialization under an assumption on data distribution, very high overparameterization and the least eigenvalue of the kernel matrix. [2] introduced the neural tangent random feature (NTRF) model, and proved the generalization bounds of one-pass SGD for deep ReLU neural networks with random initialization in the over-parameterization regime, which requires the network weights to stay very close to their initialization throughout training. Their results imply with high overparameterization $m\ge n^7$ that *if the data can be classified by a function in the NTRF function class with a small training error*, a small classification error can be achieved with the over-parameterized ReLU network.
>
> To conclude, our stability-based results do not require conditions on the random initialization or the data distribution, and provide generalization bounds under milder overparameterization. In this sense, the perspective from our stability analysis sheds new light on comprehending GD's generalization in neural networks.
>
>
> [1] S. Arora, S. Du, W. Hu, et al. Fine-grained analysis of optimization and generalization for overparameterized two-layer neural networks. ICML 2019.
>
> [2] Y. Cao and Q. Gu. Generalization bounds of stochastic gradient descent for wide and deep neural networks. NeurIPS 2019.
>
> >** Q3. How much more difficulty would it take to extend the analysis to multiple layers?
>
> The key challenge of extending our results from three layers to multiple layers is to control the maximum and minimum eigenvalues of a Hessian matrix of the empirical risk, which rely on the norm of the coupled weights of different layers. Once having the estimates of eigenvalues, we can establish the almost co-coercivity of the gradient operator, which is crucial for the stability analysis. However, it's not easy to estimate the coupled weights of different layers by using the existing technique. We acknowledge the importance of considering the applicability of our findings to networks with multiple layers.  Since the existing stability analysis mainly focused on two-layer neural networks, we regard our work as an important starting point to study multiple layers for future research directions. We have added some discussion in the Conclusion to illustrate the main technical challenges in studying the multi-layer case via algorithmic stability.

---

> ### Author Response · Authors · 2023-11-20
>
> >** Q4. Do you see a way to argue that the dependence on width (in equation (4)) is tight/necessary?
>
> Thanks for the thoughtful comment. The dependence on width $m$ in equation (4) relies on estimating the maximum/minimum eigenvalue of the Hessian matrix of the empirical risk. Therefore, it may  not be precisely tight. It remains a challenging problem to explore what is the tight bound for width $m$.  However, as mentioned in Remark 1, [1] provided a similar generalization bound with an assumption $m\gtrsim (\eta T)^5/n^2 + (\eta T)^2$, and our result with $c=1/2$ is consistent with their result. Further, [2] derived a similar excess risk bound for $c=1/2$ (refer to Theorem 3 in the revised version) with $m\gtrsim (\eta T)^5$. We relax this condition to $m\gtrsim (\eta T)^3$ by providing a better estimation of the smallest eigenvalue of a Hessian matrix.
>
>
>
> [1]  Y. Lei,  R. Jin, and Y. Ying.  Stability and generalization analysis of gradient methods for shallow neural networks. In Advances in Neural Information Processing Systems, volume 35, 2022.
>
>
> [2] D. Richards and I. Kuzborskij. Stability $\\&$ generalisation of gradient descent for shallow neural networks without the neural tangent kernel. In Advances in Neural Information Processing Systems, volume 34, 2021.

---

### Official Review · Reviewer_51QV · 2023-11-01

**Soundness:** 3 good
**Presentation:** 3 good
**Contribution:** 3 good
**Rating:** 6
**Confidence:** 3

**Summary:**

This paper studies the generalization performance of multi-layer neural networks using stability-based techniques.
Compared to previous works, this paper extends previous results to multi-layer neural networks.
The primary focus revolves around the width requirements, specifically the scaling parameter (denoted as "m^-c," where "m" represents network width and "c" is closely associated with NTK and mean-field concepts), and the model capacity (quantified by the norm of W*, denoted as "μ"). The overarching conclusion of this study is that wider width requirements are necessary when "c" and "μ" are large. Additionally, the authors observe that the width requirements are less stringent for three-layer neural networks. In summary, I recommend a modest acceptance of this paper due to its foundational concepts and valuable technical contributions.

**Strengths:**

1. This paper offers a comprehensive perspective by encompassing various existing works. Notably, it addresses the applicability of the scaling factor, including scenarios such as NTK and mean field. While the primary focus is on three-layer neural networks, the authors consider the potential for direct application to networks with multiple layers.

2. A substantial technical contribution of this paper is the derivation of an upper bound for the W-norm, as presented in Lemma B.2. The writing is clear, and the proofs appear accurate, inspiring confidence in the results. However, a comprehensive validation of all details is still pending.

**Weaknesses:**

1. It seems that the authors primarily concentrate on three-layer neural networks. It would be beneficial if the authors explicitly discuss the potential applicability of their findings to networks with multiple layers.

2. The uniformity of scaling factors across layers may appear somewhat unconventional. It would be helpful if the authors provided further insight into this choice and its implications.

3. It would be beneficial if the authors could offer more intuitive explanations regarding how the upper bound on the W-norm improves from a t-scale to a sqrt{t}-scale, shedding light on the underlying mechanisms that drive this improvement.

**Questions:**

See weaknesses.

---

> ### Author Response · Authors · 2023-11-20
> **Thank you for your invaluable and constructive comments**
>
> >** Q1. It seems that the authors primarily concentrate on three-layer neural networks. It would be beneficial if the authors explicitly discuss the potential applicability of their findings to networks with multiple layers.
>
> Great point! We acknowledge the importance of considering the applicability of our findings to networks with multiple layers. Since the existing stability analysis mainly focused on two-layer neural networks, we regard our work as an important starting point to study multiple layers for future research directions. Indeed, extending our results from three layers to multiple layers is not straightforward. The main difficulty lies in estimating the maximum and minimum eigenvalues of a Hessian matrix of the empirical risk, which rely on the interconnected weights across layers. Controlling these interconnected weights using existing techniques presents considerable difficulty. In the Conclusion section, we've included discussions highlighting the principal technical obstacles in investigating multiple layers through algorithmic stability.
>
>
>
> >** Q2. The uniformity of scaling factors across layers may appear somewhat unconventional. It would be helpful if the authors provided further insight into this choice and its implications.
>
>
> Thank you for the constructive suggestion. To more explicitly illustrate the relationship between the scaling parameter $c$ and the over/under-parameterization of the network (as shown in Corollaries 4 and 8, and Figure 1), we make the assumption that both the width and scaling factors of each hidden layer are identical. Nevertheless, our findings can readily extend to scenarios where the widths and scaling factors vary across layers. In the revised version, we've included additional clarifications to accommodate this aspect.
>
>
> >** Q3. It would be beneficial if the authors could offer more intuitive explanations regarding how the upper bound on the W-norm improves from a t-scale to a $\sqrt{t}$-scale, shedding light on the underlying mechanisms that drive this improvement.
>
>
> Thank you for the constructive suggestion. As mentioned in Remark 4, for three-layer neural networks, the technical challenge in estimating $||\mathbf{W}_t||_2$ is closely tied to the smoothness of the loss function. Simultaneously, the smoothness parameter of the loss function relies on the norm of $\mathbf{W}_t$.   We overcame this difficulty by initially providing  a rough bound $||\mathbf{W}_t-\mathbf{W}_0||_2\le \eta t m^{2c-1}$ by induction. Once having the estimate of $||\mathbf{W}_t||_2$, we can upper bound the maximum eigenvalue of the Hessian by a constant $\hat{\rho}$ if $m$ satisfies (6). It implies that the loss is $\hat{\rho}$-smooth along the trajectory of GD.  Leveraging this smoothness property, we deduce that
>
> $\eta(1-\eta \hat{\rho}/2)\sum\_{j=0}^t||\nabla L_S(\mathbf{W}_j)||_2^2\le\sum\_{j=0}^t L_S(\mathbf{W}\_j)-L_S(\mathbf{W}\_{j+1})\le L_S(\mathbf{W}\_0).$
>
> Additionally, considering the update rule of GD and $\eta \hat{\rho}\le 1/2$, we conclude that  $||\mathbf{W}_t-\mathbf{W}_0||_2^2= \eta^2 ||\sum\_{j=0}^t \nabla L_S(\mathbf{W}_j)||_2^2 \le 2\eta t L_S(\mathbf{W}_0)$.
>
> We have added more details in Remark 4 in the revised version.

---

> > ### Comment · Reviewer_51QV · 2023-11-21
> > **Thank you for your response.**
> >
> > I thank the authors for the response and keep my score unchanged.
> > I believe that extending the existing results to multi-layers, if possible, could greatly enhance the current manuscript.
> > Given the current version, I just keep my score unchanged.

---

### Official Review · Reviewer_BPeY · 2023-11-01

**Soundness:** 3 good
**Presentation:** 2 fair
**Contribution:** 2 fair
**Rating:** 5
**Confidence:** 3

**Summary:**

The paper provides generalization guarantees for the training of neural networks with one and two hidden layers (i.e., with two and three layers) under gradient descent. The paper shows what combinations of the scaling parameter and the so-called complexity parameter, achieves the generalization error bound $O(1/\sqrt{n})$, where $n$ is the number of samples, for both over- and under-parameterized regimes. Moreover, the paper shows that when an adequate minimizer (weight values for a network) that zeroes the population risk exists, their generalization error improves to $O(1/n)$.

**Strengths:**

-> The paper analyzes different scaling on the network’s width by varying its exponent $c$, defining regimes that haven’t been analyzed in previous works.

-> The paper does a full formal analysis for both two and three layered networks, and show the difficulties on such analyses and their differences, particularly, when it comes to the (almost) co-coercivity property.

-> Discussions are provided after the theoretical results and comparison to existing literature which aids the reader’s understanding of their nature and contribution.

**Weaknesses:**

There are assumptions, concepts, and results that need to be better explained and for which I also raise concerns related to their understanding and implications.


-> In Assumption 2, one of the conditions seem to depend on a specific initialization of the weights $W_0$, where that the loss should be less than $c_0$ when evaluated under the initialization. However, nowhere in the paper mentions (at least to that point) how the networks are initialized or if we are assuming a fixed initialization – I am not sure if the bound holds for $c_0$ uniformly over any initialization of the weights or not. Please, clarify.


-> In Theorem 3, it says that equation (4) should hold, but then show another bound on $m$ in equation (5). How should we interpret this? Should we just take the maximum of both lower bounds on $m$?


-> Regarding Theorem 4: I would imagine that it would be a good thing for the excess population risk to minimize as the width of the network increases, i.e., as there is over-parameterization. For this, $c>1/2$ is desirable. Then, why would someone choose $c=1/2$ from the perspective of the excess population risk or from the other terms that bound the generalization bound? Why did previous works considered $c=1/2$ to be a good thing?

So, it seems that having $c>1/2$ benefits decreasing the bound, or at least the excess population risk part of it, as $m$ increases. Therefore, I would expect in Corollary 4 some differentiation in terms of $c=1/2$ and $c>1/2$, but I don’t see such difference (the only difference is made over the value $c=3/4$).


-> Following the previous point, it is interesting that the excess population risk for a three-layer neural network in Theorem 8 (excess population risk) does not have a term $m$, and so no dependency on $c$ in the upper bound, unlike its counterpart Theorem 4. Moreover, Theorem 8 does not hold for $c=1/2$ according to its statement. Why is this? How is the derivation here different from the two-layer network?

Moreover, I would expect a $c$ factor to appear in the upper bound because the scaling that depends on $c$ is on the output layer in both two or three layers. Is there any explanation for this? I haven’t checked the math.


-> I am trying to understand Assumption 3 and see if it makes sense. Right after equation (2) it is mentioned that $W^*$ has minimum norm. Is Assumption 3 just assuming that such minimum norm must satisfy an upper bound?


-> Also, Assumption 3 introduces the parameter $\mu$ almost arbitrarily – there is no mention whether the value of $\mu$ comes from the architecture of the network of the loss function, it seems like a magical parameter! Moreover, it is afterwards that the paper calls $\mu$ the network complexity, but I think this is a void term. For example, the value of $\mu$ determines whether the minimizer with minimum norm will have its norm decreasing as the width $m$ increases or not. How is this possible? What is the motivation for this? Whether the norm increases or not depending on the width, since we simply have a two-layer network, should depend on the architecture and the loss itself, not on some external and extraneous parameter. This requires explanation.


-> So the terms “over-parameterized” and “under-parameterized” are used extensively in the paper without a proper definition. It may sound obvious that in the former you have more parameters than data samples, while in the latter the opposite; however, throughout the paper there are different conditions on the width $m$ that makes the distinction to both regimes. Can the exact definitions be defined? For example, under the conditions on $m$ in Corollary 5 and 9, how do I know in which regime I am? Could more explanation on how to define both regimes be given for Figure 1?


==

-> Scaling parameter is mentioned in the contributions without any explanation of what this exactly mean. I don’t think it is a standard term: it could mean scaling of the initialization variance, the scaling of the pre-activation, etc.

-> Also the term “network complexity” is mentioned in the introduction without much explanation. Such term seems to be independent from whether the network is over- or under- parameterized (which is kind of strange, since I remember that “complexity of the model” usually refers to how many parameters or degrees of freedom it has).

==

-> Though the work (Taheri & Thrampoulidis (2023)) use algorithmic stability, it seems they make strong assumptions on the distribution of the data, which should be mentioned in the paper.

-> The norm symbol for Euclidean norm is defined, but not for the spectral norm (when the argument is a matrix).

-> Theorem 6 and Remark 4 mention the “condition (B.9)”, however, such condition seems to not be on the main paper, making the paper not self-contained.

-> Besides the introduction, the property of almost co-coercivity is not mentioned until section 3.2. It would be a good idea to introduce its definition from the beginning.

**Questions:**

Please, see the "Weaknesses" section above.

---

> ### Author Response · Authors · 2023-11-20
> **Thank you for your careful reading and constructive comments**
>
> >**Q1. In Assumption 2, one of the conditions seem to depend on a specific initialization of the weights $\mathbf{W}_0$, where the loss should be less than $c_0$ when evaluated under the initialization. However, nowhere in the paper mentions (at least to that point) how the networks are initialized or if we are assuming a fixed initialization – I am not sure if the bound holds for $c_0$ uniformly over any initialization of the weights or not. Please, clarify.
>
> Thank you for this thoughtful comment. We would like to emphasize that our results hold for both a random initialization setting and a fixed initialization setting *as long as $\ell(\mathbf{W}_0;z)$ is uniformly bounded by a constant $c_0$*. Indeed, one can verify that such uniform bound $c_0$ holds true for some widely used activations including sigmoid activation and hyperbolic tangent activation for any initialization. Since we assume that $\mathbf{x}$ and $y$ are bounded,    $f_{\mathbf{W}_0}(\mathbf{x})$ is uniformly bounded for both two-layer and three-layer cases by noting that the sigmoid and hyperbolic tangent activation functions $\sigma(\mathbf{w}_k\^\top \mathbf{x})\le 1$ for any $k$. Consequently, for any $\mathbf{W}_0$,  the least square loss $\ell(\mathbf{W}_0;z)$ is uniformly bounded by a constant $c_0$.
>
> In addition, the assumption of the loss is standard in the stability and generalization analysis of GD with neural networks [1, 2]. We have added the clarification below Assumption 2 and above Theorem 3 in the revised version.
>
> [1] D. Richards and I. Kuzborskij. Stability $\\&$ generalisation of gradient descent for shallow neural networks without the neural tangent kernel. In Advances in Neural Information Processing Systems, volume 34, 2021.
>
> [2]  Y. Lei,  R. Jin, and Y. Ying.  Stability and generalization analysis of gradient methods for shallow neural networks. In Advances in Neural Information Processing Systems, volume 35, 2022.
>
>
>
> >** Q2.  In Theorem 3, it says that equation (4) should hold, but then show another bound on $m$ in equation (5). How should we interpret this? Should we just take the maximum of both lower bounds on $m$?
>
>
> Yes, we take the maximum of both lower bounds on $m$ to ensure that our main result for two-layer NNs (Theorem 3 in the revised version) holds. This is what we did in Corollary 4, i.e., choosing $m\gtrsim (\eta T)^{\frac{3}{2c}}$ such that the maximum of the lower bounds in (4) and (5) hold. Notably, the generalization error bounds in Theorem 1 hold under condition (4). Introducing condition (5), we further establish the optimization error bounds in Theorem 2. Combining these two results together, we derive the excess risk bounds in Theorem 3 when (4) and (5) hold simultaneously.

---

> > ### Author Response · Authors · 2023-11-20
> >
> > >** Q3. Regarding Theorem 4: I would imagine that it would be a good thing for the excess population risk to minimize as the width of the network increases, i.e., as there is over-parameterization. For this, $c>1/2$ is desirable. Then, why would someone choose $c=1/2$ from the perspective of the excess population risk or from the other terms that bound the generalization bound? Why did previous works considered $c=1/2$ to be a good thing?
> > So, it seems that having $c>1/2$ benefits decreasing the bound, or at least the excess population risk part of it, as $m$ increases. Therefore, I would expect in Corollary 4 some differentiation in terms of $c>1/2$ and $c=1/2$,  but I don’t see such difference (the only difference is made over the value $c=3/4$.
> >
> > Thanks for your careful reading. We first want to clarify that Theorem 4 (Theorem 3 in the revised version) implies that the excess risk bound becomes better for larger $c$ *only when the effect of $\mathbf{W}\^\*\_{\frac{1}{\eta T}}$ and $L(\mathbf{W}\^\*)$ are ignored.*  This result may not hold in the general case since $\mathbf{W}\^\*\_{\frac{1}{\eta T}}$ and $L(\mathbf{W}\^\*)$ may also depend on $c$. We have deleted any misleading words and discussions in the revised version.
> >
> > The popular choice of $c=1/2$ is due to its connection to the Neural Tangent Kernel (NTK) regime. Loosely speaking, under sufficient overparameterization and Gaussian initialization, the dynamics of GD on shallow neural networks are close to the dynamics of GD in RKHSs with NTK and thus the uniform-convergence-based generalization analysis for GD in the convex setting can apply, which, however, usually requires a huge overparameterization. For example, Theorem 5.1 of [1] established generalization bounds of GD for two-layer ReLU neural networks with random initialization under assumptions that the least eigenvalue of the kernel matrix is positive and very high overparameterization (at least $m\gtrsim n^6$).  In this paper, we establish the generalization results of GD without requirements on the random initialization or the data distribution under milder overparameterization (e.g., $c=1/2$ and $\mu=1/2$, $m \asymp n^{3/2}$) in the kernel-free regime via algorithmic stability, where $c$ plays an important role in our analysis since it affects the geometry of the loss function.
> >
> > Corollary 4 delineates a gradual transition from $c=1/2$ to $c=1$. Specifically, under Assumption 3 (network complexity assumption) we demonstrate that, for the fixed $\mu$, the lower bound of $m$ becomes smaller as $c$ increases (i.e., large $c$ relaxes the requirement to the width $m$ ) for GD to achieve the excess risk rate $O(1/\sqrt{n})$. Taking a different viewpoint, from Figure 1 we can see that for larger $c$, the choice of $\mu$ is more flexible for GD to achieve the desired excess risk rate. In essence, GD can learn more complex $\mathbf{W}\^\*$ with a larger $c$.  In this sense, $c>1/2$ has an advantage over $c=1/2$.   The distinction at $c=3/4$ is mainly due to the proof artifacts. We have added more clarification in the paper.
> >
> > [1] S. Arora, S. Du, W. Hu, et al. Fine-grained analysis of optimization and generalization for overparameterized two-layer neural networks. ICML 2019.

---

> > > ### Author Response · Authors · 2023-11-20
> > >
> > > >** Q4. Following the previous point, it is interesting that the excess population risk for a three-layer neural network in Theorem 8 (excess population risk) does not have a term $m$, and so no dependency on $c$ in the upper bound, unlike its counterpart Theorem 4. Moreover, Theorem 8 does not hold for $c=1/2$ according to its statement. Why is this? How is the derivation here different from the two-layer network?
> > > Moreover, I would expect a factor $c$ to appear in the upper bound because the scaling that depends on $c$ is on the output layer in both two or three layers. Is there any explanation for this? I haven’t checked the math.
> > >
> > > Thank you for this thoughtful comment. In our analysis, different estimates of the smoothness parameter lead to distinct results for two-layer and three-layer neural networks presented in Theorem 4 and Theorem 8 (refer to Theorem 3 and Theorem 7 in the revised version), respectively. Indeed, for both two-layer and three-layer neural networks, we show that the excess population risk is upper bounded by $\frac{\eta T \rho}{n}L(\mathbf{W}^*) + \Lambda_{\frac{1}{\eta T}}$ (see the proofs of Theorems 4 and 8 for more details). Here $\rho$ is the smoothness parameter of the loss. As mentioned in Remark 4 and Section 4, we show that $\rho=O(m^{1-2c})$ for two-layer neural networks, while we can only show that $\rho=O(1)$ for three-layer neural networks (our technical contribution also lies here, see Remark 4 for more details). Hence, the excess population risk for three-layer neural networks in Theorem 8 does not have a term $m$.
> > >
> > > Theorem 8 does not hold for $c=1/2$ also due to the challenge of estimating $\rho$ for the three-layer case.  Specifically, we estimate $\rho$ by upper bounding the maximum eigenvalue of a Hessian matrix of the empirical risk $\lambda_{\max}(\ell(\mathbf{W};z)$. For two-layer neural networks, we can bound $\lambda_{\max}(\ell(\mathbf{W};z)$ by $m^{1-2c}$ uniformly. However, for three-layer neural networks, we observe that $\lambda_{\max}(\ell(\mathbf{W};z)\le \rho_{\mathbf{W}}$ where $\rho_{\mathbf{W}}$ depends on the norm of $\mathbf{W}$. Conversely, the estimation of the norm of $\mathbf{W}$ relies on the smoothness of the loss. Hence, the analysis of three-layer neural networks is more challenging. To tackle this challenge, we first establish a crude estimate of $||\mathbf{W}\_t-\mathbf{W}\_0||\_2=O(\eta t m\^\{2c-1\})$ for $c>1/2$ by the induction strategy. By using this estimate, we prove that $\rho_{\mathbf{W}}=O(1)$ if $m$ is large enough. For the case $c=1/2$, the upper bound of $||\mathbf{W}_t-\mathbf{W}_0||_2$ contains a worse term $2^t$, which is not easy to control. Therefore, we only consider $c>1/2$ for three-layer neural networks. It remains a challenging problem to develop a result specifically for this setting. More details can be found in Remark 4 and Section 4.
> > >
> > > >** Q5. Is Assumption 3 just assuming that such minimum norm must satisfy an upper bound?
> > >
> > > Yes, Assumption 3 assumes that the minimum norm of $\mathbf{W}^{*}$ has an upper bound.

---

> > > > ### Author Response · Authors · 2023-11-20
> > > >
> > > > >** Q6. Also, Assumption 3 introduces the parameter $\mu$ almost arbitrarily – there is no mention whether the value of $\mu$ comes from the architecture of the network of the loss function, it seems like a magical parameter! Moreover, it is afterwards that the paper calls $\mu$ the network complexity, but I think this is a void term. For example, the value of $\mu$ determines whether the minimizer with minimum norm will have its norm decreasing as the width $m$ increases or not. How is this possible? What is the motivation for this? Whether the norm increases or not depending on the width, since we simply have a two-layer network, should depend on the architecture and the loss itself, not on some external and extraneous parameter. This requires explanation.
> > > >
> > > > We followed the same assumption in [1] to control the approximation error which can interpret the complexity of the neural network, i.e., how well the neural network approximates the least population risk. To achieve guarantees on the excess population risk, the approximation error should be sufficiently small, which can be interpreted as a complexity assumption of the learning problem.  Indeed, the approximation error
> > > >
> > > > $$L(\mathbf{W}\^\*\_\{\frac{1}{\eta T}\} ) + \frac{1}{2\eta T} || \mathbf{W}\^\*\_\{\frac{1}{\eta T}\}  -\mathbf{W}\_0  ||_2^2  -  L(\mathbf{W}\^\*)\le \inf\_\{\mathbf{W}\} L(\mathbf{W} ) + \frac{1}{2\eta T} || \mathbf{W}  -\mathbf{W}\_0  ||\_2^2  -  L(\mathbf{W}\^\*),$$
> > > >
> > > > which  is further bounded by $\frac{1}{2\eta T}|| \mathbf{W}\^\* - \mathbf{W}_0 ||_2^2 .$ In the above estimation, controlling the approximation error requires placing assumptions on the Frobenius norm of the neural network weights $|| \mathbf{W}\^\*||.$ Hence, this norm $|| \mathbf{W}\^\*||$ serves as an indicator of the network's complexity. Assumption 3 states that the population risk minimizer $\mathbf{W}\^\*$ satisfies $|| \mathbf{W}\^\*||  \le m\^\{\frac{1}{2}-\mu\}.$ Therefore, a large $\mu$ aligns with a stronger assumption which corresponds to a simpler network, as the minimum risk can be attained by a set of weights with a smaller norm. In addition, we would like to emphasize that the main results of our paper are excess population risk bounds for two-layer and three-layer neural networks (Theorems 3 and 7 in the revised version), which do not require Assumption 3.  We have added more discussion in the revised version.
> > > >
> > > >
> > > > [1] Richards, D. and Rabbat, M. Learning with gradient descent and weakly convex losses. In International Conference on Artificial Intelligence and Statistics, pp. 1990–1998. PMLR, 2021.
> > > >
> > > >
> > > > >** Q7. So the terms “over-parameterized” and “under-parameterized” are used extensively in the paper without a proper definition. It may sound obvious that in the former you have more parameters than data samples, while in the latter the opposite; however, throughout the paper there are different conditions on the width $m$ that makes the distinction to both regimes. Can the exact definitions be defined? For example, under the conditions on $m$ in Corollary 5 and 9, how do I know in which regime I am? Could more explanation on how to define both regimes be given for Figure 1?
> > > >
> > > > Thanks for the constructive suggestion. We call the network over-parameterized (resp. under-parameterized) if $m\gtrsim n$ (resp. $m\lesssim n$). We have added definitions below Corollary 4 and at the top of page 6.
> > > >
> > > > In the left panel of Figure 1, the horizontal axis corresponds to the value of $c$, the vertical axis corresponds to the value of $\mu$, and the dotted line $\mu=3/4-3/c$ corresponds to the setting $m \asymp n$. The region above the dotted line (i.e., blue Region with dots) corresponds to the under-parameterization region ($m\lesssim n$ in this region). The region below the dotted line (i.e., blue Region without dots) corresponds to the over-parameterization region  ($m\gtrsim n$ in this region).
> > > > If values of $c$ and $\mu$ are located above the dotted line, i.e., the region blue region with dots, then we know that under-parameterization networks are enough to achieve the risk rate $O(1/\sqrt{n})$. If values of $c$ and $\mu$ are located in the region blue region without dots which is between the solid lines and the dotted line, then over-parameterization is necessary for achieving the error rate $O(1/\sqrt{n})$. That is, only over-parameterized networks can achieve the desired rate in this region.  More explanation can be found in "Interpretation of the Results" on page 5.

---

> > > > > ### Author Response · Authors · 2023-11-20
> > > > >
> > > > > >** Q8. Scaling parameter is mentioned in the contributions without any explanation of what this exactly mean.
> > > > >
> > > > > Thanks for your suggestion. We have added explanations of  network scaling and scaling parameters in the abstract and at the beginning of Section 2.
> > > > >
> > > > >
> > > > >
> > > > > >** Q9. The term “network complexity” is mentioned without much explanation.
> > > > >
> > > > > Thanks for your suggestion. We have added the discussion in Section 3.1 in the revised version.
> > > > >
> > > > >
> > > > >
> > > > > >** Q10. The work (Taheri $\\&$ Thrampoulidis (2023)) should be mentioned in the paper.
> > > > >
> > > > > Thanks for mentioning this work. We have added a discussion in the introduction.
> > > > >
> > > > >
> > > > > >** Q11. The norm symbol for Euclidean norm is defined, but not for the spectral norm (when the argument is a matrix).
> > > > >
> > > > > Thanks for your careful reading.  We have defined the spectral norm in Section 2.
> > > > >
> > > > >
> > > > > >** Q12. Theorem 6 and Remark 4 mention the “condition (B.9)”, however, such condition seems to not be on the main paper, making the paper not self-contained. Besides the introduction, the property of almost co-coercivity is not mentioned until section 3.2. It would be a good idea to introduce its definition from the beginning.
> > > > >
> > > > > Thanks for your helpful suggestions. We have revised the paper according to your suggestions.

---

### Author Response · Authors · 2023-11-22
**Thank you for taking the time to review the paper**

We would like to thank all the reviewers for the constructive comments given in the initial reviews. We have uploaded the revised revision of the submission with changes according to the reviewers' constructive suggestions.

As the discussion period comes to a close, we would like to gently request that reviewers BPeY, khQG and PsiY confirm whether their questions have been answered. We sincerely hope that the updated version is satisfactory.  Please let us know if there is any other question or concern we could address to merit a higher score.

Thank you for taking the time to review and help us improve our submission.


Best Regards,

The Authors

---

### Meta-Review · Area_Chair_o631 · 2023-12-14

**Metareview:**

This paper gives a new generalization error bound for three layer neural networks that takes optimization into account. The bound is characterized by two scaling factor $\mu$ and $c$ that control norm of the optimal parameter and the network parameterization scale respectively. Then, the authors argue when $1/\sqrt{n}$ convergence is attainable in terms of choice of $\mu$ and $c$.

The analysis is technically novel. That would bring some insight to the community. On the other hand, the results are not necessarily presented in a digestible way. Key assumptions require further explanations for its plausibility. The authors addressed these issues in the rebuttal, but it appears that the paper requires substantial revision to yield sufficient improvement for publication.
For these reasons, I unfortunately recommend rejection at this stage. I personally recommend the authors to polish the manuscript so that the plausibility of assumptions becomes much more clear and the insight obtained by the analyses can be more directly explained.

**Justification For Why Not Higher Score:**

As I mentioned in meta-review, the current status of the manuscript is not easy to grasp its meaning. Especially, justification of assumptions is not sufficient, and insight from the theories is still not well exposed.

**Justification For Why Not Lower Score:**

N/A

---

### Decision · Program_Chairs · 2024-01-16

Reject